MITP/24-086

# Electroweak double-box integrals for Møller scattering

Niklas Schwanemann and Stefan Weinzierl

*PRISMA Cluster of Excellence, Institut für Physik,*

*Johannes Gutenberg-Universität Mainz,*

*D - 55099 Mainz, Germany*

**Abstract**

We present for Møller scattering all planar and non-planar two-loop double-box integrals with the exchange of three electroweak gauge bosons, among which at least one is a photon. These integrals are relevant for the NNLO electroweak corrections to Møller scattering.

# 1 Introduction

Møller scattering ($e^-e^- \rightarrow e^-e^-$) can be used to measure the weak mixing angle at low energies [1]. In order to match the anticipated experimental precision reliable calculations on the theory side are required. As higher precision on the theory side is reached by including higher-order perturbative corrections, electroweak two-loop corrections come into focus [2–10]. The most complicated Feynman integrals at the two-loop level relevant to this process are the planar and the non-planar double-box integrals. These integrals are the topic of this article.

We further note that the scattering amplitudes for Bhabha scattering ($e^+e^- \rightarrow e^+e^-$) and Møller scattering are related by crossing symmetry, and the Feynman integrals computed in this paper are useful for both processes. Bhabha scattering is typically used to monitor the luminosity at an $e^+e^-$-collider [11].

In this paper we present all planar and non-planar two-loop double-box integrals with the exchange of three electroweak gauge bosons, with the additional requirement that at least one exchanged gauge-boson is a photon. Throughout this paper we treat electrons (and neutrinos) as massless. The non-zero kinematic variables are the two Mandelstam variables $s$ and $t$ (the third Mandelstam variable is given by $u = -s - t$) and the mass $m$ of the heavy gauge boson (either a $Z$-boson or a $W$-boson). Diagrams with two massive gauge bosons will necessarily have either two $Z$-bosons or two $W$-bosons. The mixed case is excluded by the assumed initial and final states. For each diagram we distinguish the mass configuration: We consider the cases with zero, one or two internal massive gauge bosons. For one and two internal massive gauge bosons we have for each topology two inequivalent possibilities to assign the masses, leading to five different mass configurations for each topology. The complexity of the calculation increases with the number of internal massive gauge bosons: The simplest case are the Feynman integrals with three internal photons. These have been computed long time ago [12, 13]. This case is included here only for completeness. On the other side, the non-planar integrals with two internal massive gauge bosons involve elliptic curves and require state-of-the-art techniques. This paper is an example, how techniques developed for elliptic Feynman integrals [14–77] are applied to phenomenological relevant processes.

In total we present ten families of Feynman integrals, which differ by the topology (planar or non-planar) and the mass configuration. We label the planar families $A$, $B$, $C$, $D$ and $E$, where $A$ denotes the most complicated one and $E$ the simplest one. The non-planar families are labelled in a similar way by $\tilde{A}$, $\tilde{B}$, $\tilde{C}$, $\tilde{D}$ and $\tilde{E}$. The planar double-box diagrams and the non-planar double-box diagrams are shown in fig. 1.

We do not include in this paper the exchange of three massive gauge bosons. There are two reasons for this: On the one hand we expect the contributions from these integrals to Møller scattering to be suppressed by a (negative) power of the heavy gauge boson mass. On the other hand (and this is the main reason), this case goes beyond the current state-of-the-art: it is known that the non-planar double-box integral with three internal massive gauge bosons is related to a curve of genus two [78]. We will report on these integrals in a future publication.

To compute the Feynman integrals we follow to a large extent the set-up and the notation used in [79], where the (simpler) box integrals with self-energy insertion have been computed for Møller scattering: Using integration-by-parts identities [80] we first derive a differential equation

[81–83] for a pre-canonical basis of master integrals. This differential equation is in general not in an ε-factorised form. The essential step of our work is to construct a new basis, such that the differential equation is transformed to an ε-factorised form [84]. As some Feynman integrals are related to elliptic integrals, we profit from the techniques developed in [47,48,85]. We determine for all integrals the boundary values. We then solve the ε-factorised differential equation. We express the results for the simpler topologies in terms of multiple polylogarithms, the results for the more complicated topologies in terms of iterated integrals. For all integrals we provide numerical evaluation routines. Of particular interest are numerically large contributions. These arise from large logarithms. The large logarithms are easily obtained from our full result.

This paper is organised as follows: In the next section we introduce our notation. In section 3 we present the master integrals, which put the differential equation into an ε-factorised form. In section 4 we discuss the integration of the ε-factorised differential equation. Numerical results for a benchmark point are provided in section 5. In section 6 we discuss large logarithms. Finally, section 7 contains our conclusions. The article includes two appendices: In appendix A we show for all master integrals the corresponding diagrams. In appendix B we describe the content of the supplementary electronic file attached to the arxiv version of this article.

## 2   Notation and set-up

### 2.1   Kinematics

We consider two-loop electroweak corrections to Møller scattering

$$e^-(-p_1) + e^-(-p_2) \rightarrow e^-(p_3) + e^-(p_4). \tag{1}$$

It will be convenient to take all momenta as outgoing, hence the momenta of the two incoming particles have a minus sign. Momentum conservation reads

$$p_1 + p_2 + p_3 + p_4 = 0. \tag{2}$$

We assume the electrons to be massless. This implies that the external momenta satisfy

$$p_1^2 = p_2^2 = p_3^2 = p_4^2 = 0. \tag{3}$$

The Mandelstam variables are defined as usual by

$$s = (p_1 + p_2)^2, \quad t = (p_2 + p_3)^2, \quad u = (p_1 + p_3)^2. \tag{4}$$

The most complicated loop diagrams contributing to the electroweak NNLO corrections are the planar and non-planar double-box diagrams, shown in fig. 1. The wavy lines are either photons or heavy gauge bosons (i.e. either $Z$-bosons or $W$-bosons). The case where three heavy gauge bosons are exchanged is expected to be numerically suppressed by $|t|/m^2$ [4], where $m$ denotes the mass of a heavy gauge boson. In addition it is from a computational point of view signifi-cantly more involved. For these reasons we focus here on the cases, where there is a least one

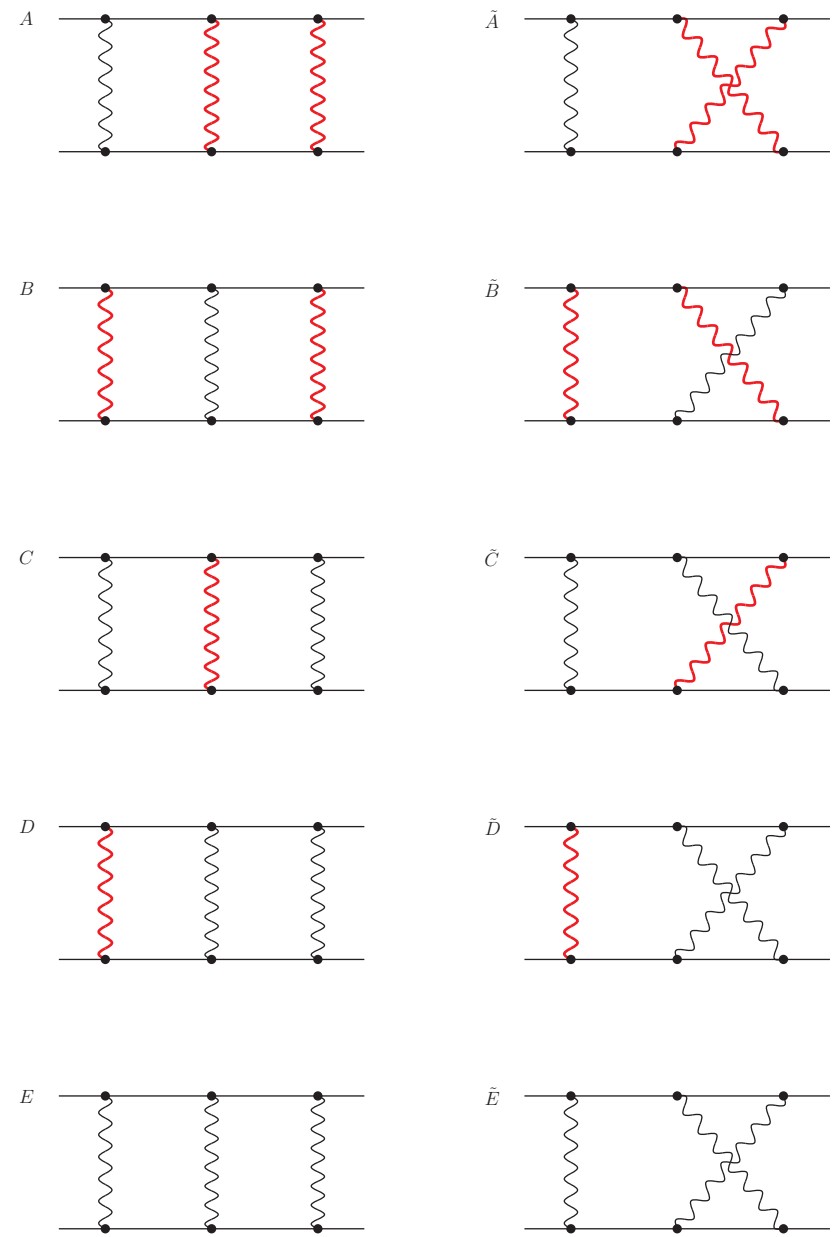

Figure 1: The planar double-box diagrams (left) and the non-planar double-box diagrams (right). The solid lines are electrons, the black wavy lines are photons and the red wavy lines are heavy gauge bosons.

photon exchanged. In this case there will be at most two massive gauge bosons. If there are two massive gauge bosons, they will have the same mass $m$.

We are interested in the case where

$$-t \; \lesssim \; s \; \ll \; m^2. \tag{5}$$

## 2.2 The families of Feynman integrals

We consider the integrals

$$I^X_{v_1 v_2 v_3 v_4 v_5 v_6 v_7 v_8 v_9} = e^{2\gamma_E \varepsilon} \left(\mu^2\right)^{v-D} \int \frac{d^D k_1}{i\pi^{\frac{D}{2}}} \frac{d^D k_2}{i\pi^{\frac{D}{2}}} \prod_{j=1}^{9} \frac{1}{\left(P^X_j\right)^{v_j}},$$

$$X \in \{A, B, C, D, E, \tilde{A}, \tilde{B}, \tilde{C}, \tilde{D}, \tilde{E}\}, \tag{6}$$

where $D = 4 - 2\varepsilon$ denotes the number of space-time dimensions, $\gamma_E$ denotes the Euler-Mascheroni constant, $\mu$ is an arbitrary scale introduced to render the Feynman integral dimensionless, and the quantity $\nu$ is defined by

$$\nu = \sum_{j=1}^{9} v_j. \tag{7}$$

We will further use the notation $p_{ij} = p_i + p_j$, $p_{ijk} = p_i + p_j + p_k$. The labels $\{A, B, C, D, E\}$ denote planar topologies, the labels $\{\tilde{A}, \tilde{B}, \tilde{C}, \tilde{D}, \tilde{E}\}$ denote non-planar topologies. The inverse propagators for the planar topologies are given by

$$
\begin{aligned}
P^X_1 &= -\left(k_1 - p_1\right)^2 + \left(m^X_1\right)^2, & P^X_2 &= -\left(k_1 - p_{12}\right)^2, & P^X_3 &= -k_1^2, \\
P^X_4 &= -\left(k_1 + k_2\right)^2 + \left(m^X_4\right)^2, & P^X_5 &= -\left(k_2 + p_{12}\right)^2, & P^X_6 &= -k_2^2, \\
P^X_7 &= -\left(k_2 + p_{123}\right)^2 + \left(m^X_7\right)^2, & P^X_8 &= -\left(k_1 - p_{13}\right)^2, & P^X_9 &= -\left(k_2 + p_{13}\right)^2.
\end{aligned} \tag{8}
$$

The auxiliary graph for the planar topologies is shown in fig. 2. The planar topologies $\{A, B, C, D, E\}$ differ by the assignment of the masses $(m^X_1, m^X_4, m^X_7)$. The cases which we consider in this paper are summarised in table 1. The first graph polynomial [86] for the auxiliary graph shown in fig. (2) reads

$$
\begin{aligned}
\mathcal{U}^X &(a_1, a_2, a_3, a_4, a_5, a_6, a_7, a_8, a_9) = \\
&(a_1 + a_2 + a_3 + a_8)(a_5 + a_6 + a_7 + a_9) + a_4(a_1 + a_2 + a_3 + a_5 + a_6 + a_7 + a_8 + a_9). \\
&X \in \{A, B, C, D, E\}.
\end{aligned} \tag{9}
$$

The inverse propagators for the non-planar topologies $\{\tilde{A}, \tilde{B}, \tilde{C}, \tilde{D}, \tilde{E}\}$ are given by

$$
\begin{aligned}
P^X_1 &= -\left(k_1 - p_1\right)^2 + \left(m^X_1\right)^2, & P^X_2 &= -\left(k_1 - p_{12}\right)^2, & P^X_3 &= -k_1^2, \\
P^X_4 &= -\left(k_1 + k_2\right)^2 + \left(m^X_4\right)^2, & P^X_5 &= -\left(k_{12} + p_3\right)^2, & P^X_6 &= -k_2^2,
\end{aligned}
$$

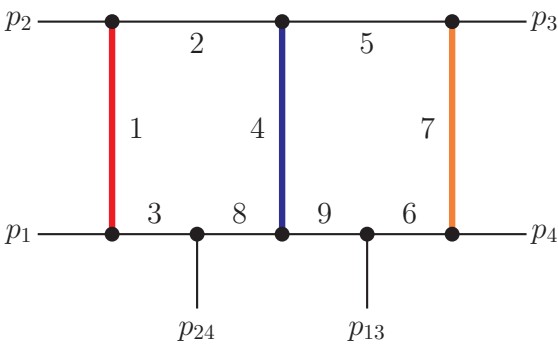

Figure 2: The auxiliary graph for the planar topologies. The coloured legs carry the masses $m_1^X$, $m_4^X$ and $m_7^X$, respectively, which can either be the mass of a heavy gauge boson or zero. All other lines are massless.

| Topology | Masses | # MIs | Square roots |
|----------|--------|-------|--------------|
| A | $(m_1^A, m_4^A, m_7^A) = (0, m, m)$ | 35 | $r_1, r_2, r_3, r_4$ |
| B | $(m_1^B, m_4^B, m_7^B) = (m, 0, m)$ | 26 | $r_1, r_4$ |
| C | $(m_1^C, m_4^C, m_7^C) = (0, m, 0)$ | 19 | – |
| D | $(m_1^D, m_4^D, m_7^D) = (m, 0, 0)$ | 21 | – |
| E | $(m_1^E, m_4^E, m_7^E) = (0, 0, 0)$ | 8 | – |

Table 1: The planar topologies and the corresponding mass configurations, number of master integrals and occurring square roots.

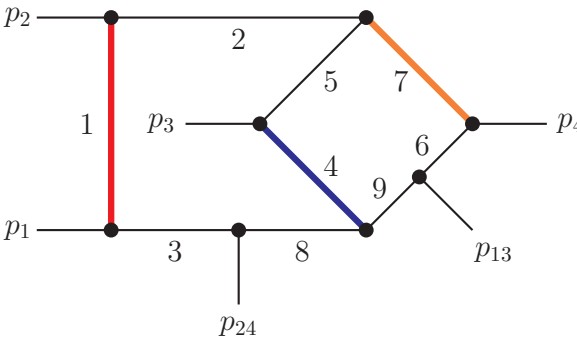

Figure 3: The auxiliary graph for the non-planar topologies. The coloured legs carry the masses $m_1^X$, $m_4^X$ and $m_7^X$, respectively, which can either be the mass of a heavy gauge boson or zero. All other lines are massless.

| Topology | Masses | # MIs | Square roots | Elliptic curves |
|----------|--------|-------|--------------|-----------------|
| $\tilde{A}$ | $(m_1^{\tilde{A}}, m_4^{\tilde{A}}, m_7^{\tilde{A}}) = (0, m, m)$ | 48 | $r_1, r_2, r_3, r_6,$ | $E^{(b)}, E^{(c)}$ |
| $\tilde{B}$ | $(m_1^{\tilde{B}}, m_4^{\tilde{B}}, m_7^{\tilde{B}}) = (m, 0, m)$ | 64 | $r_1, r_3, r_4, r_7, r_8$ | $E^{(a)}$ |
| $\tilde{C}$ | $(m_1^{\tilde{C}}, m_4^{\tilde{C}}, m_7^{\tilde{C}}) = (0, m, 0)$ | 41 | $r_5$ | $-$ |
| $\tilde{D}$ | $(m_1^{\tilde{D}}, m_4^{\tilde{D}}, m_7^{\tilde{D}}) = (m, 0, 0)$ | 23 | $-$ | $-$ |
| $\tilde{E}$ | $(m_1^{\tilde{E}}, m_4^{\tilde{E}}, m_7^{\tilde{E}}) = (0, 0, 0)$ | 12 | $-$ | $-$ |

Table 2: The non-planar topologies and the corresponding mass configurations, number of master integrals and occurring square roots.

$$P_7^X = -(k_2 + p_{123})^2 + (m_7^X)^2, \qquad P_8^X = -(k_1 - p_{13})^2, \qquad P_9^X = -(k_2 + p_{13})^2. \qquad (10)$$

Note that the momentum assignment differs between the planar topologies and the non-planar topologies only for $P_5^X$. The auxiliary graph for the non-planar topologies is shown in fig. 3. The non-planar topologies $\{\tilde{A}, \tilde{B}, \tilde{C}, \tilde{D}, \tilde{E}\}$ differ again by the assignment of the masses $(m_1^X, m_4^X, m_7^X)$. The cases which we consider in this paper are summarised in table 2. The first graph polynomial for the auxiliary graph shown in fig. (3) reads

$$\begin{aligned}
\mathcal{U}^X(a_1, a_2, a_3, a_4, a_5, a_6, a_7, a_8, a_9) = \\
(a_1 + a_2 + a_3 + a_6 + a_7 + a_8 + a_9)(a_4 + a_5) + (a_1 + a_2 + a_3)(a_6 + a_7 + a_9) \\
+ (a_6 + a_7 + a_9)a_8, \qquad X \in \{\tilde{A}, \tilde{B}, \tilde{C}, \tilde{D}, \tilde{E}\}.
\end{aligned} \qquad (11)$$

For each integral $I_{\nu_1 \nu_2 \nu_3 \nu_4 \nu_5 \nu_6 \nu_7 \nu_8 \nu_9}^X$ we define its sector id (or topology id) by

$$\text{id} = \sum_{j=1}^{9} 2^{j-1} \Theta(\nu_j). \qquad (12)$$

Here, $\Theta(x)$ denotes the Heaviside step function, defined by $\Theta(x) = 1$ for $x > 0$ and $\Theta(x) = 0$ otherwise.

## 2.3 The method of calculation

Integration-by-parts identities allow us to express any integral from a family of Feynman integrals as a linear combination of master integrals [80, 87]. We use the program `Kira` [88, 89] for the integration-by-parts reduction. The set of master integrals is finite [90]. Therefore, we only need to compute the master integrals. We use the method of differential equations [81–83, 91] to compute the latter: By differentiating under the integral sign and by using integration-by-parts identities we obtain a system of differential equations for the master integrals $I(x, \varepsilon)$ with respect to the kinematic variables $x$

$$dI(x, \varepsilon) = \tilde{A}(x, \varepsilon) I(x, \varepsilon). \qquad (13)$$

Here, the symbol $d$ is the differential with respect to all kinematic variables $x$, the variable $\varepsilon$ denotes the dimensional regularisation parameter, and $\tilde{A}$ is a square matrix with dimensions equal to the number of master integrals. The entries of the matrix $\tilde{A}$ are differential one-forms, rational in $x$ and $\varepsilon$. The essential step in computing Feynman integrals is to find a basis of master integrals $J(x,\varepsilon)$

$$J(x,\varepsilon) = U(x,\varepsilon)I(x,\varepsilon), \tag{14}$$

such that system of differential equations is in an $\varepsilon$-factorised form [84]

$$dJ(x,\varepsilon) = \varepsilon A(x)J(x,\varepsilon). \tag{15}$$

The essential point is that the entries of $A$ are now differential one-forms, depending on the kinematic variables, but independent of the dimensional regularisation parameter $\varepsilon$. A differential equation in $\varepsilon$-factorised form can be solved systematically order-by-order in $\varepsilon$ in terms of iterated integrals [92].

We introduce an operator $\mathbf{i}^{+}$, which raises the power of the propagator $i$ by one and multiplies by $\nu_i$, e.g.

$$\mathbf{1}^{+}I^X_{\nu_1\nu_2\nu_3\nu_4\nu_5\nu_6\nu_7\nu_8\nu_9} = \nu_1 \cdot I^X_{(\nu_1+1)\nu_2\nu_3\nu_4\nu_5\nu_6\nu_7\nu_8\nu_9}. \tag{16}$$

The notation with an extra prefactor $\nu_j$ follows ref. [93]. In addition we define the operator $\mathbf{D}^{-}$, which lowers the dimension of space-time by two units through

$$\mathbf{D}^{-}I^X_{\nu_1\nu_2\nu_3\nu_4\nu_5\nu_6\nu_7\nu_8\nu_9}(D) = I^X_{\nu_1\nu_2\nu_3\nu_4\nu_5\nu_6\nu_7\nu_8\nu_9}(D-2). \tag{17}$$

The dimensional shift relations read [94, 95]

$$\mathbf{D}^{-}I^X_{\nu_1\nu_2\nu_3\nu_4\nu_5\nu_6\nu_7\nu_8\nu_9}(D) = \mathcal{U}^X\left(\mathbf{1}^{+},\mathbf{2}^{+},\mathbf{3}^{+},\mathbf{4}^{+},\mathbf{5}^{+},\mathbf{6}^{+},\mathbf{7}^{+},\mathbf{8}^{+},\mathbf{9}^{+}\right)I^X_{\nu_1\nu_2\nu_3\nu_4\nu_5\nu_6\nu_7\nu_8\nu_9}(D), \tag{18}$$

where $\mathcal{U}^X$ denotes the first graph polynomial, defined in eq. (9) and eq. (11) for the planar and non-planar topologies, respectively.

## 2.4 Square roots

In defining master integrals of uniform transcendental weight we will encounter eight square roots. These are given by

$$
\begin{aligned}
r_1 &= \sqrt{-t\left(4m^2-t\right)}, \\
r_2 &= \sqrt{s\left(4m^2+s\right)}, \\
r_3 &= \sqrt{-t\left[4m^2s^2-t\left(m^2-s\right)^2\right]}, \\
r_4 &= \sqrt{-st\left[4m^2\left(m^2+s\right)-st\right]},
\end{aligned}
$$

$$r_5 = \sqrt{-tm^2\left[-4s\left(s+t\right)-tm^2\right]},$$

$$r_6 = \sqrt{-t\left(s+t\right)\left[4m^2\left(m^2-s-t\right)+t\left(s+t\right)\right]},$$

$$r_7 = \sqrt{-t\left[4m^2\left(s+t\right)^2-t\left(m^2+s+t\right)^2\right]},$$

$$r_8 = \sqrt{s\left[-4tm^4+s\left(s+t\right)^2\right]}. \tag{19}$$

We have chosen the arguments of the eight square roots such that in the region of interest ($s > 0$, $t < 0$, $u < 0$, $m^2 \gg s, (-t)$) the arguments of all eight roots are positive. In this region we chose the sign of the square roots such that all eight roots are positive.

## 2.5 Elliptic curves

The non-planar topologies $\tilde{A}$ and $\tilde{B}$ involve elliptic curves. In more detail, there are two elliptic curves related to topology $\tilde{A}$ and one elliptic curve related to topology $\tilde{B}$. In this section we discuss these elliptic curves. The defining equations for the elliptic curves are obtained from the maximal cut in the loop-by-loop Baikov representation [96].

In this section we denote by $(u,v)$ coordinates in a plane[1]. We encounter elliptic curves defined by a quartic polynomial $P_4(u)$:

$$E \quad : \quad v^2 - \frac{1}{s^2}P_4\left(u\right) = 0. \tag{20}$$

The factor of $1/s^2$ normalises the coefficient of $u^4$ in $P_4(u)$ and is chosen such that we have

$$E \quad : \quad v^2 - \left(u-u_1\right)\left(u-u_2\right)\left(u-u_3\right)\left(u-u_4\right) = 0, \tag{21}$$

where $u_1,\ldots,u_4$ are the roots of the polynomial $P_4(u)$. We set

$$U_1 = (u_3-u_2)(u_4-u_1), \quad U_2 = (u_2-u_1)(u_4-u_3), \quad U_3 = (u_3-u_1)(u_4-u_2), \tag{22}$$

and we define the modulus $k$ and the complementary modulus $\bar{k}$ of the elliptic curve by

$$k^2 = \frac{U_1}{U_3}, \quad \bar{k}^2 = 1 - k^2 = \frac{U_2}{U_3}. \tag{23}$$

We define the periods $\psi_1$ and $\psi_2$ of the elliptic curve as

$$\psi_1 = \frac{4\mu^2 K(k)}{U_3^{\frac{1}{2}}}, \quad \psi_2 = \frac{4i\mu^2 K(\bar{k})}{U_3^{\frac{1}{2}}}. \tag{24}$$

---

[1]The variable $u$ in this section is not related to the Mandelstam variable $u$ defined in eq. (4).

For topology $\tilde{B}$ we encounter in sector 123 the elliptic curve $E^{(a)}$ defined by the quartic polynomial

$$
\begin{aligned}
P_4^{(a)} \;=\;\; & u^2\left[s\left(u+m^2\right)+m^2\left(2s+t\right)\right]^2+m^2\left(s+t\right)\left[2s\left(2m^2-t\right)\left(u+3m^2\right)u\right. \\
& \left.+2m^2t\left(2m^2-2s-t\right)u+m^2\left(\left(4m^4-4m^2t+t^2\right)\left(s+t\right)-8m^2st\right)\right]. \tag{25}
\end{aligned}
$$

For topology $\tilde{A}$ we encounter two elliptic curves $E^{(b)}$ and $E^{(c)}$: The elliptic curve $E^{(b)}$ occurs in sector 123 and is defined by the quartic polynomial

$$
P_4^{(b)}=s^2u^2\left(u+m^2\right)^2+t\left(s+t\right)m^2\left[m^2\left(2u+2m^2-t\right)\left(2u+2m^2-s-t\right)-2s\left(u+m^2\right)^2\right]. \tag{26}
$$

The elliptic curve $E^{(c)}$ occurs in sectors 126 and 127 and is defined by the quartic polynomial

$$
P_4^{(c)} \;=\; s^2u\left(u-s\right)\left[4m^2\left(u+m^2-s\right)+u\left(u-s\right)\right]. \tag{27}
$$

The elliptic curve $E^{(c)}$ does not vary with $t$.

The roots of the polynomials $P_4^{(X)}$ (with $X\in\{a,b,c\}$) are in general algebraic functions of the kinematic variables. For concreteness, we list how we label them. Together with eq. (24) this defines the periods. The elliptic curves $E^{(a)}$ and $E^{(b)}$ are defined by irreducible polynomials of degree 4. The roots of $P_4^{(a)}$ are

$$
u_1^{(a)}=\frac{-2m^2s\left(3s+t\right)+\sqrt{D_1^{(a)}}-\sqrt{D_2^{(a)+}}}{4s^2},\quad u_2^{(a)}=\frac{-2m^2s\left(3s+t\right)-\sqrt{D_1^{(a)}}-\sqrt{D_2^{(a)-}}}{4s^2},
$$

$$
u_3^{(a)}=\frac{-2m^2s\left(3s+t\right)-\sqrt{D_1^{(a)}}+\sqrt{D_2^{(a)-}}}{4s^2},\quad u_4^{(a)}=\frac{-2m^2s\left(3s+t\right)+\sqrt{D_1^{(a)}}+\sqrt{D_2^{(a)+}}}{4s^2}, \tag{28}
$$

with

$$
\begin{aligned}
D_1^{(a)} \;=\;\; & \frac{1}{3}B^{(a)}-\frac{2}{3}s^2R^{(a)}+\frac{8m^4s^2}{3R^{(a)}}\left[16s^2t^2\left(s+t\right)^2+\left(s-t\right)^4m^4-8st\left(s^2-t^2\right)\left(2s+t\right)m^2\right], \\
D_2^{(a)\pm} \;=\;\; & B^{(a)}-D_1^{(a)}\pm\frac{64m^4s^5t\left(s+t\right)}{D_1^{(a)}}, \tag{29} \\
R^{(a)} \;=\;\; & \left(R_1^{(a)}+12\sqrt{R_2^{(a)}}\right)^{\frac{1}{3}}, \\
B^{(a)} \;=\;\; & 4m^2s^2\left(4st\left(s+t\right)+m^2\left(s-t\right)^2\right), \\
R_1^{(a)} \;=\;\; & 8m^6\left(64\left(s+t\right)^3s^3t^3+24\left(5s^2+2st+2t^2\right)\left(s+t\right)^2m^2s^2t^2\right. \\
& \left.-12\left(s+t\right)\left(2s+t\right)\left(s-t\right)^3m^4st+\left(s-t\right)^6m^6\right), \\
R_2^{(a)} \;=\;\; & 768m^{14}s^5t^4\left(s+t\right)^3\left(16\left(s+t\right)^2s^2t-\left(s-t\right)^3m^4-\left(s+t\right)\left(s^2-20st-8t^2\right)m^2s\right).
\end{aligned}
$$

The roots of $P_4^{(b)}$ are

$$u_1^{(b)} = \frac{-2m^2s^2 + \sqrt{D_1^{(b)}} - \sqrt{D_2^{(b)+}}}{4s^2}, \qquad u_2^{(b)} = \frac{-2m^2s^2 - \sqrt{D_1^{(b)}} - \sqrt{D_2^{(b)-}}}{4s^2},$$

$$u_3^{(b)} = \frac{-2m^2s^2 - \sqrt{D_1^{(b)}} + \sqrt{D_2^{(b)-}}}{4s^2}, \qquad u_4^{(b)} = \frac{-2m^2s^2 + \sqrt{D_1^{(b)}} + \sqrt{D_2^{(b)+}}}{4s^2}, \qquad (30)$$

with

$$
\begin{aligned}
D_1^{(b)} &= \frac{1}{3}B^{(b)} - \frac{2}{3}s^2 R^{(b)} + \frac{8m^4s^2}{3R^{(b)}}\left[16s^2t^2(s+t)^2 + (s+2t)^4 m^4\right.\\
&\quad \left. -8ts(s+2t)(2s+t)(s+t)m^2\right],\\
D_2^{(b)\pm} &= B^{(b)} - D_1^{(b)} \pm \frac{64m^4s^4t(s+t)(s+t-m^2)}{D_1^{(b)}}, \qquad\qquad (31)\\
R^{(b)} &= \left(R_1^{(b)} + 12\sqrt{R_2^{(b)}}\right)^{\frac{1}{3}},\\
B^{(b)} &= 4m^2s^2\left(4st(s+t) + (s^2 - 8st - 8t^2)m^2\right),\\
R_1^{(b)} &= 8m^6\left(64(s+t)^3 s^3 t^3 - 12ts(s+t)(2s+t)(s+2t)^3 m^4 + (s+2t)^6 m^6\right.\\
&\quad \left. + 24t^2s^2(5s^2 + 8st + 5t^2)(s+t)^2 m^2\right),\\
R_2^{(b)} &= 768m^{14}s^3t^4(s+t)^5\left(16(s+t)^2 s^2 t + (s+2t)^3 m^4 - (s+t)(s^2 + 22st + 13t^2)m^2 s\right).
\end{aligned}
$$

The roots of the elliptic curve $E^{(c)}$ are simpler:

$$u_1^{(c)} = \frac{1}{2}\left[s - 4m^2 + \sqrt{s^2 + 8m^2 s}\right], \quad u_2^{(c)} = 0, \quad u_3^{(c)} = s, \quad u_4^{(c)} = \frac{1}{2}\left[s - 4m^2 - \sqrt{s^2 + 8m^2 s}\right].$$

$$(32)$$

In the differential equation for the Feynman integrals it is advisable to avoid algebraic extensions as far as possible. In particular, we don't want to use the explicit expressions for the roots of the polynomial $P_4^{(X)}$. This can be done as follows: For each elliptic curve we may express the derivatives of $\psi_i^{(X)}$ (for $X \in \{a,b,c\}$ and $i \in \{1,2\}$) with respect to the kinematic variables as a linear combination of $\psi_i^{(X)}$ and $\frac{\partial}{\partial m^2}\psi_i^{(X)}$ with coefficients, which are rational functions. The relevant formulae for the elliptic curve $E^{(a)}$ are

$$
\begin{aligned}
\frac{\partial}{\partial s}\psi_i^{(a)} &= \frac{\left(9tm^2s + m^2s^2 + 4ts^2 + 2m^2t^2 + 4st^2\right)}{s(s+t)(3m^2s - 3tm^2 + 4st + 4s^2)}\psi_i^{(a)}\\
&\quad + \frac{m^2\left(4st^2 + 4ts^2 + m^2s^2 + 12tm^2s + 5m^2t^2\right)}{s(s+t)(3m^2s - 3tm^2 + 4st + 4s^2)}\frac{\partial}{\partial m^2}\psi_i^{(a)},
\end{aligned}
$$

$$\frac{\partial}{\partial t}\psi_i^{(a)} = -\frac{\left(4m^2s^2 + 9tm^2s - m^2t^2 + 4s^3 + 12ts^2 + 8st^2\right)}{t(s+t)(3m^2s - 3tm^2 + 4st + 4s^2)}\psi_i^{(a)}$$
$$-2\frac{m^2\left(2m^2s^2 + 6tm^2s + m^2t^2 + 2s^3 + 6ts^2 + 4st^2\right)}{t(s+t)(3m^2s - 3tm^2 + 4st + 4s^2)}\frac{\partial}{\partial m^2}\psi_i^{(a)},$$

$$\left(\frac{\partial}{\partial m^2}\right)^2\psi_i^{(a)} = \frac{N_1^{(a)}}{(3m^2s - 3tm^2 + 4st + 4s^2)m^4Q^{(a)}}\psi_i^{(a)}$$
$$+\frac{N_2^{(a)}}{m^2(3m^2s - 3tm^2 + 4st + 4s^2)Q^{(a)}}\frac{\partial}{\partial m^2}\psi_i^{(a)}, \tag{33}$$

with

$$
\begin{aligned}
N_1^{(a)} &= 48s^5tm^2 + 28m^4s^4t + 152m^2t^2s^4 - 28m^4s^2t^3 + 6m^4t^2s^3 + 144m^2t^3s^3 + 2m^4st^4 \\
&\quad + 44m^2s^2t^4 + 12m^6st^3 - 18m^6t^2s^2 + 12m^6ts^3 - 3m^6s^4 - 3m^6t^4 - 8m^4s^5 - 4m^2s^6 \\
&\quad + 16s^6t + 48s^5t^2 + 48s^4t^3 + 16s^3t^4, \\
N_2^{(a)} &= 36m^6ts^3 - 54m^6t^2s^2 + 48m^2s^2t^4 + 128s^6t + 384s^5t^2 + 384s^4t^3 + 128s^3t^4 + 36m^6st^3 \\
&\quad + 152m^4s^4t + 54m^4t^2s^3 - 152m^4s^2t^3 - 32m^4st^4 + 264s^5tm^2 + 612m^2t^2s^4 \\
&\quad + 384m^2t^3s^3 - 9m^6s^4 - 9m^6t^4 - 22m^4s^5 - 12m^2s^6, \\
Q^{(a)} &= m^4s^3 - 3m^4s^2t + 3m^4st^2 - m^4t^3 - 16s^2t^3 - 32s^3t^2 - 16s^4t + m^2s^4 - 19m^2s^3t \\
&\quad - 28m^2t^2s^2 - 8m^2st^3. \tag{34}
\end{aligned}
$$

For the elliptice curve $E^{(b)}$ we have

$$\frac{\partial}{\partial s}\psi_i^{(b)} = \frac{m^2(s-4t)}{(s+t)(-4st - 4s^2 + 3m^2s + 6tm^2)}\psi_i^{(b)}$$
$$+\frac{m^2\left(4st^2 + 4ts^2 + m^2s^2 - 10tm^2s - 6m^2t^2\right)}{s(s+t)(-4st - 4s^2 + 3m^2s + 6tm^2)}\frac{\partial}{\partial m^2}\psi_i^{(b)},$$

$$\frac{\partial}{\partial t}\psi_i^{(b)} = -\frac{\left(4m^2s^2 + 5tm^2s + 6m^2t^2 - 4s^3 - 8ts^2 - 4st^2\right)}{t(s+t)(-4st - 4s^2 + 3m^2s + 6tm^2)}\psi_i^{(b)}$$
$$-\frac{m^2s\left(4m^2s - tm^2 - 4st - 4s^2\right)}{t(s+t)(-4st - 4s^2 + 3m^2s + 6tm^2)}\frac{\partial}{\partial m^2}\psi_i^{(b)},$$

$$\left(\frac{\partial}{\partial m^2}\right)^2\psi_i^{(b)} = \frac{N_1^{(b)}}{m^4(-4st - 4s^2 + 3m^2s + 6tm^2)Q^{(b)}}\psi_i^{(b)}$$
$$+\frac{N_2^{(b)}}{m^2(-4st - 4s^2 + 3m^2s + 6tm^2)Q^{(b)}}\frac{\partial}{\partial m^2}\psi_i^{(b)}, \tag{35}$$

with

$$
\begin{aligned}
N_1^{(b)} &= -3m^6s^4 - 4m^2s^6 + 8m^4s^5 - 72m^6t^2s^2 - 24m^6ts^3 - 16m^2s^2t^4 + 76m^4st^4 - 96m^6st^3 \\
&\quad + 48s^5t^2 + 48s^4t^3 + 16s^3t^4 + 16s^6t - 48m^6t^4 + 68m^4s^4t - 148m^2t^2s^4 + 186m^4t^2s^3
\end{aligned}
$$

$$-72\,s^5tm^2+202\,m^4s^2t^3-96\,m^2t^3s^3,$$

$$N_2^{(b)} = -216\,m^6t^2s^2-72\,m^6ts^3+128\,s^3t^4+384\,s^4t^3+384\,s^5t^2+128\,s^6t-252\,m^2s^2t^4$$
$$-816\,m^2t^3s^3-888\,m^2t^2s^4-336\,s^5tm^2+284\,m^4st^4+818\,m^4s^2t^3+774\,m^4t^2s^3$$
$$+262\,m^4s^4t-288\,m^6st^3-12\,m^2s^6+22\,m^4s^5-144\,m^6t^4-9\,m^6s^4,$$

$$Q^{(b)} = 16\,s^2t^3+32\,s^3t^2+16\,s^4t+m^4s^3+6\,m^4s^2t+12\,m^4st^2+8\,m^4t^3-m^2s^4$$
$$-23\,m^2s^3t-35\,m^2t^2s^2-13\,m^2st^3. \tag{36}$$

Finally, for the elliptic curve $E^{(c)}$ we have

$$\frac{\partial}{\partial s}\psi_i^{(c)} = -\frac{1}{s}\psi_i^{(c)}-\frac{m^2}{s}\frac{\partial}{\partial m^2}\psi_i^{(c)}, \tag{37}$$

$$\frac{\partial}{\partial t}\psi_i^{(c)} = 0,$$

$$\left(\frac{\partial}{\partial m^2}\right)^2\psi_i^{(c)} = -2\frac{(-s+4m^2)}{m^2\,(8\,m^2+s)\,(m^2-s)}\psi_i^{(c)}-\frac{(-14\,m^2s+24\,m^4-s^2)}{m^2\,(8\,m^2+s)\,(m^2-s)}\frac{\partial}{\partial m^2}\psi_i^{(c)}.$$

The Wronskian $W_{m^2}^{(X)}$ (for $X\in\{a,b,c\}$) is defined by

$$W_{m^2}^{(X)} = \mu^2\left(\psi_1^{(X)}\frac{\partial}{\partial m^2}\psi_2^{(X)}-\psi_2^{(X)}\frac{\partial}{\partial m^2}\psi_1^{(X)}\right). \tag{38}$$

The Wronskian is also a rational function in the kinematic variables. Explicitly we have

$$W_{m^2}^{(a)}=-\frac{2\pi i\mu^6s^2\left[3m^2(s-t)+4s(s+t)\right]}{m^4\left[m^4(s-t)^3-16s^2t(s+t)^2+m^2s(s+t)\left(s^2-20st-8t^2\right)\right]}.$$

$$W_{m^2}^{(b)}=\frac{2\pi i\mu^6s^2\left[4s(s+t)-3m^2(s+2t)\right]}{m^4\left[16s^2t(s+t)^2+m^4(s+2t)^3-m^2s(s+t)(s^2+22st+13t^2)\right]}.$$

$$W_{m^2}^{(c)}=-\frac{12\pi i\mu^6}{m^2(m^2-s)(8m^2+s)}.$$

All three elliptic curves degenerate for generic values of $s$ and $t$ in the limit $m^2\to\infty$ to a genus zero curve. More concretely, the curve $E^{(a)}$ degenerates for $s-t\neq 0$ in the limit $m^2\to\infty$ to a genus zero curve, the curve $E^{(b)}$ degenerates for $s+2t\neq 0$ in the limit $m^2\to\infty$ to a genus zero curve. There are no restrictions for curve $E^{(c)}$.

The expansion of $\psi_1^{(X)}$ for $X\in\{a,b,c\}$ around $m^2=\infty$ reads

$$\psi_1^{(a)} = 2\pi i\mu^2\left[-\frac{s}{(s-t)}\frac{1}{m^2}-2\frac{s^2t\,(s+t)\,(2s+t)}{(s-t)^4}\frac{1}{m^4}-6\frac{s^3t^2\,(s+t)^2\,(6s^2+8st+t^2)}{(s-t)^7}\frac{1}{m^6}\right]$$

$$+O\left(m^{-8}\right),$$

$$\psi_1^{(b)} = 2\pi i\mu^2 \left[\frac{s}{(s+2t)}\frac{1}{m^2} + 2\frac{s^2t\,(s+t)\,(2s+t)}{(s+2t)^4}\frac{1}{m^4} + 6\frac{s^3t^2\,(s+t)^2\,(6s^2+4st-t^2)}{(s+2t)^7}\frac{1}{m^6}\right]$$
$$+O\left(m^{-8}\right),$$

$$\psi_1^{(c)} = 2\pi i\mu^2 \left[-\frac{1}{2m^2} - \frac{s}{8m^4} - \frac{5s^2}{64m^6}\right] + O\left(m^{-8}\right). \tag{39}$$

For these expansions we have to choose the signs of several square roots. We adopted the convention, that all square roots are continuous under Feynman's $i\delta$-prescription. For the kinematic region of interest the periods are continuous under the substitution $s \to s + i\delta$ with $\delta \geq 0$.

# 3 Master integrals

In this section we present the master integrals, which lead to an $\varepsilon$-factorised differential equation. These master integrals are constructed by analysing the maximal cut in the loop-by-loop Baikov representation [96]. We list the master integrals for the planar topologies $A$, $B$, $C$, $D$ and $E$ in section 3.1. The master integrals for the non-planar topologies $\tilde{A}$, $\tilde{B}$, $\tilde{C}$, $\tilde{D}$ and $\tilde{E}$ are listed in section 3.2.

## 3.1 The planar topologies

### 3.1.1 Topology $A$

Sector 72: $\quad J_1^A = \varepsilon^2 I_{000200200}^A,$

Sector 14: $\quad J_2^A = \varepsilon^2 \left(\frac{-s}{\mu^2}\right) \mathbf{D}^- I_{011100000}^A,$

Sector 25: $\quad J_3^A = \varepsilon^2 \left(\frac{m^2}{\mu^2}\right) \mathbf{D}^- I_{100110000}^A,$

Sector 28: $\quad J_4^A = \varepsilon^2 \left(\frac{m^2-s}{\mu^2}\right) \mathbf{D}^- I_{001110000}^A,$

$\quad\quad\quad\quad\quad J_5^A = \varepsilon^2 \left[\left(\frac{m^2}{\mu^2}\right) \mathbf{D}^- I_{001110000}^A + \mathbf{D}^- I_{00111(-1)000}^A\right],$

Sector 73: $\quad J_6^A = \varepsilon^2 \left[\left(\frac{r_1}{\mu^2}\right) \mathbf{D}^- I_{100100100}^A + \left(\frac{-t}{\mu^2}\right) I_{100200200}^A\right],$

$\quad\quad\quad\quad\quad J_7^A = \varepsilon^2 \left(\frac{-t}{\mu^2}\right) I_{100200200}^A,$

Sector 54: $\quad J_8^A = \varepsilon^2 \left(\frac{s^2}{\mu^4}\right) \mathbf{D}^- I_{011011000}^A,$

Sector 57: $\quad J_9^A = \varepsilon^3 \left( \dfrac{-s}{\mu^2} \right) I_{100211000}^A$,

$\qquad\qquad J_{10}^A = \varepsilon^3 \left( \dfrac{-s}{\mu^2} \right) I_{200111000}^A$,

Sector 78: $\quad J_{11}^A = \varepsilon^2 \left( \dfrac{r_2}{\mu^2} \right) \left[ \left( \dfrac{-s}{\mu^2} \right) I_{021100200}^A + \dfrac{\varepsilon}{2(1+2\varepsilon)} \left( \dfrac{\mu^2}{m^2} \right) I_{000200200}^A \right]$,

$\qquad\qquad J_{12}^A = \varepsilon^3 \left( \dfrac{-s}{\mu^2} \right) I_{011100200}^A$,

Sector 89: $\quad J_{13}^A = \varepsilon^3 \left( \dfrac{-t}{\mu^2} \right) I_{100210100}^A$,

Sector 92: $\quad J_{14}^A = \varepsilon^2 \left( \dfrac{-s}{\mu^2} \right) \left[ \left( \dfrac{m^2}{\mu^2} \right) \mathbf{D}^- I_{001110100}^A - I_{002120000}^A + \dfrac{1}{2} I_{002210000}^A \right]$,

$\qquad\qquad J_{15}^A = \varepsilon^3 \left( \dfrac{-s}{\mu^2} \right) I_{001210100}^A$,

Sector 120: $\quad J_{16}^A = \varepsilon^3 \left( \dfrac{-s}{\mu^2} \right) I_{000211100}^A$,

Sector 59: $\quad J_{17}^A = \varepsilon^4 \left( \dfrac{-s}{\mu^2} \right) I_{110111000}^A$,

Sector 62: $\quad J_{18}^A = \varepsilon^3 (-1 + 2\varepsilon) \left( \dfrac{-s}{\mu^2} \right) I_{011111000}^A$,

Sector 79: $\quad J_{19}^A = \varepsilon^3 \left( \dfrac{-s}{\mu^2} \right) \left[ I_{011100200}^A - I_{101100200}^A - I_{1111002(-1)0}^A \right]$,

$\qquad\qquad J_{20}^A = \varepsilon^3 \left( \dfrac{-s}{\mu^2} \right) \left( \dfrac{r_1}{\mu^2} \right) I_{111100200}^A$,

Sector 91: $\quad J_{21}^A = \varepsilon^4 \left( \dfrac{-t}{\mu^2} \right) I_{110110100}^A$,

Sector 93: $\quad J_{22}^A = \varepsilon^4 \left( \dfrac{-s-t}{\mu^2} \right) I_{101110100}^A$,

$\qquad\qquad J_{23}^A = \varepsilon^3 \left( \dfrac{r_3}{\mu^4} \right) I_{101210100}^A$,

$\qquad\qquad J_{24}^A = \varepsilon^3 \left[ \left( \dfrac{-s-t}{\mu^2} \right) \left( \dfrac{m^2}{\mu^2} \right) I_{101110200}^A - \left( \dfrac{s(2m^2 - t) + tm^2}{2\mu^4} \right) I_{101210100}^A \right]$,

Sector 94: $\quad J_{25}^A = \varepsilon^4 \left( \dfrac{-s}{\mu^2} \right) I_{011110100}^A$,

Sector 118: $\quad J_{26}^A = \varepsilon^3 \left( \dfrac{s^2}{\mu^4} \right) I_{021011100}^A$,

Sector 121: $\quad J_{27}^A = \varepsilon^3 \left( \dfrac{-s}{\mu^2} \right) \left( \dfrac{m^2 - t}{\mu^2} \right) I_{100211100}^A$,

$$J_{28}^A = \varepsilon^3 \left(\frac{r_4}{\mu^4}\right) \left[I_{200111100}^A + I_{100211100}^A\right],$$

$$J_{29}^A = \varepsilon^3 \left(\frac{-s}{\mu^2}\right) \left[\left(\frac{2m^2 - t}{\mu^2}\right) \left[I_{200111100}^A + I_{100211100}^A\right]\right.$$

$$\left. + \left[I_{20011110(-1)}^A + I_{10021110(-1)}^A\right]\right] - \left(\frac{s}{t}\right) J_7^A,$$

Sector 95:  $$J_{30}^A = \varepsilon^4 \left(\frac{-s}{\mu^2}\right) \left(\frac{-t}{\mu^2}\right) I_{111110100}^A,$$

Sector 126:  $$J_{31}^A = \varepsilon^4 \left(\frac{s^2}{\mu^4}\right) I_{011111100}^A,$$

Sector 127:  $$J_{32}^A = \varepsilon^4 \left(\frac{-s}{\mu^2}\right) \left(\frac{r_4}{\mu^4}\right) I_{111111100}^A,$$

$$J_{33}^A = \varepsilon^4 \left(\frac{s^2}{\mu^4}\right) \left[\left(\frac{2m^2 - t}{\mu^2}\right) I_{111111100}^A + I_{11111110(-1)}^A\right] + \left(\frac{s}{t}\right) J_{30}^A,$$

$$J_{34}^A = \varepsilon^4 \left(\frac{s^2}{\mu^4}\right) I_{1111111(-1)0}^A - \left(\frac{2m^2 + s}{2(m^2 - t)}\right) \left[J_3^A + \frac{3}{2} J_4^A - \frac{5}{2} J_5^A + 3 J_7^A \right.$$

$$\left. + 2 J_9^A + 4 J_{13}^A - \frac{2}{3} J_{14}^A + \frac{2}{3} J_{15}^A + 2 J_{17}^A + 2 J_{21}^A - 6 J_{22}^A + 4 J_{24}^A - 2 J_{27}^A\right],$$

$$J_{35}^A = \varepsilon^4 \left(\frac{-s}{\mu^2}\right) \left[I_{1111111(-1)(-1)}^A + \left(\frac{-t}{\mu^2}\right) \left(\left(\frac{2m^2 - t}{\mu^2}\right) I_{111111100}^A\right.\right.$$

$$\left.\left. + I_{11111110(-1)}^A\right) + \left(\frac{m^2 - t}{\mu^2}\right) I_{1111111(-1)0}^A\right] - \left(\frac{t + (4s + 2t)\varepsilon}{s(-1 + 2\varepsilon)}\right) J_{18}^A$$

$$+ \left(\frac{t}{s}\right) \left[J_{10}^A + 2 J_{25}^A + J_{26}^A\right] - \left(\frac{s}{t}\right) J_{21}^A - J_{30}^A$$

$$- \left(\frac{2tm^2 + s(2m^2 + t)}{s^2}\right) J_{31}^A.$$

### 3.1.2  Topology $B$

Sector 65:  $$J_1^B = \varepsilon^2 \mathbf{D}^- I_{100000100}^B,$$

Sector 25:  $$J_2^B = \varepsilon^2 \left(\frac{m^2}{\mu^2}\right) \mathbf{D}^- I_{100110000}^B,$$

Sector 28:  $$J_3^B = \varepsilon^2 \left(\frac{-s}{\mu^2}\right) \mathbf{D}^- I_{001110000}^B,$$

Sector 49:  $$J_4^B = \varepsilon^2 \left(\frac{-s}{\mu^2}\right) \mathbf{D}^- I_{100011000}^B,$$

Sector 73: $\quad J_5^B = \varepsilon^2 \left( \dfrac{r_1}{\mu^2} \right) \mathbf{D}^- I_{100100100}^B,$

$\qquad\qquad J_6^B = \varepsilon^2 \left( \dfrac{-t}{\mu^2} \right) I_{200100200}^B,$

Sector 29: $\quad J_7^B = \varepsilon^3 \left( \dfrac{-s}{\mu^2} \right) I_{101210000}^B,$

Sector 54: $\quad J_8^B = \varepsilon^2 \left( \dfrac{s^2}{\mu^4} \right) \mathbf{D}^- I_{011011000}^B,$

Sector 57: $\quad J_9^B = \varepsilon^3 \left( \dfrac{-s}{\mu^2} \right) I_{100211000}^B,$

$\qquad\qquad J_{10}^B = \varepsilon^3 \left( \dfrac{-s}{\mu^2} \right) I_{200111000}^B,$

Sector 71: $\quad J_{11}^B = \varepsilon^3 \left( \dfrac{-s}{\mu^2} \right) I_{111000200}^B,$

Sector 75: $\quad J_{12}^B = \varepsilon^3 \left( \dfrac{-t}{\mu^2} \right) I_{110100200}^B,$

Sector 55: $\quad J_{13}^B = \varepsilon^3 \left( \dfrac{s^2}{\mu^2} \right) I_{111021000}^B,$

Sector 59: $\quad J_{14}^B = \varepsilon^4 \left( \dfrac{-s}{\mu^2} \right) I_{110111000}^B,$

Sector 79: $\quad J_{15}^B = \varepsilon^3 \left( \dfrac{-s}{\mu^2} \right) \left( \dfrac{m^2 - t}{\mu^2} \right) I_{111100200}^B,$

$\qquad\qquad J_{16}^B = \varepsilon^3 \left( \dfrac{-s}{\mu^2} \right) \left[ \left( \dfrac{m^2 - t}{\mu^2} \right) I_{111100200}^B + I_{1111002(-1)0}^B + I_{110100200}^B \right],$

$\qquad\qquad J_{17}^B = \varepsilon^3 \left( \dfrac{r_4}{\mu^4} \right) \left[ I_{111100200}^B + I_{111200100}^B \right]$

Sector 91: $\quad J_{18}^B = \varepsilon^4 \left( \dfrac{-t}{\mu^2} \right) I_{110110100}^B,$

Sector 93: $\quad J_{19}^B = \varepsilon^4 \left( \dfrac{-s - t}{\mu^2} \right) I_{101110100}^B,$

$\qquad\qquad J_{20}^B = \varepsilon^3 \left( \dfrac{r_4}{\mu^4} \right) I_{101210100}^B,$

$\qquad\qquad J_{21}^B = \varepsilon^3 \left( \dfrac{-s - t}{\mu^2} \right) I_{1(-1)1210100}^B + \varepsilon^2 \left( \dfrac{-s}{\mu^2} \right) I_{200100200}^B,$

$\qquad\qquad J_{22}^B = \varepsilon^3 \left( \dfrac{-s}{\mu^2} \right) \left[ \left( \dfrac{2m^2 - t}{\mu^2} \right) I_{101210100}^B + I_{1012101(-1)0}^B \right]$

$\qquad\qquad\qquad - \varepsilon^2 \left( \dfrac{-s}{\mu^2} \right) I_{200100200}^B,$

Sector 119: $J_{23}^B = \varepsilon^4 \left( \dfrac{s^2}{\mu^4} \right) I_{111011100}^B,$

Sector 127: $J_{24}^B = \varepsilon^4 \left( \dfrac{-s}{\mu^2} \right) \left( \dfrac{r_4}{\mu^4} \right) I_{111111100}^B,$

$$J_{25}^B = \varepsilon^4 \left( \dfrac{s^2}{\mu^4} \right) \left[ \left( \dfrac{2m^2 - t}{\mu^2} \right) I_{111111100}^B + I_{11111110(-1)}^B \right]$$
$$- \left( \dfrac{2m^2 + s}{m^2 - t} \right) \left[ 2J_6^B - 2J_7^B + J_{10}^B + 2J_{12}^B + J_{14}^B - J_{15}^B + J_{18}^B - J_{19}^B \right.$$
$$\left. + 2J_{21}^B + J_{22}^B \right],$$

$$J_{26}^B = \varepsilon^4 \left( \dfrac{-s}{\mu^2} \right) \left[ I_{1111111(-1)(-1)}^B + 2 \left( \dfrac{m^2 - t}{\mu^2} \right) I_{11111110(-1)}^B \right.$$
$$\left. + \left( \dfrac{-t}{\mu^2} \right) \left( \dfrac{2m^2 - t}{\mu^2} \right) I_{111111100}^B \right] + \left( \dfrac{2s + 4t\varepsilon}{3s(1 - 2\varepsilon)} \right) J_3^B$$
$$- \left( \dfrac{2m^2(s + t) + st}{2s^2} \right) \left[ 8J_7^B - 4J_9^B - 4J_{13}^B \right]$$
$$- \left( \dfrac{(s + t)\varepsilon}{s(1 - 2\varepsilon)} \right) J_8^B - \left( \dfrac{8m^2(s + t)}{s^2} \right) J_{14}^B - \left( \dfrac{s}{t} \right) J_{18}^B - \left( \dfrac{t}{s} \right) J_{23}^B.$$

### 3.1.3 Topology $C$

Sector 14: $\quad J_1^C = \varepsilon^2 \left( \dfrac{-s}{\mu^2} \right) \mathbf{D}^- I_{011100000}^C$

Sector 25: $\quad J_2^C = \varepsilon^2 \left( \dfrac{m^2}{\mu^2} \right) \mathbf{D}^- I_{100110000}^C$

Sector 28: $\quad J_3^C = \varepsilon^2 \left( \dfrac{m^2 - s}{\mu^2} \right) \mathbf{D}^- I_{001110000}^C$

$\quad J_4^C = \varepsilon^2 \left[ \left( \dfrac{m^2}{\mu^2} \right) \mathbf{D}^- I_{001110000}^C + \mathbf{D}^- I_{00111(-1)000}^C \right]$

Sector 73: $\quad J_5^C = \varepsilon^2 \left( \dfrac{m^2 - t}{\mu^2} \right) \mathbf{D}^- I_{100100100}^C$

$\quad J_6^C = \varepsilon^2 \left( \dfrac{-t}{\mu^2} \right) I_{100200200}^C$

Sector 54: $\quad J_7^C = \varepsilon^2 \left( \dfrac{s^2}{\mu^4} \right) \mathbf{D}^- I_{011011000}^C$

Sector 57: $\quad J_8^C = \varepsilon^3 \left( \dfrac{-s}{\mu^2} \right) I_{200111000}^C$

$\quad J_9^C = \varepsilon^3 \left( \dfrac{-s}{\mu^2} \right) I_{100211000}^C$

Sector 59: $\quad J_{10}^C = \varepsilon^4 \left( \dfrac{-s}{\mu^2} \right) I_{110111000}^C$

Sector 62: $\quad J_{11}^C = \varepsilon^3 (-1 + 2\varepsilon) \left( \dfrac{-s}{\mu^2} \right) I_{011111000}^C$

Sector 79: $\quad J_{12}^C = \varepsilon^3 \left( \dfrac{-s}{\mu^2} \right) \left( \dfrac{-t}{\mu^2} \right) I_{111200100}^C$

$\qquad\quad J_{13}^C = \varepsilon^3 \left( \dfrac{-s}{\mu^2} \right) I_{1112001(-1)0}^C - \dfrac{s}{2t} \left( J_2^C - J_5^C + 4 J_6^C \right)$

Sector 91: $\quad J_{14}^C = \varepsilon^4 \left( \dfrac{-t}{\mu^2} \right) I_{110110100}^C$

Sector 93: $\quad J_{15}^C = \varepsilon^4 \left( \dfrac{-s-t}{\mu^2} \right) I_{101110100}^C$

$\qquad\quad J_{16}^C = \varepsilon^3 \left( \dfrac{-m^2(s+t) + st}{\mu^4} \right) I_{101210100}^C$

Sector 127: $\quad J_{17}^C = \varepsilon^4 \left( \dfrac{s^2}{\mu^4} \right) \left( \dfrac{m^2 - t}{\mu^2} \right) I_{111111100}^C$

$\qquad\quad J_{18}^C = \varepsilon^4 \left( \dfrac{s^2}{\mu^4} \right) I_{1111111(-1)0}^C$
$\qquad\qquad\quad - \left( \dfrac{s}{2t} \right) \Big[ J_2^C - J_3^C + J_4^C - J_5^C + 2J_6^C - 2J_9^C - 2J_{10}^C + 2J_{12}^C$
$\qquad\qquad\qquad - 2J_{14}^C + 6J_{15}^C - 2J_{16}^C \Big]$

$\qquad\quad J_{19}^C = \varepsilon^4 \left( \dfrac{-s}{\mu^2} \right) \Big[ I_{1111111(-1)(-1)}^C + \left( \dfrac{-t}{\mu^2} \right) \left( \dfrac{m^2 - t}{\mu^2} \right) I_{111111100}^C$
$\qquad\qquad\qquad + 2 \left( \dfrac{-t}{\mu^2} \right) I_{1111111(-1)0}^C \Big]$
$\qquad\qquad\quad - \left( \dfrac{t}{4s} \right) \left( J_7^C - 8 J_8^C \right) - \dfrac{2\frac{t}{s} + 4\varepsilon}{-1 + 2\varepsilon} J_{11}^C - \left( \dfrac{s}{t} \right) J_{14}^C$

### 3.1.4 Topology $D$

Sector 25: $\quad J_1^D = \varepsilon^2 \left( \dfrac{m^2}{\mu^2} \right) \mathbf{D}^- I_{100110000}^D,$

Sector 28: $\quad J_2^D = \varepsilon^2 \left( \dfrac{-s}{\mu^2} \right) \mathbf{D}^- I_{001110000}^D,$

Sector 49: $\quad J_3^D = \varepsilon^2 \left( \dfrac{-s}{\mu^2} \right) \mathbf{D}^- I_{100011000}^D,$

Sector 73: $\quad J_4^D = \varepsilon^2 \left( \dfrac{m^2 - t}{\mu^2} \right) \mathbf{D}^- I_{100100100}^D,$

$$J_5^D = \varepsilon^2 \left( \frac{-t}{\mu^2} \right) I_{200100200}^D,$$

Sector 29: $\quad J_6^D = \varepsilon^3 \left( \frac{-s}{\mu^2} \right) I_{101210000}^D,$

Sector 54: $\quad J_7^D = \varepsilon^2 \left( \frac{s^2}{\mu^4} \right) \mathbf{D}^- I_{011011000}^D,$

Sector 57: $\quad J_8^D = \varepsilon^3 \left( \frac{-s}{\mu^2} \right) I_{100211000}^D,$

$$J_9^D = \varepsilon^3 \left( \frac{-s}{\mu^2} \right) I_{200111000}^D,$$

Sector 78: $\quad J_{10}^D = \varepsilon^3 \left( \frac{-s}{\mu^2} \right) I_{011200100}^D,$

Sector 55: $\quad J_{11}^D = \varepsilon^3 \left( \frac{s^2}{\mu^4} \right) I_{111021000}^D,$

Sector 59: $\quad J_{12}^D = \varepsilon^4 \left( \frac{-s}{\mu^2} \right) I_{110111000}^D,$

Sector 79: $\quad J_{13}^D = \varepsilon^3 \left( \frac{-s}{\mu^2} \right) \left( \frac{m^2 - t}{\mu^2} \right) I_{111200100}^D - J_5^D,$

$$J_{14}^D = \varepsilon^3 \left( \frac{-s}{\mu^2} \right) \left[ \left( \frac{m^2 - t}{\mu^2} \right) I_{111200100}^D - I_{1112001(-1)0}^D \right] + \frac{s}{2t} J_5^D,$$

Sector 93: $\quad J_{15}^D = \varepsilon^4 \left( \frac{-s-t}{\mu^2} \right) I_{101110100}^D,$

$$J_{16}^D = \varepsilon^3 \left( \frac{-s}{\mu^2} \right) \left( \frac{m^2 - t}{\mu^2} \right) I_{101210100}^D,$$

Sector 121: $\quad J_{17}^D = \varepsilon^3 \left( \frac{-s}{\mu^2} \right) \left[ \left( \frac{-t}{\mu^2} \right) I_{200111100}^D + I_{20011110(-1)}^D \right]$

$$- \left( \frac{s}{2t} \right) \left[ J_1^D - J_4^D + 4J_5^D \right],$$

$$J_{18}^D = \varepsilon^3 \left( \frac{-s}{\mu^2} \right) I_{20011110(-1)}^D - \left( \frac{s}{2t} \right) \left[ J_1^D - J_4^D + 4J_5^D \right],$$

Sector 127: $\quad J_{19}^D = \varepsilon^4 \left( \frac{s^2}{\mu^4} \right) \left[ \left( \frac{m^2 - t}{\mu^2} \right) I_{111111100}^D + I_{11111110(-1)}^D \right]$

$$+ \frac{1}{2} \left( \frac{2m^2 + s}{m^2 - t} \right) \left[ 2J_5^D + 2J_{13}^D - J_{16}^D \right],$$

$$J_{20}^D = \varepsilon^4 \left( \frac{s^2}{\mu^4} \right) I_{11111110(-1)}^D + \frac{1}{2} \left( \frac{2m^2 + s}{m^2 - t} \right) \left[ 2J_5^D + 2J_{13}^D - J_{16}^D \right],$$

$$J_{21}^D = \varepsilon^4 \left( \frac{s^2}{\mu^4} \right) \left[ \left( \frac{m^2 - t}{\mu^2} \right) I_{111111100}^D + I_{1111111(-1)0}^D \right]$$

$$- \left( \frac{s}{24t} \right) \left[ 9J_1^D - J_2^D - 9J_4^D + 36J_5^D - 12J_6^D - 24J_9^D - 24J_{12}^D \right.$$
$$\left. - 36J_{15}^D - 6J_{16}^D + 24J_{17}^D - 24J_{18}^D \right].$$

### 3.1.5 Topology $E$

$$\text{Sector 28:} \quad J_1^E = \varepsilon^2 \left( \frac{-s}{\mu^2} \right) \mathbf{D}^- I_{001110000}^E,$$

$$\text{Sector 73:} \quad J_2^E = \varepsilon^2 \left( \frac{-t}{\mu^2} \right) \mathbf{D}^- I_{100100100}^E,$$

$$\text{Sector 54:} \quad J_3^E = \varepsilon^2 \left( \frac{s^2}{\mu^4} \right) \mathbf{D}^- I_{011011000}^E,$$

$$\text{Sector 57:} \quad J_4^E = \varepsilon^3 \left( \frac{-s}{\mu^2} \right) I_{200111000}^E,$$

$$\text{Sector 79:} \quad J_5^E = \varepsilon^3 \left( \frac{-s}{\mu^2} \right) \left( \frac{-t}{\mu^2} \right) I_{111100200}^E,$$

$$\text{Sector 93:} \quad J_6^E = \varepsilon^4 \left( \frac{-s-t}{\mu^2} \right) I_{101110100}^E,$$

$$\text{Sector 127:} \quad J_7^E = \varepsilon^4 \left( \frac{s^2}{\mu^4} \right) \left( \frac{-t}{\mu^2} \right) I_{111111100}^E,$$

$$J_8^E = \varepsilon^4 \left( \frac{s^2}{\mu^4} \right) I_{1111111(-1)0}^E - \frac{s}{4t} \left[ J_1^E + J_2^E + 4J_5^E - 12J_6^E \right].$$

## 3.2 The non-planar topologies

### 3.2.1 Topology $\tilde{A}$

$$\text{Sector 72:} \quad J_1^{\tilde{A}} = \varepsilon^2 \mathbf{D}^- I_{000100100}^{\tilde{A}},$$

$$\text{Sector 14:} \quad J_2^{\tilde{A}} = \varepsilon^2 \left( \frac{-s}{\mu^2} \right) \mathbf{D}^- I_{011100000}^{\tilde{A}},$$

$$\text{Sector 41:} \quad J_3^{\tilde{A}} = \varepsilon^2 \left( \frac{m^2}{\mu^2} \right) \mathbf{D}^- I_{100101000}^{\tilde{A}},$$

$$\text{Sector 42:} \quad J_4^{\tilde{A}} = \varepsilon^2 \left( \frac{-s}{\mu^2} \right) I_{020201000}^{\tilde{A}},$$

$$J_5^{\tilde{A}} = \varepsilon^2 \left( \frac{m^2-s}{\mu^2} \right) \mathbf{D}^- I_{010101000}^{\tilde{A}},$$

$$\text{Sector 49:} \quad J_6^{\tilde{A}} = \varepsilon^2 \left( \frac{-s-t}{\mu^2} \right) \mathbf{D}^- I_{100011000}^{\tilde{A}},$$

Sector 73: $\quad J_7^{\tilde{A}} = \varepsilon^2 \left( \dfrac{r_1}{\mu^2} \right) \mathbf{D}^- I_{100100100}^{\tilde{A}},$

$\qquad\qquad J_8^{\tilde{A}} = \varepsilon^2 \left( \dfrac{-t}{\mu^2} \right) I_{100200200}^{\tilde{A}},$

Sector 54: $\quad J_9^{\tilde{A}} = \varepsilon^3 \left( \dfrac{-s}{\mu^2} \right) I_{011021000}^{\tilde{A}},$

Sector 57: $\quad J_{10}^{\tilde{A}} = \varepsilon^3 \left( \dfrac{-s-t}{\mu^2} \right) I_{100112000}^{\tilde{A}},$

Sector 78: $\quad J_{11}^{\tilde{A}} = \varepsilon^2 \left( \dfrac{r_2}{\mu^2} \right) \left[ \left( \dfrac{-s}{\mu^2} \right) I_{021100200}^{\tilde{A}} + \left( \dfrac{\mu^2}{m^2} \right) \dfrac{\varepsilon}{2(1+2\varepsilon)} I_{000200200}^{\tilde{A}} \right],$

$\qquad\qquad J_{12}^{\tilde{A}} = \varepsilon^3 \left( \dfrac{-s}{\mu^2} \right) I_{011100200}^{\tilde{A}},$

Sector 89: $\quad J_{13}^{\tilde{A}} = \varepsilon^3 \left( \dfrac{-t}{\mu^2} \right) I_{100110200}^{\tilde{A}},$

Sector 92: $\quad J_{14}^{\tilde{A}} = \varepsilon^3 \left( \dfrac{-s}{\mu^2} \right) I_{001110200}^{\tilde{A}},$

$\qquad\qquad J_{15}^{\tilde{A}} = \varepsilon^3 \left( \dfrac{-s}{\mu^2} \right) I_{002110100}^{\tilde{A}},$

Sector 55: $\quad J_{16}^{\tilde{A}} = \varepsilon^3 \left( \dfrac{-s}{\mu^2} \right) \left( \dfrac{-s-t}{\mu^2} \right) I_{111021000}^{\tilde{A}},$

Sector 59: $\quad J_{17}^{\tilde{A}} = \varepsilon^4 \left( \dfrac{-t}{\mu^2} \right) I_{110111000}^{\tilde{A}},$

$\qquad\qquad J_{18}^{\tilde{A}} = \varepsilon^3 \left( \dfrac{-t}{\mu^2} \right) \left( \dfrac{m^2}{\mu^2} \right) I_{110211000}^{\tilde{A}},$

Sector 61: $\quad J_{19}^{\tilde{A}} = \varepsilon^4 \left( \dfrac{-s-t}{\mu^2} \right) I_{101111000}^{\tilde{A}},$

Sector 62: $\quad J_{20}^{\tilde{A}} = \varepsilon^4 \left( \dfrac{-s}{\mu^2} \right) I_{011111000}^{\tilde{A}},$

$\qquad\qquad J_{21}^{\tilde{A}} = \varepsilon^3 \left( \dfrac{-s}{\mu^2} \right) \left( \dfrac{m^2}{\mu^2} \right) I_{011211000}^{\tilde{A}},$

Sector 79: $\quad J_{22}^{\tilde{A}} = \varepsilon^3 \left( \dfrac{-s}{\mu^2} \right) \left( \dfrac{r_1}{\mu^2} \right) I_{111100200}^{\tilde{A}},$

$\qquad\qquad J_{23}^{\tilde{A}} = \varepsilon^3 \left( \dfrac{-s}{\mu^2} \right) I_{1111002(-1)0}^{\tilde{A}} - \left( \dfrac{s}{2t} \right) J_8^{\tilde{A}},$

Sector 91: $\quad J_{24}^{\tilde{A}} = \varepsilon^4 \left( \dfrac{-t}{\mu^2} \right) I_{110110100}^{\tilde{A}},$

Sector 93: $\quad J_{25}^{\tilde{A}} = \varepsilon^4 \left( \dfrac{-s-t}{\mu^2} \right) I_{101110100}^{\tilde{A}},$

$$J_{26}^{\tilde{A}} = \varepsilon^3 \left( \frac{r_3}{\mu^2} \right) I_{101110200}^{\tilde{A}},$$

$$J_{27}^{\tilde{A}} = \varepsilon^3 \left[ \left( \frac{-s-t}{\mu^2} \right) \left( \frac{m^2}{\mu^2} \right) I_{101210100}^{\tilde{A}} + \frac{1}{2} \left( \frac{st - m^2(2s+t)}{\mu^2} I_{101110200}^{\tilde{A}} \right) \right],$$

Sector 94:  $$J_{28}^{\tilde{A}} = \varepsilon^4 \left( \frac{-s}{\mu^2} \right) I_{011110100}^{\tilde{A}},$$

Sector 121:  $$J_{29}^{\tilde{A}} = \varepsilon^4 \left( \frac{-s}{\mu^2} \right) I_{100111100}^{\tilde{A}},$$

$$J_{30}^{\tilde{A}} = \varepsilon^3 \left( \frac{r_6}{\mu^4} \right) I_{200111100}^{\tilde{A}},$$

$$J_{31}^{\tilde{A}} = \varepsilon^3 \left( \frac{-s}{\mu^2} \right) \left( \frac{m^2}{\mu^2} \right) I_{100211100}^{\tilde{A}},$$

$$J_{32}^{\tilde{A}} = \varepsilon^3 \left( \frac{m^2}{\mu^2} \right) \left[ \left( \frac{-s-t}{\mu^2} \right) I_{100121100}^{\tilde{A}} + \left( \frac{-t}{\mu^2} \right) I_{100211100}^{\tilde{A}} \right],$$

Sector 122:  $$J_{33}^{\tilde{A}} = \varepsilon^4 \left( \frac{-s}{\mu^2} \right) I_{010111100}^{\tilde{A}},$$

$$J_{34}^{\tilde{A}} = \varepsilon^3 \left( \frac{-s}{\mu^2} \right) \left( \frac{m^2}{\mu^2} \right) I_{020111100}^{\tilde{A}},$$

Sector 63:  $$J_{35}^{\tilde{A}} = \varepsilon^4 \left( \frac{-s}{\mu^2} \right) \left( \frac{-s-t}{\mu^2} \right) I_{111111000}^{\tilde{A}},$$

$$J_{36}^{\tilde{A}} = \varepsilon^4 \left( \frac{-s}{\mu^2} \right) I_{1111110(-1)0}^{\tilde{A}},$$

Sector 95:  $$J_{37}^{\tilde{A}} = \varepsilon^4 \left( \frac{-s}{\mu^2} \right) \left( \frac{-t}{\mu^2} \right) I_{111110100}^{\tilde{A}},$$

Sector 123:  $$J_{38}^{\tilde{A}} = \varepsilon^4 \frac{\pi}{\psi_1^{(b)}} \left( \frac{-s}{\mu^2} \right) I_{110111100}^{\tilde{A}},$$

$$J_{39}^{\tilde{A}} = \varepsilon^4 \left( \frac{-s-t}{\mu^2} \right) I_{1101111(-1)0}^{\tilde{A}} + F_{39,38}^{\tilde{A}} J_{38}^{\tilde{A}} - \left( \frac{t}{s} \right) J_{33}^{\tilde{A}},$$

$$J_{40}^{\tilde{A}} = \varepsilon^4 \left[ \left( \frac{-t}{\mu^2} \right) I_{11(-1)111100}^{\tilde{A}} + \left( \frac{-s}{\mu^2} \right) I_{1101111(-1)0}^{\tilde{A}} - \left( \frac{-t}{\mu^2} \right) I_{100111100}^{\tilde{A}} \right],$$

$$\begin{aligned} J_{41}^{\tilde{A}} = {} & \frac{\left( \psi_1^{(b)} \right)^2}{2\pi i W_{m^2}^{(b)} \varepsilon} \frac{\partial}{\partial m^2} J_{38}^{\tilde{A}} + F_{41,38}^{\tilde{A}} J_{38}^{\tilde{A}} \\ & + \frac{1}{4} F_{41,40}^{\tilde{A}} \Big[ -12 J_3^{\tilde{A}} - 53 J_4^{\tilde{A}} + 12 J_5^{\tilde{A}} - 35 J_8^{\tilde{A}} - 59 J_{10}^{\tilde{A}} + 7 J_{13}^{\tilde{A}} - 26 J_{14}^{\tilde{A}} \\ & + 20 J_{15}^{\tilde{A}} + 18 J_{17}^{\tilde{A}} + 10 J_{18}^{\tilde{A}} + 26 J_{19}^{\tilde{A}} - 26 J_{24}^{\tilde{A}} + 18 J_{25}^{\tilde{A}} - 46 J_{27}^{\tilde{A}} - 120 J_{29}^{\tilde{A}} \\ & + 104 J_{31}^{\tilde{A}} + 51 J_{32}^{\tilde{A}} + 38 J_{33}^{\tilde{A}} - 10 J_{34}^{\tilde{A}} - 84 J_{39}^{\tilde{A}} + 60 J_{40}^{\tilde{A}} \Big] \end{aligned}$$

$$+ F^{\tilde{A}}_{41,26} J^{\tilde{A}}_{26} + F^{\tilde{A}}_{41,30} J^{\tilde{A}}_{30},$$

Sector 126: $\quad J^{\tilde{A}}_{42} = \varepsilon^4 \frac{\pi}{\psi^{(c)}_1} \left( \frac{-s}{\mu^2} \right) I^{\tilde{A}}_{011111100},$

$$J^{\tilde{A}}_{43} = \varepsilon^4 \left( \frac{s^2}{\mu^4} \right) \left( \frac{\mu^2}{-s - 2t} \right) I^{\tilde{A}}_{(-1)11111100} + \frac{1}{3} \left( 2\frac{m^2}{\mu^2} - \frac{s(2s+t)}{s+2t} \right) \frac{\psi^{(c)}_1}{\pi} J^{\tilde{A}}_{42}$$

$$- \left( \frac{s}{s+2t} \right) J^{\tilde{A}}_{33},$$

$$J^{\tilde{A}}_{44} = \frac{\left( \psi^{(c)}_1 \right)^2}{2\pi i W^{(c)}_{m^2} \varepsilon} \frac{\partial}{\partial m^2} J^{\tilde{A}}_{42} + \frac{\left( \psi^{(c)}_1 \right)^2}{\pi} \frac{76m^4 - 44m^2 s - 5s^2}{6im^2 \left( 8m^4 - 7m^2 s - s^2 \right) W^{(c)}_{m^2}} J^{\tilde{A}}_{42},$$

Sector 127 $\quad J^{\tilde{A}}_{45} = \varepsilon^4 \frac{\pi}{\psi^{(c)}_1} \left[ \left( \frac{-s}{\mu^2} \right) I^{\tilde{A}}_{1111111(-1)0} - \frac{(s+2t)}{\mu^2} I^{\tilde{A}}_{110111100} \right],$

$$J^{\tilde{A}}_{46} = \varepsilon^4 \left( \frac{-s}{\mu^2} \right) \left( \frac{r_6}{\mu^2} \right) I^{\tilde{A}}_{111111100} + F^{\tilde{A}}_{46,45} \left[ J^{\tilde{A}}_{45} - J^{\tilde{A}}_{42} \right] + F^{\tilde{A}}_{46,38} J^{\tilde{A}}_{38},$$

$$J^{\tilde{A}}_{47} = \varepsilon^4 \left( \frac{-s}{\mu^2} \right) I_{1111111(-2)0} + \frac{\psi^{(c)}_1}{\pi} \frac{2m^2 - 2s - 3t}{3\mu^2} J^{\tilde{A}}_{45} - 2 \left( \frac{t}{s} \right) J^{\tilde{A}}_{43}$$

$$+ \frac{\psi^{(c)}_1}{\pi} \frac{4(m^2 - s)(s+t)}{3s} J^{\tilde{A}}_{42} - \left( \frac{s}{t} \right) J^{\tilde{A}}_{40} + \left( \frac{s}{t} + \frac{t}{s+t} \right) J^{\tilde{A}}_{39}$$

$$+ F^{\tilde{A}}_{47,38} J^{\tilde{A}}_{38} + \frac{s}{s+t} J^{\tilde{A}}_{33},$$

$$J^{\tilde{A}}_{48} = \frac{\left( \psi^{(c)}_1 \right)^2}{2\pi i W^{(c)}_{m^2} \varepsilon} \frac{\partial}{\partial m^2} J^{\tilde{A}}_{45} + F^{\tilde{A}}_{48,46} J^{\tilde{A}}_{46} + F^{\tilde{A}}_{48,45} J^{\tilde{A}}_{45} + F^{\tilde{A}}_{48,42} J^{\tilde{A}}_{42} + F^{\tilde{A}}_{48,38} J^{\tilde{A}}_{38} + F^{\tilde{A}}_{48,30} J^{\tilde{A}}_{30}.$$

The functions $F^{\tilde{A}}_{39,38}$, $F^{\tilde{A}}_{41,30}$, $F^{\tilde{A}}_{41,26}$, $F^{\tilde{A}}_{46,45}$, $F^{\tilde{A}}_{48,42}$, $F^{\tilde{A}}_{46,38}$, $F^{\tilde{A}}_{47,38}$, $F^{\tilde{A}}_{48,38}$ and $F^{\tilde{A}}_{48,30}$ are determined by a triangular system of first-order differential equations. The functions $F^{\tilde{A}}_{41,40}$, $F^{\tilde{A}}_{41,38}$, $F^{\tilde{A}}_{48,46}$ and $F^{\tilde{A}}_{48,45}$ are related algebraically to the ones above. As the explicit expressions are rather long, these functions are given in the supplementary electronic file attached to the arxiv version of this article, see appendix B.

### 3.2.2 Topology $\tilde{B}$

Sector 65: $\quad J^{\tilde{B}}_1 = \varepsilon^2 I^{\tilde{B}}_{200000200},$

Sector 41: $\quad J^{\tilde{B}}_2 = \varepsilon^2 \left( \frac{m^2}{\mu^2} \right) \mathbf{D}^- I^{\tilde{B}}_{100101000},$

Sector 42: $\quad J^{\tilde{B}}_3 = \varepsilon^2 \left( \frac{-s}{\mu^2} \right) \mathbf{D}^- I^{\tilde{B}}_{010101000},$

$$\text{Sector 49:} \quad J_4^{\tilde{B}} = \varepsilon^2 \left( \frac{m^2 + s + t}{\mu^2} \right) \mathbf{D}^- I_{100011000}^{\tilde{B}},$$

$$J_5^{\tilde{B}} = \varepsilon^2 \left[ \left( \frac{m^2}{\mu^2} \right) \mathbf{D}^- I_{100011000}^{\tilde{B}} + \mathbf{D}^- I_{10001100(-1)}^{\tilde{B}} \right],$$

$$\text{Sector 70:} \quad J_6^{\tilde{B}} = \varepsilon^2 \left( \frac{-s}{\mu^2} \right) \mathbf{D}^- I_{011000100}^{\tilde{B}},$$

$$\text{Sector 73:} \quad J_7^{\tilde{B}} = \varepsilon^2 \left( \frac{r_1}{\mu^2} \right) \mathbf{D}^- I_{100100100}^{\tilde{B}},$$

$$J_8^{\tilde{B}} = \varepsilon^2 \left( \frac{-t}{\mu^2} \right) I_{200100200}^{\tilde{B}},$$

$$\text{Sector 84:} \quad J_9^{\tilde{B}} = \varepsilon^2 \left( \frac{m^2 - s}{\mu^2} \right) \mathbf{D}^- I_{001010100}^{\tilde{B}},$$

$$J_{10}^{\tilde{B}} = \varepsilon^2 \left( \frac{-s}{\mu^2} \right) I_{001020200}^{\tilde{B}},$$

$$\text{Sector 43:} \quad J_{11}^{\tilde{B}} = \varepsilon^3 \left( \frac{-s}{\mu^2} \right) I_{110201000}^{\tilde{B}},$$

$$\text{Sector 54:} \quad J_{12}^{\tilde{B}} = \varepsilon^3 \left( \frac{-s}{\mu^2} \right) I_{011021000}^{\tilde{B}},$$

$$\text{Sector 71:} \quad J_{13}^{\tilde{B}} = \varepsilon^3 \left( \frac{-s}{\mu^2} \right) I_{111000200}^{\tilde{B}},$$

$$\text{Sector 75:} \quad J_{14}^{\tilde{B}} = \varepsilon^3 \left( \frac{-t}{\mu^2} \right) I_{110100200}^{\tilde{B}},$$

$$\text{Sector 78:} \quad J_{15}^{\tilde{B}} = \varepsilon^3 \left( \frac{-s}{\mu^2} \right) I_{011200100}^{\tilde{B}},$$

$$J_{16}^{\tilde{B}} = \varepsilon^3 \left( \frac{-s}{\mu^2} \right) I_{011100200}^{\tilde{B}},$$

$$\text{Sector 85:} \quad J_{17}^{\tilde{B}} = \varepsilon^3 \left( \frac{-s}{\mu^2} \right) I_{101020100}^{\tilde{B}},$$

$$J_{18}^{\tilde{B}} = \varepsilon^3 \left( \frac{-s}{\mu^2} \right) I_{101010200}^{\tilde{B}},$$

$$\text{Sector 113:} \quad J_{19}^{\tilde{B}} = \varepsilon^3 \left( \frac{-s - t}{\mu^2} \right) I_{200011100}^{\tilde{B}},$$

$$J_{20}^{\tilde{B}} = \varepsilon^3 \left( \frac{-s - t}{\mu^2} \right) I_{100021100}^{\tilde{B}},$$

$$\text{Sector 55:} \quad J_{21}^{\tilde{B}} = \varepsilon^3 (1 - 2\varepsilon) \left( \frac{-s}{\mu^2} \right) I_{111011000}^{\tilde{B}},$$

$$J_{22}^{\tilde{B}} = \varepsilon^3 \left( \frac{-s}{\mu^2} \right) \left( \frac{m^2 + s + t}{\mu^2} \right) I_{111021000}^{\tilde{B}},$$

Sector 59: $\quad J_{23}^{\tilde{B}} = \varepsilon^4 \left( \dfrac{-t}{\mu^2} \right) I_{110111000}^{\tilde{B}},$

$$J_{24}^{\tilde{B}} = \varepsilon^3 \left( \dfrac{-t}{\mu^2} \right) \left( \dfrac{m^2}{\mu^2} \right) I_{210111000}^{\tilde{B}},$$

Sector 79: $\quad J_{25}^{\tilde{B}} = \varepsilon^3 \left( \dfrac{-s}{\mu^2} \right) \left( \dfrac{m^2 - t}{\mu^2} \right) I_{111100200}^{\tilde{B}},$

$$J_{26}^{\tilde{B}} = \varepsilon^3 \left( \dfrac{-s}{\mu^2} \right) \left[ \left( \dfrac{m^2 - t}{\mu^2} \right) I_{111100200}^{\tilde{B}} + I_{1111002(-1)0}^{\tilde{B}} + I_{110100200}^{\tilde{B}} \right],$$

$$J_{27}^{\tilde{B}} = \varepsilon^3 \left( \dfrac{r_4}{\mu^2} \right) \left[ I_{111100200}^{\tilde{B}} + I_{111200100}^{\tilde{B}} \right],$$

Sector 91: $\quad J_{28}^{\tilde{B}} = \varepsilon^4 \left( \dfrac{-t}{\mu^2} \right) I_{110110100}^{\tilde{B}},$

Sector 93: $\quad J_{29}^{\tilde{B}} = \varepsilon^4 \left( \dfrac{-s-t}{\mu^2} \right) I_{101110100}^{\tilde{B}},$

$$J_{30}^{\tilde{B}} = \varepsilon^3 \left( \dfrac{r_3}{\mu^4} \right) I_{101110200}^{\tilde{B}},$$

$$J_{31}^{\tilde{B}} = \varepsilon^3 \left( \dfrac{m^2}{\mu^2} \right) \left( \dfrac{-s-t}{\mu^2} \right) I_{201110100}^{\tilde{B}} - \left( \dfrac{-s(-2m^2+t) + m^2 t}{2 r_3} \right) J_{30}^{\tilde{B}},$$

Sector 94: $\quad J_{32}^{\tilde{B}} = \varepsilon^4 \left( \dfrac{-s}{\mu^2} \right) I_{011110100}^{\tilde{B}},$

Sector 107: $\quad J_{33}^{\tilde{B}} = \varepsilon^4 \left( \dfrac{-s-t}{\mu^2} \right) I_{110101100}^{\tilde{B}},$

$$J_{34}^{\tilde{B}} = \varepsilon^3 \left( \dfrac{r_4}{\mu^4} \right) I_{110201100}^{\tilde{B}},$$

$$J_{35}^{\tilde{B}} = \varepsilon^3 \left( \dfrac{m^2}{\mu^2} \right) \left( \dfrac{-s-t}{\mu^2} \right) I_{110101200}^{\tilde{B}},$$

$$J_{36}^{\tilde{B}} = \varepsilon^3 \left( \dfrac{m^2}{\mu^2} \right) \left[ \left( \dfrac{-s}{\mu^2} \right) I_{110102100}^{\tilde{B}} - \left( \dfrac{-t}{\mu^2} \right) I_{110101200}^{\tilde{B}} \right],$$

Sector 109: $\quad J_{37}^{\tilde{B}} = \varepsilon^4 \left( \dfrac{-t}{\mu^2} \right) I_{101101100}^{\tilde{B}},$

Sector 110: $\quad J_{38}^{\tilde{B}} = \varepsilon^4 \left( \dfrac{-s}{\mu^2} \right) I_{011101100}^{\tilde{B}},$

Sector 115: $\quad J_{39}^{\tilde{B}} = \varepsilon^4 \left( \dfrac{-s-t}{\mu^2} \right) I_{110011100}^{\tilde{B}},$

$$J_{40}^{\tilde{B}} = \varepsilon^3 \left( \dfrac{m^2}{\mu^2} \right) \left( \dfrac{-s-t}{\mu^2} \right) I_{110021100}^{\tilde{B}},$$

Sector 117: $\quad J_{41}^{\tilde{B}} = \varepsilon^4 \left( \dfrac{-t}{\mu^2} \right) I_{101011100}^{\tilde{B}},$

$$J_{42}^{\tilde{B}} = \varepsilon^3 \left(\frac{-t}{\mu^2}\right) \left(\frac{m^2}{\mu^2}\right) I_{201011100}^{\tilde{B}},$$

$$J_{43}^{\tilde{B}} = \varepsilon^3 \left(\frac{-m^2 t + s^2 + st}{\mu^2}\right) I_{101021100}^{\tilde{B}},$$

$$J_{44}^{\tilde{B}} = \varepsilon^3 \left(\frac{-t}{\mu^2}\right) \left(\frac{m^2}{\mu^2}\right) I_{101011200}^{\tilde{B}},$$

Sector 118: $\quad J_{45}^{\tilde{B}} = \varepsilon^4 \left(\dfrac{-s}{\mu^2}\right) I_{011011100}^{\tilde{B}},$

$$J_{46}^{\tilde{B}} = \varepsilon^3 \left(\frac{-s}{\mu^2}\right) \left(\frac{m^2}{\mu^2}\right) I_{011021100}^{\tilde{B}},$$

Sector 121: $\quad J_{47}^{\tilde{B}} = \varepsilon^4 \left(\dfrac{-s}{\mu^2}\right) I_{100111100}^{\tilde{B}},$

$$J_{48}^{\tilde{B}} = \varepsilon^3 \left(\frac{r_7}{\mu^4}\right) I_{200111100}^{\tilde{B}},$$

$$J_{49}^{\tilde{B}} = \varepsilon^3 \left[\left(\frac{m^2}{\mu^2}\right)\left(\frac{-s}{\mu^2}\right) I_{100111200}^{\tilde{B}} \right.$$
$$\left. -\frac{1}{2}\left(\frac{2m^2 s + m^2 t - st - t^2}{\mu^4}\right) I_{200111100}^{\tilde{B}}\right],$$

Sector 122: $\quad J_{50}^{\tilde{B}} = \varepsilon^4 \left(\dfrac{-s}{\mu^2}\right) I_{010111100}^{\tilde{B}},$

Sector 119: $\quad J_{51}^{\tilde{B}} = \varepsilon^4 \left(\dfrac{-s}{\mu^2}\right)\left(\dfrac{m^2 + s + t}{\mu^2}\right) I_{111011100}^{\tilde{B}},$

$$J_{52}^{\tilde{B}} = \varepsilon^3 \left(\frac{r_8}{\mu^4}\right)\left(\frac{m^2}{\mu^2}\right)\left[2 I_{111021100}^{\tilde{B}} + I_{111011200}^{\tilde{B}}\right]$$
$$+ \left(\frac{r_8}{10s(m^2 + s + t)}\right)\left[-5 J_2^{\tilde{B}} + 6 J_4^{\tilde{B}} - 6 J_5^{\tilde{B}} + 5 J_9^{\tilde{B}} - 18 J_{10}^{\tilde{B}}\right.$$
$$+ 4 J_{17}^{\tilde{B}} + 8 J_{18}^{\tilde{B}} - 28 J_{19}^{\tilde{B}} - 24 J_{20}^{\tilde{B}} + 20 J_{22}^{\tilde{B}} - 20 J_{39}^{\tilde{B}} - 20 J_{40}^{\tilde{B}} - 36 J_{41}^{\tilde{B}}$$
$$\left.+ 24 J_{42}^{\tilde{B}} + \left(\frac{4(4s(s+t) + m^2(5s+t))}{s(s+t) - m^2 t}\right) J_{43}^{\tilde{B}} + 4 J_{44}^{\tilde{B}}\right],$$

$$J_{53}^{\tilde{B}} = \varepsilon^4 \left(\frac{-s}{\mu^2}\right)\left[2\left(\frac{-s-t}{\mu^2}\right) I_{111011100}^{\tilde{B}} - I_{011011100}^{\tilde{B}} + I_{1110111(-1)0}^{\tilde{B}}\right.$$
$$\left.+ I_{101011100}^{\tilde{B}}\right] + \varepsilon^4 \left(\frac{-t}{\mu^2}\right) I_{110011100}^{\tilde{B}},$$

Sector 123: $\quad J_{54}^{\tilde{B}} = \varepsilon^4 \dfrac{\pi}{\psi_1^{(a)}} \left(\dfrac{-s}{\mu^2}\right) I_{110111100}^{\tilde{B}},$

$$J_{55}^{\tilde{B}} = \varepsilon^4 \left[\left(\frac{-s-t}{\mu^2}\right) I_{1101111(-1)0}^{\tilde{B}} - \left(\frac{-t}{\mu^2}\right) I_{010111100}^{\tilde{B}}\right] + F_{55,54}^{\tilde{B}} J_{54}^{\tilde{B}},$$

$$J_{56}^{\tilde{B}} = \varepsilon^4 \left( \frac{-t}{\mu^2} \right) \left[ I_{11(-1)111100}^{\tilde{B}} - I_{1101111(-1)0}^{\tilde{B}} + I_{010111100}^{\tilde{B}} \right] + F_{56,54}^{\tilde{B}} J_{54}^{\tilde{B}}$$
$$- \left( \frac{t}{s} \right) J_{47}^{\tilde{B}},$$

$$J_{57}^{\tilde{B}} = \frac{\left( \psi_1^{(a)} \right)^2}{2\pi i W_{m^2}^{(a)} \varepsilon} \frac{\partial}{\partial m^2} J_{54}^{\tilde{B}} + \frac{1}{8} F_{57,55}^{\tilde{B}} \left[ 5J_4^{\tilde{B}} - 5J_5^{\tilde{B}} - 2J_8^{\tilde{B}} - 22J_{11}^{\tilde{B}} - 14J_{14}^{\tilde{B}} \right.$$
$$- 4J_{19}^{\tilde{B}} - 8J_{20}^{\tilde{B}} + 36J_{23}^{\tilde{B}} - 20J_{24}^{\tilde{B}} - 20J_{28}^{\tilde{B}} + 16J_{35}^{\tilde{B}} + 6J_{36}^{\tilde{B}} - 20J_{39}^{\tilde{B}}$$
$$- 20J_{40}^{\tilde{B}} - 12J_{47}^{\tilde{B}} + 4J_{49}^{\tilde{B}} - 44J_{50}^{\tilde{B}} + 48J_{55}^{\tilde{B}} + 24J_{56}^{\tilde{B}} \Big]$$
$$- F_{57,34}^{\tilde{B}} J_{34}^{\tilde{B}} - F_{57,48}^{\tilde{B}} J_{48}^{\tilde{B}} + F_{57,54}^{\tilde{B}} J_{54}^{\tilde{B}},$$

Sector 125: $\quad J_{58}^{\tilde{B}} = \varepsilon^4 \left( \frac{-t}{\mu^2} \right) \left( \frac{m^2}{\mu^2} \right) I_{101111100}^{\tilde{B}},$

$$J_{59}^{\tilde{B}} = \varepsilon^4 \left( \frac{-t}{\mu^2} \right) \left[ \left( \frac{-s-t}{\mu^2} \right) I_{101111100}^{\tilde{B}} + I_{1011111(-1)0}^{\tilde{B}} \right] - \frac{t}{4s} \left[ J_2^{\tilde{B}} - J_9^{\tilde{B}} + 4J_{10}^{\tilde{B}} \right],$$

Sector 126: $\quad J_{60}^{\tilde{B}} = \varepsilon^4 \left( \frac{-s}{\mu^2} \right) \left( \frac{m^2+s}{\mu^2} \right) I_{011111100}^{\tilde{B}},$

Sector 127: $\quad J_{61}^{\tilde{B}} = \varepsilon^4 \left( \frac{r_4}{\mu^4} \right) \left[ \left( \frac{m^2-s-t}{\mu^2} \right) I_{111111100}^{\tilde{B}} + I_{1111111(-1)0}^{\tilde{B}} - I_{011111100}^{\tilde{B}} \right]$
$$+ F_{61,54}^{\tilde{B}} J_{54}^{\tilde{B}},$$

$$J_{62}^{\tilde{B}} = \varepsilon^4 \left( \frac{-s}{\mu^2} \right) \left\{ \left( \frac{m^2+s+t}{\mu^2} \right) \left[ \left( \frac{2m^2-t}{\mu^2} \right) I_{111111100}^{\tilde{B}} \right. \right.$$
$$\left. + I_{1111111(-1)0}^{\tilde{B}} + I_{101111100}^{\tilde{B}} \right] + \left( \frac{-t}{\mu^2} \right) I_{011111100}^{\tilde{B}} \Bigg\} - F_{62,54}^{\tilde{B}} J_{54}^{\tilde{B}},$$

$$J_{63}^{\tilde{B}} = \varepsilon^4 \left( \frac{-s}{\mu^2} \right) \left[ \left( \frac{-t}{\mu^2} \right) I_{1111111(-1)0}^{\tilde{B}} + \left( \frac{-s}{\mu^2} \right) I_{11111110(-1)}^{\tilde{B}} \right.$$
$$- \left( \frac{2m^2-t}{\mu^2} \right) \left( \frac{s+t}{\mu^2} \right) I_{111111100}^{\tilde{B}} - \left( \frac{m^2-t}{\mu^2} \right) I_{011111100}^{\tilde{B}} \Bigg]$$
$$- \left( \frac{s(s+t)}{m^2 t} \right) J_{58}^{\tilde{B}} + F_{63,54}^{\tilde{B}} J_{54}^{\tilde{B}}$$
$$+ \frac{1}{12} \left( \frac{m^2}{m^2+s+t} \right) \left[ 12J_{51}^{\tilde{B}} - 24J_{24}^{\tilde{B}} + 36J_{23}^{\tilde{B}} + 12J_{22}^{\tilde{B}} + 3J_4^{\tilde{B}} + J_3^{\tilde{B}} \right.$$
$$\left. - 3J_2^{\tilde{B}} \right] + \left( \frac{s+t}{m^2-t} \right) \left[ J_{38}^{\tilde{B}} + J_{37}^{\tilde{B}} - J_{36}^{\tilde{B}} - 3J_{35}^{\tilde{B}} + 3J_{33}^{\tilde{B}} \right]$$
$$+ \frac{1}{2} \left( \frac{t}{m^2-t} \right) \left[ -2J_{32}^{\tilde{B}} - 4J_{31}^{\tilde{B}} + 6J_{29}^{\tilde{B}} - 2J_{28}^{\tilde{B}} + 2J_{17}^{\tilde{B}} - 4J_{10}^{\tilde{B}} \right.$$

$$+ J_9^{\tilde{B}} - 3J_8^{\tilde{B}} - J_2^{\tilde{B}} \Big]$$

$$+ \left( \frac{s}{m^2 - t} \right) \left[ -J_{25}^{\tilde{B}} + J_{16}^{\tilde{B}} + 2J_{14}^{\tilde{B}} \right] + \left( \frac{(2m^2 + s)(s + t)}{(m^2 - t)(m^2 + s + t)} \right) J_{11}^{\tilde{B}},$$

$$J_{64}^{\tilde{B}} = \varepsilon^4 \left( \frac{-s}{\mu^2} \right) \left[ I_{1111111(-2)0}^{\tilde{B}} + \left( \frac{4m^2 - t}{\mu^2} \right) I_{1111111(-1)0}^{\tilde{B}} \right.$$

$$+ \left( \frac{2m^2}{\mu^2} \right) \left( \frac{2m^2 - t}{\mu^2} \right) I_{111111100}^{\tilde{B}} + \left( \frac{m^2 - s}{\mu^2} \right) \left( \frac{t}{s} \right) I_{011111100}^{\tilde{B}} \right]$$

$$+ \left( \frac{s}{t} \right) J_{59}^{\tilde{B}} + \left( \frac{s(2m^2 + s + t)}{m^2 t} \right) J_{58}^{\tilde{B}} - \frac{1}{2} \left( \frac{s - t}{s + t} \right) \left[ J_{55}^{\tilde{B}} - J_{50}^{\tilde{B}} \right]$$

$$- F_{64,54}^{\tilde{B}} J_{54}^{\tilde{B}} + \frac{1}{12} \left( \frac{t}{s} \right) \left[ 12 J_{38}^{\tilde{B}} + 12 J_{32}^{\tilde{B}} + 12 J_{16}^{\tilde{B}} + 12 J_{12}^{\tilde{B}} + 12 J_{10}^{\tilde{B}} \right.$$

$$\left. - 3 J_9^{\tilde{B}} + J_3^{\tilde{B}} + 3 J_2^{\tilde{B}} \right].$$

The functions $F_{55,54}^{\tilde{B}}$, $F_{56,54}^{\tilde{B}}$, $F_{57,48}^{\tilde{B}}$, $F_{57,34}^{\tilde{B}}$, $F_{61,54}^{\tilde{B}}$, $F_{62,54}^{\tilde{B}}$ and $F_{64,54}^{\tilde{B}}$ are determined by a triangular system of first-order differential equations. The functions $F_{57,55}^{\tilde{B}}$, $F_{57,54}^{\tilde{B}}$ and $F_{63,54}^{\tilde{B}}$ are related algebraically to the ones above. As the explicit expressions are rather long, these functions are given in the supplementary electronic file attached to the arxiv version of this article, see appendix B.

### 3.2.3 Topology $\tilde{C}$

Sector 14: $\quad J_1^{\tilde{C}} = \varepsilon^2 \left( \frac{-s}{\mu^2} \right) \mathbf{D}^- I_{011100000}^{\tilde{C}},$

Sector 41: $\quad J_2^{\tilde{C}} = \varepsilon^2 \left( \frac{m^2}{\mu^2} \right) \mathbf{D}^- I_{100101000}^{\tilde{C}},$

Sector 42: $\quad J_3^{\tilde{C}} = \varepsilon^2 \left( \frac{-s}{\mu^2} \right) I_{020201000}^{\tilde{C}},$

$\qquad\qquad\quad J_4^{\tilde{C}} = \varepsilon^2 \left( \frac{m^2 - s}{\mu^2} \right) \mathbf{D}^- I_{010101000}^{\tilde{C}},$

Sector 49: $\quad J_5^{\tilde{C}} = \varepsilon^2 \left( \frac{-s - t}{\mu^2} \right) \mathbf{D}^- I_{100011000}^{\tilde{C}},$

Sector 73: $\quad J_6^{\tilde{C}} = \varepsilon^2 \left( \frac{m^2 - t}{\mu^2} \right) \mathbf{D}^- I_{100100100}^{\tilde{C}},$

$\qquad\qquad\quad J_7^{\tilde{C}} = \varepsilon^2 \left( \frac{-t}{\mu^2} \right) I_{100200200}^{\tilde{C}},$

Sector 84: $\quad J_8^{\tilde{C}} = \varepsilon^2 \left( \frac{-s}{\mu^2} \right) \mathbf{D}^- I_{001010100}^{\tilde{C}},$

Sector 54: $\quad J_9^{\tilde{C}} = \varepsilon^3 \left( \dfrac{-s}{\mu^2} \right) I_{011021000}^{\tilde{C}},$

Sector 57: $\quad J_{10}^{\tilde{C}} = \varepsilon^3 \left( \dfrac{-s-t}{\mu^2} \right) I_{100112000}^{\tilde{C}},$

Sector 78: $\quad J_{11}^{\tilde{C}} = \varepsilon^3 \left( \dfrac{-s}{\mu^2} \right) I_{011200100}^{\tilde{C}},$

$\quad\qquad\qquad J_{12}^{\tilde{C}} = \varepsilon^3 \left( \dfrac{-s}{\mu^2} \right) I_{011100200}^{\tilde{C}},$

Sector 92: $\quad J_{13}^{\tilde{C}} = \varepsilon^3 \left( \dfrac{-s}{\mu^2} \right) I_{001110200}^{\tilde{C}},$

Sector 55: $\quad J_{14}^{\tilde{C}} = \varepsilon^3 \left( \dfrac{-s}{\mu^2} \right) \left( \dfrac{-s-t}{\mu^2} \right) I_{111021000}^{\tilde{C}},$

Sector 59: $\quad J_{15}^{\tilde{C}} = \varepsilon^4 \left( \dfrac{-t}{\mu^2} \right) I_{110111000}^{\tilde{C}},$

$\quad\qquad\qquad J_{16}^{\tilde{C}} = \varepsilon^3 \left( \dfrac{-t}{\mu^2} \right) \left( \dfrac{m^2}{\mu^2} \right) I_{110211000}^{\tilde{C}},$

Sector 61: $\quad J_{17}^{\tilde{C}} = \varepsilon^4 \left( \dfrac{-s-t}{\mu^2} \right) I_{101111000}^{\tilde{C}},$

Sector 62: $\quad J_{18}^{\tilde{C}} = \varepsilon^4 \left( \dfrac{-s}{\mu^2} \right) I_{011111000}^{\tilde{C}},$

$\quad\qquad\qquad J_{19}^{\tilde{C}} = \varepsilon^3 \left( \dfrac{-s}{\mu^2} \right) \left( \dfrac{m^2}{\mu^2} \right) I_{011211000}^{\tilde{C}},$

Sector 79: $\quad J_{20}^{\tilde{C}} = \varepsilon^3 \left( \dfrac{-s}{\mu^2} \right) \left( \dfrac{-t}{\mu^2} \right) I_{111200100}^{\tilde{C}},$

$\quad\qquad\qquad J_{21}^{\tilde{C}} = \varepsilon^3 \left( \dfrac{-s}{\mu^2} \right) I_{1112001(-1)0}^{\tilde{C}} - \left( \dfrac{s}{2t} \right) \left[ J_2^{\tilde{C}} - J_6^{\tilde{C}} + 4 J_7^{\tilde{C}} \right],$

Sector 93: $\quad J_{22}^{\tilde{C}} = \varepsilon^4 \left( \dfrac{-s-t}{\mu^2} \right) I_{101110100}^{\tilde{C}},$

$\quad\qquad\qquad J_{23}^{\tilde{C}} = \varepsilon^3 \left( \dfrac{-s-t}{\mu^2} \right) \left( \dfrac{m^2}{\mu^2} \right) I_{101210100}^{\tilde{C}},$

Sector 94: $\quad J_{24}^{\tilde{C}} = \varepsilon^4 \left( \dfrac{-s}{\mu^2} \right) I_{011110100}^{\tilde{C}},$

Sector 107: $\quad J_{25}^{\tilde{C}} = \varepsilon^4 \left( \dfrac{-s-t}{\mu^2} \right) I_{110101100}^{\tilde{C}},$

$\quad\qquad\qquad J_{26}^{\tilde{C}} = \varepsilon^3 \left( \dfrac{(-s-t)m^2 + st}{\mu^4} \right) I_{110201100}^{\tilde{C}},$

Sector 109: $\quad J_{27}^{\tilde{C}} = \varepsilon^4 \left( \dfrac{-t}{\mu^2} \right) I_{101101100}^{\tilde{C}},$

Sector 110: $\quad J_{28}^{\tilde{C}} = \varepsilon^4 \left( \dfrac{-s}{\mu^2} \right) I_{011101100}^{\tilde{C}},$

Sector 117: $\quad J_{29}^{\tilde{C}} = \varepsilon^4 \left( \dfrac{-t}{\mu^2} \right) I_{101011100}^{\tilde{C}},$

Sector 121: $\quad J_{30}^{\tilde{C}} = \varepsilon^4 \left( \dfrac{-s}{\mu^2} \right) I_{100111100}^{\tilde{C}},$

$$J_{31}^{\tilde{C}} = \varepsilon^3 \left( \dfrac{-s}{\mu^2} \right) \left( \dfrac{m^2}{\mu^2} \right) I_{100211100}^{\tilde{C}},$$

Sector 124: $\quad J_{32}^{\tilde{C}} = \varepsilon^4 \left( \dfrac{-s}{\mu^2} \right) I_{001111100}^{\tilde{C}},$

Sector 63: $\quad J_{33}^{\tilde{C}} = \varepsilon^4 \left( \dfrac{-s}{\mu^2} \right) \left( \dfrac{-s-t}{\mu^2} \right) I_{111111000}^{\tilde{C}},$

$$J_{34}^{\tilde{C}} = \varepsilon^4 \left( \dfrac{-s}{\mu^2} \right) I_{1111110(-1)0}^{\tilde{C}},$$

Sector 123: $\quad J_{35}^{\tilde{C}} = \varepsilon^4 \left( \dfrac{-s-t}{\mu^2} \right) \left( \dfrac{m^2}{\mu^2} \right) I_{110111100}^{\tilde{C}},$

Sector 125: $\quad J_{36}^{\tilde{C}} = \varepsilon^4 \left( \dfrac{r_5}{\mu^4} \right) I_{101111100}^{\tilde{C}},$

$$J_{37}^{\tilde{C}} = \varepsilon^4 \left[ \left( \dfrac{-s-t}{\mu^2} \right) I_{1(-1)1111100}^{\tilde{C}} - \dfrac{1}{2} \left( \dfrac{-s}{\mu^2} \right) \left( \dfrac{m^2 - 2(s+t)}{\mu^2} \right) I_{101111100}^{\tilde{C}} \right]$$
$$- \left( \dfrac{s}{2t} \right) \left[ J_{10}^{\tilde{C}} + J_{13}^{\tilde{C}} - J_{17}^{\tilde{C}} + 3 J_{22}^{\tilde{C}} - J_{23}^{\tilde{C}} + J_{27}^{\tilde{C}} - J_{29}^{\tilde{C}} + 3 J_{30}^{\tilde{C}} - J_{31}^{\tilde{C}} \right.$$
$$\left. - J_{32}^{\tilde{C}} \right] - \left( \dfrac{t}{s} \right) J_{30}^{\tilde{C}},$$

Sector 126: $\quad J_{38}^{\tilde{C}} = \varepsilon^4 \left( \dfrac{-s}{\mu^2} \right) \left( \dfrac{m^2 + s}{\mu^2} \right) I_{011111100}^{\tilde{C}},$

Sector 127: $\quad J_{39}^{\tilde{C}} = \varepsilon^4 \left( \dfrac{s^2}{\mu^4} \right) I_{11111110(-1)}^{\tilde{C}}$
$$- \left( \dfrac{s}{2t} \right) \left[ -J_2^{\tilde{C}} + 2 J_3^{\tilde{C}} + J_6^{\tilde{C}} - 2 J_7^{\tilde{C}} + 2 J_{11}^{\tilde{C}} - 2 J_{20}^{\tilde{C}} - 6 J_{25}^{\tilde{C}} + 2 J_{26}^{\tilde{C}} \right.$$
$$\left. + 2 J_{27}^{\tilde{C}} + 2 J_{28}^{\tilde{C}} \right] + \left( \dfrac{2s^2 + tm^2 + 2st}{2r_5} \right) J_{36}^{\tilde{C}},$$

$$J_{40}^{\tilde{C}} = \varepsilon^4 \left( \dfrac{-s}{\mu^2} \right) \left( \dfrac{m^2 - t}{\mu^2} \right) \left[ \left( \dfrac{-s-t}{\mu^2} \right) I_{111111100}^{\tilde{C}} + I_{1111111(-1)0}^{\tilde{C}} \right]$$
$$- \left( \dfrac{2st - m^2(s-t)}{2m^2(-s-t)} \right) J_{35}^{\tilde{C}} - \left( \dfrac{(2s+t)m^2}{2r_5} \right) J_{36}^{\tilde{C}} - \left( \dfrac{m^2 - t}{m^2 + s} \right) J_{38}^{\tilde{C}},$$

$$J_{41}^{\tilde{C}} = \varepsilon^4 \left( \dfrac{-s}{\mu^2} \right) \left[ I_{1111111(-2)0}^{\tilde{C}} - \left( \dfrac{-t}{\mu^2} \right) I_{11111110(-1)}^{\tilde{C}} \right.$$

$$+\left(\frac{m^2-t}{\mu^2}\right)I^{\tilde{C}}_{1111111(-1)0}-I^{\tilde{C}}_{1111111(-1)(-1)}\Bigg]$$

$$+\frac{1}{8}\left(\frac{t}{s}\right)\left(\frac{\mu^2}{-s-t}\right)\left[\left(\frac{-14s-8t}{\mu^2}\right)J^{\tilde{C}}_3+\left(\frac{3s+2t}{\mu^2}\right)J^{\tilde{C}}_4\right]$$

$$+\left(\frac{s}{t}\right)\left[-J^{\tilde{C}}_{27}+J^{\tilde{C}}_{29}\right]$$

$$+\left(\frac{t}{8s}\right)\left[2J^{\tilde{C}}_2+J^{\tilde{C}}_8+12J^{\tilde{C}}_9+4J^{\tilde{C}}_{11}+8J^{\tilde{C}}_{18}-4J^{\tilde{C}}_{24}+12J^{\tilde{C}}_{28}-4J^{\tilde{C}}_{32}\right]$$

$$+\left(\frac{t}{24(s+t)}\right)\left[J^{\tilde{C}}_5+3J^{\tilde{C}}_6-18J^{\tilde{C}}_7+36J^{\tilde{C}}_{15}-12J^{\tilde{C}}_{16}-12J^{\tilde{C}}_{25}-36J^{\tilde{C}}_{30}\right.$$

$$\left.+12J^{\tilde{C}}_{31}+12J^{\tilde{C}}_{35}\right]+\left(\frac{m^2t}{2r_5}\right)J^{\tilde{C}}_{36}+\left(\frac{2(m^2-t)}{m^2+s}+\frac{3t}{2s}\right)J^{\tilde{C}}_{38}.$$

### 3.2.4 Topology $\tilde{D}$

Sector 41: $\quad J^{\tilde{D}}_1=\varepsilon^2\left(\frac{m^2}{\mu^2}\right)\mathbf{D}^-I^{\tilde{D}}_{100101000},$

Sector 42: $\quad J^{\tilde{D}}_2=\varepsilon^2\left(\frac{-s}{\mu^2}\right)\mathbf{D}^-I^{\tilde{D}}_{010101000},$

Sector 49: $\quad J^{\tilde{D}}_3=\varepsilon^2\left(\frac{m^2+s+t}{\mu^2}\right)\mathbf{D}^-I^{\tilde{D}}_{100011000},$

$$J^{\tilde{D}}_4=\varepsilon^2\left[\left(\frac{m^2}{\mu^2}\right)\mathbf{D}^-I^{\tilde{D}}_{100011000}+\mathbf{D}^-I^{\tilde{D}}_{10001100(-1)}\right],$$

Sector 73: $\quad J^{\tilde{D}}_5=\varepsilon^2\left(\frac{m^2-t}{\mu^2}\right)\mathbf{D}^-I^{\tilde{D}}_{100100100}$

$$J^{\tilde{D}}_6=\varepsilon^2\left(\frac{-t}{\mu^2}\right)I^{\tilde{D}}_{200100200},$$

Sector 43: $\quad J^{\tilde{D}}_7=\varepsilon^3\left(\frac{-s}{\mu^2}\right)I^{\tilde{D}}_{110102000},$

Sector 54: $\quad J^{\tilde{D}}_8=\varepsilon^3\left(\frac{-s}{\mu^2}\right)I^{\tilde{D}}_{011021000},$

Sector 55: $\quad J^{\tilde{D}}_9=\varepsilon^3(-1+2\varepsilon)\left(\frac{-s}{\mu^2}\right)I^{\tilde{D}}_{111011000},$

$$J^{\tilde{D}}_{10}=\varepsilon^3\left(\frac{-s}{\mu^2}\right)\left(\frac{m^2+s+t}{\mu^2}\right)I^{\tilde{D}}_{111021000},$$

Sector 59: $\quad J^{\tilde{D}}_{11}=\varepsilon^4\left(\frac{-t}{\mu^2}\right)I^{\tilde{D}}_{110111000},$

$$J^{\tilde{D}}_{12}=\varepsilon^3\left(\frac{-t}{\mu^2}\right)\left(\frac{m^2}{\mu^2}\right)I^{\tilde{D}}_{210111000},$$

Sector 79: $\quad J_{13}^{\tilde{D}} = \varepsilon^3 \left( \dfrac{-s}{\mu^2} \right) \left( \dfrac{m^2 - t}{\mu^2} \right) I_{111200100}^{\tilde{D}} - J_6^{\tilde{D}},$

$$J_{14}^{\tilde{D}} = \varepsilon^3 \left( \dfrac{-s}{\mu^2} \right) \left[ \left( \dfrac{m^2 - t}{\mu^2} \right) I_{111200100}^{\tilde{D}} - I_{1112001(-1)0}^{\tilde{D}} \right] + \left( \dfrac{s}{2t} \right) J_6^{\tilde{D}},$$

Sector 93: $\quad J_{15}^{\tilde{D}} = \varepsilon^4 \left( \dfrac{-s-t}{\mu^2} \right) I_{101110100}^{\tilde{D}},$

$$J_{16}^{\tilde{D}} = \varepsilon^3 \left( \dfrac{-s-t}{\mu^2} \right) \left( \dfrac{m^2}{\mu^2} \right) I_{201110100}^{\tilde{D}},$$

Sector 121: $\quad J_{17}^{\tilde{D}} = \varepsilon^4 \left( \dfrac{-s}{\mu^2} \right) I_{100111100}^{\tilde{D}},$

$$J_{18}^{\tilde{D}} = \varepsilon^3 \left( \dfrac{s(m^2 - t) - t^2}{\mu^4} \right) I_{200111100}^{\tilde{D}},$$

Sector 123: $\quad J_{19}^{\tilde{D}} = \varepsilon^4 \left( \dfrac{-s}{\mu^2} \right) \left( \dfrac{m^2}{\mu^2} \right) I_{110111100}^{\tilde{D}},$

Sector 126: $\quad J_{20}^{\tilde{D}} = \varepsilon^4 \left( \dfrac{s^2}{\mu^4} \right) I_{011111100}^{\tilde{D}},$

Sector 127: $\quad J_{21}^{\tilde{D}} = \varepsilon^4 \left( \dfrac{-s}{\mu^2} \right) \left( \dfrac{m^2 + s + t}{\mu^2} \right) \left[ \left( \dfrac{m^2 - t}{\mu^2} \right) I_{111111100}^{\tilde{D}} + I_{1111111(-1)0}^{\tilde{D}} \right]$

$$\qquad - \left( \dfrac{-s-t}{m^2} \right) J_{19}^{\tilde{D}} - \left( \dfrac{m^2 + t}{-s} \right) J_{20}^{\tilde{D}},$$

$$J_{22}^{\tilde{D}} = \varepsilon^4 \left( \dfrac{-s}{\mu^2} \right) \left( \dfrac{m^2 - t}{\mu^2} \right) \left[ \left( \dfrac{m^2 - s - t}{\mu^2} \right) I_{111111100}^{\tilde{D}} + I_{1111111(-1)0}^{\tilde{D}} \right]$$

$$\qquad - \left( \dfrac{-t}{m^2} \right) J_{19}^{\tilde{D}} - \left( \dfrac{m^2 - t}{-s} \right) J_{20}^{\tilde{D}},$$

$$J_{23}^{\tilde{D}} = \varepsilon^4 \left( \dfrac{-s}{\mu^2} \right) \left[ I_{1111111(-2)0}^{\tilde{D}} + \left( \dfrac{3m^2 - t}{\mu^2} \right) I_{1111111(-1)0}^{\tilde{D}} \right.$$

$$\qquad \left. + 2 \left( \dfrac{m^2}{\mu^2} \right) \left( \dfrac{m^2 - t}{\mu^2} \right) I_{111111100}^{\tilde{D}} \right]$$

$$\qquad - \left( \dfrac{\mu^2}{-t} \right) \left( \dfrac{\mu^2}{-s-t} \right) \left[ \left( \dfrac{s^2 + st + t^2}{72\mu^4} \right) \left( 36 J_1^{\tilde{D}} - J_2^{\tilde{D}} + 48 J_7^{\tilde{D}} + 12 J_{19}^{\tilde{D}} \right) \right.$$

$$\qquad - \left( \dfrac{-s}{8\mu^2} \right) \left( \dfrac{2t - s}{\mu^2} \right) \left( J_4^{\tilde{D}} + 4 J_{11}^{\tilde{D}} \right)$$

$$\qquad + \left( \dfrac{-s}{8\mu^2} \right) \left( \dfrac{3s + 2t}{\mu^2} \right) \left( J_5^{\tilde{D}} - 2 J_6^{\tilde{D}} - 4 J_{15}^{\tilde{D}} \right)$$

$$\qquad - \left( \dfrac{s^2 + st - 5t^2}{6\mu^4} \right) J_{12}^{\tilde{D}} + \left( \dfrac{-s}{6\mu^2} \right) \left( \dfrac{5s + 6t}{\mu^2} \right) J_{16}^{\tilde{D}}$$

$$-\left(\frac{s^2}{6\mu^4}\right)(3J_{17}^{\check{D}}+2J_{18}^{\check{D}})\Bigg]-\left(\frac{-s-t}{m^2}\right)J_{19}^{\check{D}}-\left(\frac{2m^2+t}{-s}\right)J_{20}^{\check{D}}.$$

### 3.2.5 Topology $\tilde{E}$

Sector 42: $\quad J_1^{\tilde{E}}=\varepsilon^2\left(\frac{-s}{\mu^2}\right)\mathbf{D}^-I_{010101000}^{\tilde{E}},$

Sector 49: $\quad J_2^{\tilde{E}}=\varepsilon^2\left(\frac{-s-t}{\mu^2}\right)\mathbf{D}^-I_{100011000}^{\tilde{E}},$

Sector 73: $\quad J_3^{\tilde{E}}=\varepsilon^2\left(\frac{-t}{\mu^2}\right)\mathbf{D}^-I_{100100100}^{\tilde{E}},$

Sector 54: $\quad J_4^{\tilde{E}}=\varepsilon^3\left(\frac{-s}{\mu^2}\right)I_{011021000}^{\tilde{E}},$

Sector 55: $\quad J_5^{\tilde{E}}=\varepsilon^3\left(\frac{-s}{\mu^2}\right)\left(\frac{-s-t}{\mu^2}\right)I_{111021000}^{\tilde{E}},$

Sector 59: $\quad J_6^{\tilde{E}}=\varepsilon^4\left(\frac{-t}{\mu^2}\right)I_{110111000}^{\tilde{E}},$

Sector 79: $\quad J_7^{\tilde{E}}=\varepsilon^3\left(\frac{-s}{\mu^2}\right)\left(\frac{-t}{\mu^2}\right)I_{111200100}^{\tilde{E}},$

Sector 93: $\quad J_8^{\tilde{E}}=\varepsilon^4\left(\frac{-s-t}{\mu^2}\right)I_{101110100}^{\tilde{E}},$

Sector 121: $\quad J_9^{\tilde{E}}=\varepsilon^4\left(\frac{-s}{\mu^2}\right)I_{100111100}^{\tilde{E}},$

Sector 126: $\quad J_{10}^{\tilde{E}}=\varepsilon^4\left(\frac{s^2}{\mu^4}\right)I_{011111100}^{\tilde{E}},$

Sector 127: $\quad J_{11}^{\tilde{E}}=\varepsilon^4\left(\frac{s^2}{\mu^2}\right)I_{11111110(-1)}^{\tilde{E}}$

$$+\frac{1}{4(1+4\varepsilon)}\left(\frac{\mu^2}{t}\right)\Bigg[2\left(\frac{-t}{\mu^2}\right)\left((1+3\varepsilon)J_2^{\tilde{E}}-\varepsilon J_3^{\tilde{E}}+12\varepsilon J_6^{\tilde{E}}+12\varepsilon J_8^{\tilde{E}}\right)$$

$$+\left(\frac{s}{\mu^2}\right)\left((1+6\varepsilon)J_1^{\tilde{E}}+J_3^{\tilde{E}}+4J_7^{\tilde{E}}-12J_8^{\tilde{E}}\right.$$

$$+2\varepsilon\left[J_2^{\tilde{E}}+2J_3^{\tilde{E}}+8J_7^{\tilde{E}}-36J_8^{\tilde{E}}-12J_9^{\tilde{E}}\right]\Bigg)\Bigg],$$

$$J_{12}^{\tilde{E}}=\varepsilon^4\left(\frac{-s}{\mu^2}\right)\left(\frac{-s-t}{\mu^2}\right)\left[\left(\frac{-t}{\mu^2}\right)I_{111111100}^{\tilde{E}}+I_{1111111(-1)0}^{\tilde{E}}\right]-\frac{t}{s}J_{10}^{\tilde{E}}$$

$$+\frac{1}{1+4\varepsilon}\left(\frac{\mu^2}{t}\right)\Bigg[\frac{\varepsilon}{2}\left(\frac{s}{\mu^2}\right)(J_1^{\tilde{E}}+J_2^{\tilde{E}})$$

$$+\left(\frac{t}{\mu^2}\right)\left(-\frac{4+15\varepsilon}{2}J_2^{\tilde{E}}+\frac{1+6\varepsilon}{4}J_3^{\tilde{E}}-6\varepsilon J_6^{\tilde{E}}-6\varepsilon J_8^{\tilde{E}}\right)$$

$$-6\varepsilon\left(\frac{s}{\mu^2}\right)\left(J_8^{\tilde{E}}+J_9^{\tilde{E}}\right)\Bigg].$$

# 4 Integration of the differential equation

In this section we provide details on the integration of the differential equation. We discuss the alphabets for the various topologies, i.e. the set of differential one-forms appearing in the differential equation. We express the results for the topologies $B,C,D,E,\tilde{C},\tilde{D}$ and $\tilde{E}$ in terms of multiple polylogarithms. This requires the rationalisation of square roots, where the methods of refs. [97, 98] are used. The more complicated topologies $A,\tilde{A}$ and $\tilde{B}$ are expressed in terms of iterated integrals. We choose a common boundary point for all topologies. Furthermore we discuss the path of integration for the various topologies.

## 4.1 The alphabet

We may write the matrix $A$ appearing in eq. (15) as

$$A = \sum_{j=1}^{N_L} M_j \omega_j, \tag{40}$$

where the $M_j$'s are $(N_F \times N_F)$-matrices with constant entries and the $\omega_j$'s differential one-forms. We may divide the latter into dlog-forms with rational arguments, dlog-forms with algebraic (and non-rational) arguments and one-forms related to elliptic curves. For the dlog-forms with rational arguments we write

$$\omega_j = d\ln e_j. \tag{41}$$

The rational letters are

$$e_1 = \frac{-s}{\mu^2}, \qquad e_2 = \frac{-t}{\mu^2}, \qquad e_3 = \frac{-s-t}{\mu^2},$$

$$e_4 = \frac{m^2}{\mu^2}, \qquad e_5 = \frac{m^2+s}{\mu^2}, \qquad e_6 = \frac{m^2-t}{\mu^2},$$

$$e_7 = \frac{m^2-s}{\mu^2}, \qquad e_8 = \frac{m^2(s+t)-st}{\mu^4}, \qquad e_9 = \frac{4m^2+s}{\mu^2}$$

$$e_{10} = \frac{4m^2-t}{\mu^2}, \qquad e_{11} = \frac{4m^2s^2-\left(m^2-s\right)^2 t}{\mu^6}, \qquad e_{12} = \frac{4m^2\left(m^2+s\right)-st}{\mu^4},$$

$$e_{13} = \frac{m^2+s+t}{\mu^2}, \qquad e_{14} = \frac{m^2-s-t}{\mu^2}, \qquad e_{15} = \frac{m^2s-t(s+t)}{\mu^2},$$

$$e_{16} = \frac{4s(s+t)+m^2t}{\mu^2}, \qquad e_{17} = \frac{m^2t-s(s+t)}{\mu^2}, \qquad e_{18} = \frac{4m^2(s+t)^2-t(m^2+s+t)^2}{\mu^2},$$

$$e_{19} = \frac{4m^4(-t)+s(s+t)^2}{\mu^2}, \qquad e_{20} = \frac{4m^2(m^2-s-t)+t(s+t)}{\mu^2}.$$

For the dlog-forms with algebraic arguments we write

$$\omega_j = d\ln o_j. \tag{42}$$

The algebraic letters involve the square roots $r_1$-$r_8$ and read

$$o_{21} = \frac{2m^2 - t - r_1}{2m^2 - t + r_1},$$
$$o_{22} = \frac{2m^2 + s - r_2}{2m^2 + s + r_2},$$

$$o_{23} = \frac{2m^2(t-s) + st - r_1 r_2}{2m^2(t-s) + st + r_1 r_2},$$
$$o_{24} = \frac{(m^2 - s)t - r_3}{(m^2 - s)t + r_3},$$

$$o_{25} = \frac{(m^2 + s)t - r_3}{(m^2 + s)t + r_3},$$
$$o_{26} = \frac{st^2 - m^2 t(4s+t) - r_1 r_3}{st^2 - m^2 t(4s+t) + r_1 r_3},$$

$$o_{27} = \frac{s(2m^2 - t) - r_4}{s(2m^2 - t) + r_4},$$
$$o_{28} = \frac{s\left(2m^2 + 2s + t\right) - r_4}{s\left(2m^2 + 2s + t\right) + r_4},$$

$$o_{29} = \frac{t\left[2m^2\left(m^2 + 2s\right) - st\right] - r_1 r_4}{t\left[2m^2\left(m^2 + 2s\right) - st\right] + r_1 r_4},$$
$$o_{30} = \frac{st\left[s\left(4m^2 - t\right) + m^2\left(2m^2 + t\right)\right] - r_3 r_4}{st\left[s\left(4m^2 - t\right) + m^2\left(2m^2 + t\right)\right] + r_3 r_4},$$

$$o_{31} = \frac{m^2 t + 2st - r_5}{m^2 t + 2st + r_5},$$
$$o_{32} = \frac{m^2 t - 2t(s+t) - r_5}{m^2 t - 2t(s+t) + r_5},$$

$$o_{33} = \frac{2m^2\left(m^2 - s - t\right) + t(s+t) - ir_6}{2m^2\left(m^2 - s - t\right) + t(s+t) + ir_6},$$
$$o_{34} = \frac{\left(t - 2m^2\right)(s+t) - ir_6}{\left(t - 2m^2\right)(s+t) + ir_6},$$

$$o_{35} = \frac{t\left(4m^2 - t\right)(s+t) - 2m^4 t - ir_1 r_6}{t\left(4m^2 - t\right)(s+t) - 2m^4 t + ir_1 r_6},$$
$$o_{36} = \frac{-t\left(m^2 + s + t\right) - r_7}{-t\left(m^2 + s + t\right) + r_7},$$

$$o_{37} = \frac{-t\left(m^2 - s - t\right) - r_7}{-t\left(m^2 - s - t\right) + r_7},$$
$$o_{38} = \frac{-t\left[-s\left(4m^2 - t\right) - t\left(3m^2 - t\right)\right] - r_1 r_7}{-t\left[-s\left(4m^2 - t\right) - t\left(3m^2 - t\right)\right] + r_1 r_7},$$

$$o_{39} = \frac{s(s+t) - r_8}{s(s+t) + r_8},$$
$$o_{40} = \frac{-s\left(2m^2 - s - t\right) - r_8}{-s\left(2m^2 - s - t\right) + r_8},$$

$$o_{41} = \frac{-s\left(2m^2 + s + t\right) - r_8}{-s\left(2m^2 + s + t\right) + r_8}.$$

We denote the set of differential one-forms related to the two elliptic curves of topology $\tilde{A}$ by $H^{\tilde{A}}$, the corresponding set of differential one-forms related to the elliptic curve of topology $\tilde{B}$ by $H^{\tilde{B}}$. The explicit expressions for the one-forms related to elliptic curves are rather lengthy and we list them in the supplementary electronic file attached to the arxiv version of this article, see appendix B.

Not every differential one-form occurs in every topology. We call the set of differential one-forms occurring in the differential equation for a particular topology the alphabet for this topology. The alphabets for the various topologies are

$$\mathcal{A}^A = \{\omega_1, \omega_2, \omega_3, \omega_4, \omega_5, \omega_6, \omega_7, \omega_9, \omega_{10}, \omega_{11}, \omega_{12}, \omega_{21}, \omega_{22}, \omega_{23}, \omega_{24}, \omega_{25}, \omega_{26}, \omega_{27}, \omega_{28},$$
$$\omega_{29}, \omega_{30}\},$$

$$
\begin{aligned}
\mathcal{A}^B &= \{\omega_1, \omega_2, \omega_3, \omega_4, \omega_5, \omega_6, \omega_{10}, \omega_{12}, \omega_{21}, \omega_{27}, \omega_{28}, \omega_{29}\}, \\
\mathcal{A}^C &= \{\omega_1, \omega_2, \omega_3, \omega_4, \omega_5, \omega_6, \omega_7, \omega_8\}, \\
\mathcal{A}^D &= \{\omega_1, \omega_2, \omega_3, \omega_4, \omega_5, \omega_6\}, \\
\mathcal{A}^E &= \{\omega_1, \omega_2, \omega_3\}, \\
\mathcal{A}^{\tilde{A}} &= \{\omega_1, \omega_2, \omega_3, \omega_4, \omega_5, \omega_6, \omega_7, \omega_9, \omega_{10}, \omega_{11}, \omega_{14}, \omega_{20}, \omega_{21}, \omega_{22}, \omega_{23}, \omega_{24}, \omega_{25}, \omega_{26}, \\
&\quad \omega_{33}, \omega_{34}, \omega_{35}\} \cup H^{\tilde{A}}, \\
\mathcal{A}^{\tilde{B}} &= \{\omega_1, \omega_2, \omega_3, \omega_4, \omega_5, \omega_6, \omega_7, \omega_{10}, \omega_{11}, \omega_{12}, \omega_{13}, \omega_{14}, \omega_{17}, \omega_{18}, \omega_{19}, \omega_{21}, \omega_{24}, \omega_{25}, \omega_{26}, \\
&\quad \omega_{27}, \omega_{28}, \omega_{29}, \omega_{30}, \omega_{36}, \omega_{37}, \omega_{38}, \omega_{39}, \omega_{40}, \omega_{41}\} \cup H^{\tilde{B}}, \\
\mathcal{A}^{\tilde{C}} &= \{\omega_1, \omega_2, \omega_3, \omega_4, \omega_5, \omega_6, \omega_7, \omega_8, \omega_{14}, \omega_{16}, \omega_{31}, \omega_{32}\}, \\
\mathcal{A}^{\tilde{D}} &= \{\omega_1, \omega_2, \omega_3, \omega_4, \omega_5, \omega_6, \omega_{13}, \omega_{15}\}, \\
\mathcal{A}^{\tilde{E}} &= \{\omega_1, \omega_2, \omega_3\}.
\end{aligned}
\tag{43}
$$

## 4.2 Boundary values and integration path

For the Moller experiment we are interested in the region (see eq. (44))

$$
-t \;\lesssim\; s \;\ll\; m^2.
\tag{44}
$$

We choose the boundary point and the integration path such that the CPU time required to evaluate the master integrals is optimised for this specific region. As boundary point we choose

$$
t = 0, \quad m^2 = \infty.
\tag{45}
$$

Without loss of generality we set $\mu^2 = s$. The Feynman integrals depend then only on dimensionless kinematic variables, which we may take initially as

$$
x_t = \frac{-t}{s}, \quad x_{m^2}^{-1} = \frac{s}{m^2}.
\tag{46}
$$

Our chosen boundary point corresponds to

$$
\left(x_t, x_{m^2}^{-1}\right) = (0,0).
\tag{47}
$$

At this boundary point the boundary constants are of uniform weight and spanned as a vector space in the lowest weights by

$$
\begin{aligned}
\text{weight } 0 \;&:\; 1, \\
\text{weight } 1 \;&:\; i\pi, \\
\text{weight } 2 \;&:\; \zeta_2, \\
\text{weight } 3 \;&:\; \zeta_3, i\pi\zeta_2, \\
\text{weight } 4 \;&:\; \zeta_4, i\pi\zeta_3.
\end{aligned}
\tag{48}
$$

The non-zero boundary constants of the simpler master integrals are easily calculated. The following two constraints provide an efficient way to determine the boundary constants of the more complicated master integrals: First of all, some of the master integrals must vanish on the hypersurface $m^2 = \infty$ due to power counting in the variable $m^2$. Secondly, we also note that topologies $A$, $B$, $D$, $\tilde{B}$ and $\tilde{D}$ do not depend on the hypersurface $m^2 = \infty$ on the Mandelstam variable $t$. This can also be used to determine some boundary constants. In addition we may always use the PSLQ algorithm [99–102] to determine the boundary values from high-precision numerical results, assuming that the boundary values are linear combinations of the constants in eq. (48) with rational coefficients. The explicit values of the boundary constants for all topologies are given in the supplementary electronic file attached to the arxiv version of this article.

Topologies $E$ and $\tilde{E}$ depend only on $x_t$ and we simply integrate the differential equation in the variable $x_t$. The result is expressed in terms of harmonic polylogarithms [103, 104].

Topologies $C$, $D$ and $\tilde{D}$ depend on both variables $x_t$ and $x_{m^2}^{-1}$. They do not involve any square roots nor any elliptic curves. We first integrate in $x_t$ at $x_{m^2}^{-1} = 0$, followed by an integration in $x_{m^2}^{-1}$ at $x_t = $ const. The result is expressed in terms of multiple polylogarithms [105, 106].

Topology $\tilde{C}$ involves the square root $r_5$. The square root $r_5$ is rationalised by

$$m^2 = -\frac{s\,(s+t)\,(1+\hat{z})^2}{t}\frac{}{\hat{z}}, \qquad r_5 = s\,(s+t)\frac{(1-\hat{z}^2)}{\hat{z}}. \tag{49}$$

The inverse transformation is given by

$$\hat{z} = -\frac{2s\,(s+t)+m^2 t+r_5}{2s\,(s+t)}. \tag{50}$$

The boundary point $(x_t, x_{m^2}^{-1}) = (0,0)$ corresponds to $(x_t, \hat{z}) = (0,0)$. We first integrate in $x_t$ at $\hat{z} = 0$, followed by an integration in $\hat{z}$ at $x_t = $ const. The result is again expressed in terms of multiple polylogarithms.

Topology $B$ involves the square root $r_1$ and $r_4$. These square roots occurred in ref. [79] and we may use the same rationalisation. The square root $r_1$ is rationalised by

$$t = -\frac{\tilde{y}^2}{1-\tilde{y}}m^2, \qquad r_1 = \frac{\tilde{y}\,(2-\tilde{y})}{1-\tilde{y}}m^2. \tag{51}$$

The inverse transformation is given by

$$\tilde{y} = \frac{t+r_1}{2m^2}. \tag{52}$$

The square root $r_4$ is rationalised by

$$m^2 = \frac{(1-2\tilde{z})\,(2-\tilde{y})^2}{4\tilde{z}^2\,(1-\tilde{y})}s, \qquad r_4 = \frac{(1-\tilde{z})\,(1-2\tilde{z})\,\tilde{y}\,(2-\tilde{y})^3}{4\tilde{z}^3\,(1-\tilde{y})^2}s^2. \tag{53}$$

The inverse transformation is given by

$$\tilde{z} = -\frac{(4m^2-t)}{4m^2}\left(\frac{s}{m^2}-\frac{r_4}{m^2 r_1}\right). \tag{54}$$

The boundary point $(x_t, x_{m^2}^{-1}) = (0,0)$ corresponds to $(\tilde{z}, \tilde{y}) = (0,0)$. We first integrate in $\tilde{z}$ at $\tilde{y} = 0$, followed by an integration in $\tilde{y}$ at $\tilde{z} = \text{const}$. The result is again expressed in terms of multiple polylogarithms.

Topology $A$ involves the four square roots $r_1$, $r_2$, $r_3$ and $r_4$. An efficient way to treat this case is to introduce the variables

$$w_t = \sqrt{x_t}, \qquad w_{m^2}^{-1} = \sqrt{x_{m^2}^{-1}}. \tag{55}$$

The boundary point $(x_t, x_{m^2}^{-1}) = (0,0)$ corresponds to $(w_t, w_{m^2}^{-1}) = (0,0)$. We first integrate in $w_t$ at $w_{m^2}^{-1} = 0$, the result of this integration is expressed in terms of multiple polylogarithms. We then integrate in $w_{m^2}^{-1}$ at $w_t = \text{const}$. In the result of this integration we first isolate all logarithms $\ln(w_{m^2}^{-1})$, the remainder is a regular function at $w_{m^2}^{-1} = 0$, which we evaluate as a Taylor series.

Topologies $\tilde{A}$ and $\tilde{B}$ involve elliptic curves. From a numerical point of view it is again most efficient to evaluate them in a similar way as topology $A$. In particular we use the same integration path as for topology $A$. At $m^2 = \infty$ (corresponding to $x_{m^2}^{-1} = w_{m^2}^{-1} = 0$) the master integrals of topology $\tilde{B}$ are independent of $x_t$, leaving just the integration in the variable $w_{m^2}^{-1}$. For topology $\tilde{A}$ we first integrate in $x_t$ at $w_{m^2}^{-1} = 0$, followed by an integration in the variable $w_{m^2}^{-1}$ at fixed $x_t$. The result of the integration in $x_t$ is expressed in terms of harmonic polylogarithms [104], for the integration in $w_{m^2}^{-1}$ we expand all differential one-forms in the variable $(w_{m^2}^{-1})^{25}$. Whereas in the case of topology $A$, the expansion of the differential one-forms can be done on the fly to any desired order, this approach is computationally more expensive for topologies $\tilde{A}$ and $\tilde{B}$ due to the differential one-forms depending on the elliptic curves. It is more efficient to pre-compute the expansions once and for all to a fixed order in the variable $w_{m^2}^{-1}$. In the numerical program we use an expansion up to order $(w_{m^2}^{-1})^{25}$.

# 5 Numerical results

In this section we give numerical results for all Feynman integrals at a specific kinematic point. As kinematic point we choose

$$s = 0.0112 \, \text{GeV}^2, \quad t = -2 \cdot 10^{-3} \, \text{GeV}^2, \quad m_Z^2 = 8.32 \cdot 10^3 \, \text{GeV}^2. \tag{56}$$

The Moller experiment has a beam energy of $E_{\text{beam}} = 11 \, \text{GeV}$ [1], the corresponding value of the Mandelstam variable $s$ is obtained with the help of

$$s = 2 m_e E_{\text{beam}},$$

where $m_e$ is the mass of the electron. For the value of $t$ (or correspondingly the scattering angle) we choose a generic value. In the supplementary electronic file attached to the arxiv version of this article we provide C++-programs which provide numerical evaluation routines for all master integrals of a given topology. The algorithms for the numerical evaluation of multiple polylogarithms are based on [107], topologies $A$, $\tilde{A}$ and $\tilde{B}$ use in addition the class `user_defined_kernel`

from ref. [60] for the integration in the variable $w_{m^2}^{-1}$. The numerical results for the first five terms of the $\varepsilon$-expansions of all master integrals evaluated at the point in eq. (56) are listed in the following. In addition, we compared our results to the results of the program AMFlow [108–110] and found perfect agreement. Our numerical evaluation routines are significantly faster than AMFlow and stable in the limit $m^2 \to \infty$.

| | $\varepsilon^0$ | $\varepsilon^1$ | $\varepsilon^2$ | $\varepsilon^3$ | $\varepsilon^4$ |
|---|---|---|---|---|---|
| $J_1^A$ | 1 | −27.036518 | 367.13159 | −3339.1035 | 22888.178 |
| $J_2^A$ | −2 | 27.036518 | −172.87372 | 696.44831 | −1969.4134 |
| | | −6.2831853i | +84.937727i | −563.76967i | +2467.3908i |
| $J_3^A$ | 1 | −27.036518 | 370.42146 | −3430.4542 | 24170.732 |
| $J_4^A$ | 1 | −27.036513 | 370.42132 | −3430.4524 | 24170.716 |
| $J_5^A$ | 1 | −27.036515 | 370.42139 | −3430.4533 | 24170.724 |
| $J_6^A$ | 0 | −0.00098058067 | 0.028472889 | −0.41695418 | 4.1082507 |
| $J_7^A$ | 0 | 0 | 2.4038461e − 07 | −6.7395475e − 06 | 9.5473101e − 05 |
| $J_8^A$ | 4 | 25.132741i | −85.536571 | −22.438396 | 366.90758 |
| | | | | −206.70851i | −140.9846i |
| $J_9^A$ | 0 | 0 | 0 | −2.0889958e − 05 | 0.00039240758 |
| | | | | −4.2290656e − 06i | +4.4482410e − 05i |
| $J_{10}^A$ | 0 | 0 | 1.9543798e − 05 | −0.00035735832 | 0.0034638444 |
| | | | +4.2290642e − 06i | −4.4482397e − 05i | +0.00023755484i |
| $J_{11}^A$ | 0 | 0 | −0.018004884 | 0.39305146 | −4.4568117 |
| | | | −0.0036449972i | +0.049274018i | −0.3270537i |
| $J_{12}^A$ | 0 | 0 | 0 | −1.1118058e − 05 | 0.00022465136 |
| | | | | −2.1145333e − 06i | +2.4355742e − 05i |
| $J_{13}^A$ | 0 | 0 | 0 | 3.9541684e − 07 | −1.0584234e − 05 |
| $J_{14}^A$ | 0 | 2.0192321e − 06 | −4.8535306e − 05 | 0.00058659725 | −0.0047460836 |
| $J_{15}^A$ | 0 | 0 | 0 | −2.2143346e − 06 | 5.9271717e − 05 |
| $J_{16}^A$ | 0 | 0 | −1.9543798e − 05 | 0.00037381959 | −0.0037377082 |
| | | | −4.2290642e − 06i | +4.8711457e − 05i | −0.00028203719i |
| $J_{17}^A$ | 0 | 0 | 0 | 1.9543803e − 05 | −0.00035735841 |
| | | | | +4.2290656e − 06i | −4.4482410e − 05i |
| $J_{18}^A$ | 0 | 0 | 0 | 0.00027353844 | −0.0038636584 |
| | | | | +0.0001227973i | −0.00068877199i |
| $J_{19}^A$ | 0 | 0 | −2.4038461e − 07 | 1.3886870e − 05 | −0.00023989182 |
| | | | | +1.3593428e − 06i | −1.5657262e − 05i |
| $J_{20}^A$ | 0 | 0.00049029033 | −0.0066278718 | 0.04237916 | −0.17073098 |
| | | | +0.0015402925i | −0.020822073i | +0.13820541i |
| $J_{21}^A$ | 0 | 0 | 0 | −2.4038461e − 07 | 6.3441307e − 06 |
| $J_{22}^A$ | 0 | 0 | 0 | 1.1057694e − 06 | −2.9183006e − 05 |
| $J_{23}^A$ | 0 | 0 | −1.2019401e − 07 | 2.8890454e − 06 | −3.5048783e − 05 |
| $J_{24}^A$ | 0 | 0 | 1.1658658e − 06 | −2.8415975e − 05 | 0.00034964705 |
| $J_{25}^A$ | 0 | 0 | 0 | 0 | −2.1367936e − 05 |
| | | | | | −4.2290668e − 06i |
| $J_{26}^A$ | 0 | 0 | 1.9543798e − 05 | −0.00012290746 | 0.00031490245 |
| | | | +4.2290642e − 06i | +6.9856779e − 05i | −0.0003622524i |

| | $\varepsilon^0$ | $\varepsilon^1$ | $\varepsilon^2$ | $\varepsilon^3$ | $\varepsilon^4$ |
|---|---|---|---|---|---|
| $J_{27}^A$ | 0 | 0 | 0 | $-1.8675616e-05$ $-4.2290638e-06i$ | $0.00033313574$ $+4.4482390e-05i$ |
| $J_{28}^A$ | 0 | 0 | $-1.5379816e-05$ $-3.5742116e-06i$ | $0.00028778079$ $+4.1168700e-05i$ | $-0.0028072667$ $-0.00023836498i$ |
| $J_{29}^A$ | 0 | 0 | $3.9087595e-05$ $+8.4581284e-06i$ | $-0.00075649656$ $-9.7422925e-05i$ | $0.007712504$ $+0.00056407449i$ |
| $J_{30}^A$ | 0 | 0 | $2.4038461e-07$ | $-3.0091969e-06$ $+7.5519053e-07i$ | $1.7768897e-05$ $-9.4536706e-06i$ |
| $J_{31}^A$ | 0 | 0 | 0 | 0 | $3.3946007e-10$ $+1.5961098e-10i$ |
| $J_{32}^A$ | 0 | 0 | $1.5379816e-05$ $+3.5742116e-06i$ | $-8.9633670e-05$ $+5.5465536e-05i$ | $0.00018204374$ $-0.00026141618i$ |
| $J_{33}^A$ | 0 | 0 | $-3.9087595e-05$ $-8.4581284e-06i$ | $0.00024581498$ $-0.00013971355i$ | $-0.00062980573$ $+0.00072450468i$ |
| $J_{34}^A$ | 0 | 0 | 0 | $-3.6395284e-05$ $-8.4581276e-06i$ | $0.00064366285$ $+8.8965043e-05i$ |
| $J_{35}^A$ | 0 | 0 | $-4.8076922e-07$ | $-1.4871575e-05$ $-5.7394495e-06i$ | $0.0004000839$ $+7.1847942e-05i$ |

Table 3: Numerical results for the first five terms of the $\varepsilon$-expansion of the master integrals $J_1^A$-$J_{35}^A$ for the kinematic point specified by eq. (56).

| | $\varepsilon^0$ | $\varepsilon^1$ | $\varepsilon^2$ | $\varepsilon^3$ | $\varepsilon^4$ |
|---|---|---|---|---|---|
| $J_1^B$ | 1 | $-27.036518$ | $367.13159$ | $-3339.1035$ | $22888.178$ |
| $J_2^B$ | 1 | $-27.036518$ | $370.42146$ | $-3430.4542$ | $24170.732$ |
| $J_3^B$ | 3 | $18.849556i$ | $-64.152429$ | $-38.465821$ $-155.03138i$ | $245.95795$ $-241.68788i$ |
| $J_4^B$ | $-2$ | $27.036518$ $-6.2831853i$ | $-172.87372$ $+84.937727i$ | $696.44831$ $-563.76967i$ | $-1969.4134$ $+2467.3908i$ |
| $J_5^B$ | 0 | $-0.00098058067$ | $0.028472648$ | $-0.41694744$ | $4.1081553$ |
| $J_6^B$ | 0 | 0 | $2.4038461e-07$ | $-6.7395475e-06$ | $9.5473101e-05$ |
| $J_7^B$ | 0 | 0 | $1.9543798e-05$ $+4.2290642e-06i$ | $-0.00024013289$ $+1.2687191e-05i$ | $0.0020435243$ $+3.2789125e-06i$ |
| $J_8^B$ | 4 | $25.132741i$ | $-85.536571$ | $-22.438396$ $-206.70851i$ | $366.90758$ $-140.9846i$ |
| $J_9^B$ | 0 | 0 | $1.9543798e-05$ $+4.2290642e-06i$ | $-0.00035735832$ $-4.4482397e-05i$ | $0.0034638444$ $+0.00023755484i$ |
| $J_{10}^B$ | 0 | 0 | 0 | $-2.0889958e-05$ $-4.2290656e-06i$ | $0.00039240758$ $+4.4482410e-05i$ |
| $J_{11}^B$ | 0 | 0 | $-1.9543798e-05$ $-4.2290642e-06i$ | $0.00037381959$ $+4.8711457e-05i$ | $-0.0037377082$ $-0.00028203719i$ |
| $J_{12}^B$ | 0 | 0 | 0 | $3.9541684e-07$ | $-1.0584234e-05$ |
| $J_{13}^B$ | 0 | 0 | $1.9543798e-05$ $+4.2290642e-06i$ | $-0.00012290746$ $+6.9856779e-05i$ | $0.00031490245$ $-0.0003622524i$ |

| | | | | | |
|---|---|---|---|---|---|
| $J_{14}^B$ | 0 | 0 | 0 | 0 | $-0.00015900538$ $-6.5627738e-05i$ |
| $J_{15}^B$ | 0 | 0 | 0 | $-1.8675616e-05$ $-4.2290638e-06i$ | $0.00033313574$ $+4.4482390e-05i$ |
| $J_{16}^B$ | 0 | 0 | 0 | $-2.3104304e-05$ $-4.2290685e-06i$ | $0.0004516795$ $+4.4482440e-05i$ |
| $J_{17}^B$ | 0 | 0 | $-1.5379816e-05$ $-3.5742116e-06i$ | $0.00028778079$ $+4.1168700e-05i$ | $-0.0028072667$ $-0.00023836498i$ |
| $J_{18}^B$ | 0 | 0 | 0 | 0 | $7.9083368e-07$ |
| $J_{19}^B$ | 0 | 0 | 0 | 0 | $-3.6378453e-06$ $-2.3381840e-12i$ |
| $J_{20}^B$ | 0 | 0 | $-1.5379816e-05$ $-3.5742116e-06i$ | $0.00017292344$ $-1.0722632e-05i$ | $-0.0013253259$ $-2.7711815e-06i$ |
| $J_{21}^B$ | 0 | 0 | $-2.4038461e-07$ | $8.5584749e-06$ $+2.3381830e-12i$ | $-0.00014416067$ $+1.0521821e-11i$ |
| $J_{22}^B$ | 0 | 0 | $3.9087595e-05$ $+8.4581284e-06i$ | $-0.00048248013$ $+2.5374379e-05i$ | $0.0041463203$ $+6.5578122e-06i$ |
| $J_{23}^B$ | 0 | 0 | 0 | 0 | $3.6407504e-10$ $+1.6530395e-10i$ |
| $J_{24}^B$ | 0 | 0 | 0 | 0 | $8.2058873e-06$ $+1.4933021e-10i$ |
| $J_{25}^B$ | 0 | 0 | 0 | 0 | $0.00028973495$ $+0.00013125521i$ |
| $J_{26}^B$ | 2 | $12.566371i$ | $-42.768286$ | $-25.643881$ $-103.35426i$ | $163.9723$ $-161.12512i$ |

Table 4: Numerical results for the first five terms of the ε-expansion of the master integrals $J_1^B$-$J_{26}^B$ for the kinematic point specified by eq. (56).

| | $\varepsilon^0$ | $\varepsilon^1$ | $\varepsilon^2$ | $\varepsilon^3$ | $\varepsilon^4$ |
|---|---|---|---|---|---|
| $J_1^C$ | $-2$ | $27.036518$ $-6.2831853i$ | $-172.87372$ $+84.937727i$ | $696.44831$ $-563.76967i$ | $-1969.4134$ $+2467.3908i$ |
| $J_2^C$ | 1 | $-27.036518$ | $370.42146$ | $-3430.4542$ | $24170.732$ |
| $J_3^C$ | 1 | $-27.036513$ | $370.42132$ | $-3430.4524$ | $24170.716$ |
| $J_4^C$ | 1 | $-27.036515$ | $370.42139$ | $-3430.4533$ | $24170.724$ |
| $J_5^C$ | 1 | $-27.036519$ | $370.42148$ | $-3430.4545$ | $24170.735$ |
| $J_6^C$ | 0 | $-2.4038459e-07$ | $6.2587777e-06$ | $-8.2784834e-05$ | $0.00074184352$ |
| $J_7^C$ | 4 | $25.132741i$ | $-85.536571$ | $-22.438396$ $-206.70851i$ | $366.90758$ $-140.9846i$ |
| $J_8^C$ | 0 | 0 | $1.9543798e-05$ $+4.2290642e-06i$ | $-0.00035735832$ $-4.4482397e-05i$ | $0.0034638444$ $+0.00023755484i$ |
| $J_9^C$ | 0 | 0 | 0 | $-2.0889958e-05$ $-4.2290656e-06i$ | $0.00039240758$ $+4.4482410e-05i$ |
| $J_{10}^C$ | 0 | 0 | 0 | $1.9543803e-05$ | $-0.00035735841$ |

| | $\varepsilon^0$ | $\varepsilon^1$ | $\varepsilon^2$ | $\varepsilon^3$ | $\varepsilon^4$ |
|---|---|---|---|---|---|
| | | | | $+4.2290656e-06i$ | $-4.4482410e-05i$ |
| $J_{11}^C$ | 0 | 0 | 0 | $0.00027353844$ $+0.0001227973i$ | $-0.0038636584$ $-0.00068877199i$ |
| $J_{12}^C$ | 0 | $2.4038459e-07$ | $-3.0091966e-06$ $+7.5519045e-07i$ | $1.7768896e-05$ $-9.4536695e-06i$ | $-6.5938874e-05$ $+5.8307098e-05i$ |
| $J_{13}^C$ | 0 | 0 | $4.8076920e-07$ | $-4.6836770e-05$ $-6.9477502e-06i$ | $0.00081023924$ $+7.3078241e-05i$ |
| $J_{14}^C$ | 0 | 0 | $2.4038460e-07$ | $-6.0183934e-06$ | $7.6526059e-05$ |
| $J_{15}^C$ | 0 | 0 | $-1.1057695e-06$ | $2.7684618e-05$ | $-0.00035201996$ |
| $J_{16}^C$ | 0 | $-1.1057702e-06$ | $2.5473093e-05$ | $-0.00029665089$ | $0.002327632$ |
| $J_{17}^C$ | 1 | $-9.6153835e-07$ $+6.2831853i$ | $-21.384104$ $+2.4166048e-06i$ | $-5.6098211$ $-51.676989i$ | $91.727546$ $-35.246805i$ |
| $J_{18}^C$ | 0 | $4.8076917e-07$ | $-3.8126057e-05$ $-5.4373666e-06i$ | $0.00023745723$ $-0.00013367202i$ | $-0.00064921824$ $+0.00071174302i$ |
| $J_{19}^C$ | 0 | $4.8076917e-07$ | $-5.5376244e-06$ $+1.5103809e-06i$ | $-4.0714338e-05$ $-3.2802839e-05i$ | $0.0012930442$ $+0.00030662686i$ |

Table 5: Numerical results for the first five terms of the $\varepsilon$-expansion of the master integrals $J_1^C$-$J_{19}^C$ for the kinematic point specified by eq. (56).

| | $\varepsilon^0$ | $\varepsilon^1$ | $\varepsilon^2$ | $\varepsilon^3$ | $\varepsilon^4$ |
|---|---|---|---|---|---|
| $J_1^D$ | 1 | $-27.036518$ | $370.42146$ | $-3430.4542$ | $24170.732$ |
| $J_2^D$ | 3 | $18.849556i$ | $-64.152429$ | $-38.465821$ $-155.03138i$ | $245.95795$ $-241.68788i$ |
| $J_3^D$ | $-2$ | $27.036518$ $-6.2831853i$ | $-172.87372$ $+84.937727i$ | $696.44831$ $-563.76967i$ | $-1969.4134$ $+2467.3908i$ |
| $J_4^D$ | 1 | $-27.036519$ | $370.42148$ | $-3430.4545$ | $24170.735$ |
| $J_5^D$ | 0 | $-2.4038459e-07$ | $6.2587777e-06$ | $-8.2784834e-05$ | $0.00074184352$ |
| $J_6^D$ | 0 | 0 | $1.9543798e-05$ $+4.2290642e-06i$ | $-0.00024013289$ $+1.2687191e-05i$ | $0.0020435243$ $+3.2789125e-06i$ |
| $J_7^D$ | 4 | $25.132741i$ | $-85.536571$ | $-22.438396$ $-206.70851i$ | $366.90758$ $-140.9846i$ |
| $J_8^D$ | 0 | 0 | $1.9543798e-05$ $+4.2290642e-06i$ | $-0.00035735832$ $-4.4482397e-05i$ | $0.0034638444$ $+0.00023755484i$ |
| $J_9^D$ | 0 | 0 | 0 | $-2.0889958e-05$ $-4.2290656e-06i$ | $0.00039240758$ $+4.4482410e-05i$ |
| $J_{10}^D$ | $-0.25$ | $-1.5707963i$ | $4.5235687$ | $2.6044566$ $+7.7515692i$ | $-5.343971$ $+16.364284i$ |
| $J_{11}^D$ | 0 | 0 | $1.9543798e-05$ $+4.2290642e-06i$ | $-0.00012290746$ $+6.9856779e-05i$ | $0.00031490245$ $-0.0003622524i$ |
| $J_{12}^D$ | 0 | 0 | 0 | 0 | $-0.00015900538$ $-6.5627738e-05i$ |
| $J_{13}^D$ | $-0.25$ | $4.8076917e-07$ $-1.5707963i$ | $4.5235436$ $-2.7186833e-06i$ | $2.6047751$ $+7.751561i$ | $-5.3467737$ $+16.364273i$ |

| | | | | | |
|---|---|---|---|---|---|
| $J^D_{14}$ | $-0.25$ | $4.8076917e-07$ $-1.5707963i$ | $4.5235115$ $-9.6664308e-06i$ | $2.605166$ $+7.7515402i$ | $-5.3501444$ $+16.364244i$ |
| $J^D_{15}$ | $0$ | $0$ | $0$ | $1.7159607e-05$ $+3.4738751e-06i$ | $-0.00019172322$ $+1.7369376e-05i$ |
| $J^D_{16}$ | $-0.5$ | $4.8076917e-07$ $-3.1415927i$ | $10.692034$ $-5.4373666e-06i$ | $6.4114039$ $+25.838541i$ | $-40.996637$ $+40.281274i$ |
| $J^D_{17}$ | $0$ | $2.4038459e-07$ | $-2.5284274e-06$ $+7.5519045e-07i$ | $-2.9067874e-05$ $-1.6401420e-05i$ | $0.00074430036$ $+0.00013138534i$ |
| $J^D_{18}$ | $0$ | $0$ | $4.8076920e-07$ | $-4.6836770e-05$ $-6.9477502e-06i$ | $0.00081023924$ $+7.3078241e-05i$ |
| $J^D_{19}$ | $0.25$ | $-2.4038459e-07$ $+1.5707963i$ | $-4.5235499$ $+2.7186833e-06i$ | $4.6078066$ $-7.7514954i$ | $15.084724$ $+28.951873i$ |
| $J^D_{20}$ | $0$ | $0$ | $-1.6449343$ | $-1.2020192$ $-10.335419i$ | $30.304558$ $-7.5527294i$ |
| $J^D_{21}$ | $0.25$ | $1.5707963i$ | $-2.8786542$ $-4.2290642e-06i$ | $5.8100602$ $+2.5837865i$ | $-15.220507$ $+36.505275i$ |

Table 6: Numerical results for the first five terms of the ε-expansion of the master integrals $J^D_1$-$J^D_{21}$ for the kinematic point specified by eq. (56).

| | $\varepsilon^0$ | $\varepsilon^1$ | $\varepsilon^2$ | $\varepsilon^3$ | $\varepsilon^4$ |
|---|---|---|---|---|---|
| $J^E_1$ | $3$ | $18.849556i$ | $-64.152429$ | $-38.465821$ $-155.03138i$ | $245.95795$ $-241.68788i$ |
| $J^E_2$ | $3$ | $10.3366$ | $12.872746$ | $-35.016679$ | $-190.47967$ |
| $J^E_3$ | $4$ | $25.132741i$ | $-85.536571$ | $-22.438396$ $-206.70851i$ | $366.90758$ $-140.9846i$ |
| $J^E_4$ | $-0.25$ | $-1.5707963i$ | $4.5235687$ | $2.6044566$ $+7.7515692i$ | $-5.343971$ $+16.364284i$ |
| $J^E_5$ | $-2.25$ | $-5.1682998$ $-4.712389i$ | $11.58622$ $-16.236693i$ | $79.269683$ $-15.261768i$ | $183.48426$ $+59.534707i$ |
| $J^E_6$ | $0$ | $0$ | $1.4839624$ $-5.4122309i$ | $21.221271$ $-16.35364i$ | $56.133181$ $+2.8635786i$ |
| $J^E_7$ | $4$ | $8.613833$ $+9.424778i$ | $-18.738162$ $+32.473385i$ | $-119.78077$ $+64.835609i$ | $-338.51852$ $+84.701754i$ |
| $J^E_8$ | $-1.5$ | $-5.1682998$ | $-8.4032506$ $-10.824462i$ | $30.435804$ $-35.59504i$ | $99.441356$ $-40.090141i$ |

Table 7: Numerical results for the first five terms of the ε-expansion of the master integrals $J^E_1$-$J^E_8$ for the kinematic point specified by eq. (56).

| | $\varepsilon^0$ | $\varepsilon^1$ | $\varepsilon^2$ | $\varepsilon^3$ | $\varepsilon^4$ |
|---|---|---|---|---|---|
| $J^A_1$ | $1$ | $-27.036518$ | $367.13159$ | $-3339.1035$ | $22888.178$ |

| | | | | | |
|---|---|---|---|---|---|
| $J_2^{\tilde{A}}$ | $-2$ | 27.036518 $-6.2831853i$ | $-172.87372$ $+84.937727i$ | 696.44831 $-563.76967i$ | $-1969.4134$ $+2467.3908i$ |
| $J_3^{\tilde{A}}$ | 1 | $-27.036518$ | 370.42146 | $-3430.4542$ | 24170.732 |
| $J_4^{\tilde{A}}$ | 0 | $1.3461548e-06$ | $-3.5049179e-05$ | 0.00046359534 | $-0.0041543258$ |
| $J_5^{\tilde{A}}$ | 1 | $-27.036513$ | 370.42132 | $-3430.4524$ | 24170.716 |
| $J_6^{\tilde{A}}$ | $-3$ | $-1.1802618$ | 4.7026326 | 40.376827 | 61.781473 |
| $J_7^{\tilde{A}}$ | 0 | $-0.00098058067$ | 0.028472648 | $-0.41694744$ | 4.1081553 |
| $J_8^{\tilde{A}}$ | 0 | 0 | $2.4038461e-07$ | $-6.7395475e-06$ | $9.5473101e-05$ |
| $J_9^{\tilde{A}}$ | $-0.25$ | $-1.5707963i$ | 4.5235687 | 2.6044566 $+7.7515692i$ | $-5.343971$ $+16.364284i$ |
| $J_{10}^{\tilde{A}}$ | 0 | 0 | $1.6271369e-05$ | $-0.00018564339$ | 0.0017130853 |
| $J_{11}^{\tilde{A}}$ | 0 | 0 | $-0.018004884$ $-0.0036449972i$ | 0.39305146 $+0.049274018i$ | $-4.4568117$ $-0.3270537i$ |
| $J_{12}^{\tilde{A}}$ | 0 | 0 | 0 | $-1.1118058e-05$ $-2.1145333e-06i$ | 0.00022465136 $+2.4355742e-05i$ |
| $J_{13}^{\tilde{A}}$ | 0 | 0 | 0 | $3.9541684e-07$ | $-1.0584234e-05$ |
| $J_{14}^{\tilde{A}}$ | 0 | 0 | 0 | $-2.2143346e-06$ | $5.9271717e-05$ |
| $J_{15}^{\tilde{A}}$ | 0 | 0 | $1.3461543e-06$ | $-3.5527143e-05$ | 0.00047537778 |
| $J_{16}^{\tilde{A}}$ | 2.25 | 0.59013088 $+4.712389i$ | $-15.980065$ $+1.8539508i$ | $-32.897804$ $-21.125246i$ | $-19.652476$ $-55.947576i$ |
| $J_{17}^{\tilde{A}}$ | 0 | 0 | 0 | $-3.7776390e-06$ | $3.9060762e-05$ |
| $J_{18}^{\tilde{A}}$ | 0 | 0 | $-3.5372564e-06$ | $3.2801999e-05$ | $-0.00029428849$ |
| $J_{19}^{\tilde{A}}$ | 0 | 0 | 0 | $1.7377134e-05$ | $-0.00017967946$ |
| $J_{20}^{\tilde{A}}$ | 0 | 0 | 0 | 0 | $-0.0001612198$ $-6.5627778e-05i$ |
| $J_{21}^{\tilde{A}}$ | 0 | 0 | 0 | $-0.00014032988$ $-6.1398731e-05i$ | 0.0014820288 $+0.00013358997i$ |
| $J_{22}^{\tilde{A}}$ | 0 | 0.00049029033 | $-0.0066278718$ $+0.0015402925i$ | 0.04237916 $-0.020822073i$ | $-0.17073098$ $+0.13820541i$ |
| $J_{23}^{\tilde{A}}$ | 0 | 0 | $2.4038461e-07$ | $-2.5004928e-05$ $-3.4738760e-06i$ | 0.00046454318 $+4.0013005e-05i$ |
| $J_{24}^{\tilde{A}}$ | 0 | 0 | 0 | $-2.4038461e-07$ | $6.3441307e-06$ |
| $J_{25}^{\tilde{A}}$ | 0 | 0 | 0 | $1.1057694e-06$ | $-2.9183006e-05$ |
| $J_{26}^{\tilde{A}}$ | 0 | 0 | $-1.2019401e-07$ | $2.8890454e-06$ | $-3.5048783e-05$ |
| $J_{27}^{\tilde{A}}$ | 0 | 0 | $1.1658658e-06$ | $-2.8415975e-05$ | 0.00034964705 |
| $J_{28}^{\tilde{A}}$ | 0 | 0 | 0 | 0 | $-2.1367936e-05$ $-4.2290668e-06i$ |
| $J_{29}^{\tilde{A}}$ | 0 | 0 | 0 | 0 | $-4.4286581e-06$ |
| $J_{30}^{\tilde{A}}$ | 0 | 0 | $-1.4141994e-05$ | 0.00014590005 | $-0.0012333275$ |
| $J_{31}^{\tilde{A}}$ | 0 | 0 | 0 | $-2.2143242e-06$ | $5.0414304e-05$ |
| $J_{32}^{\tilde{A}}$ | 0 | 0 | $1.6271369e-05$ | $-0.00018161015$ | 0.0016139836 |
| $J_{33}^{\tilde{A}}$ | 0 | 0 | 0 | 0 | $-4.4286691e-06$ |

| | | | | | |
|---|---|---|---|---|---|
| $J_{34}^{\tilde{A}}$ | 0 | $-6.7307738e-07$ | $1.6178435e-05$ | $-0.00019627053$ | $0.0016017851$ |
| $J_{35}^{\tilde{A}}$ | 0 | 0 | $-1.6271369e-05$ | $8.7770955e-05$ $-5.1118013e-05i$ | $-0.00027624501$ $+0.0002693086i$ |
| $J_{36}^{\tilde{A}}$ | 0 | 0 | 0 | $-1.3599495e-05$ | $-0.0001241996$ $-0.00010918375i$ |
| $J_{37}^{\tilde{A}}$ | 0 | 0 | $2.4038461e-07$ | $-3.0091969e-06$ $+7.5519053e-07i$ | $1.7768897e-05$ $-9.4536706e-06i$ |
| $J_{38}^{\tilde{A}}$ | 0 | 0 | 0 | $-6.3670767e-06i$ | $5.8890108e-05i$ |
| $J_{39}^{\tilde{A}}$ | 0 | 0 | 0 | $-7.5753153e-12$ | $-4.4286099e-06$ |
| $J_{40}^{\tilde{A}}$ | 0 | 0 | 0 | 0 | $-4.4286798e-06$ |
| $J_{41}^{\tilde{A}}$ | 0 | 0 | $1.0929748e-05i$ | $-0.0001194604i$ | $0.0010190409i$ |
| $J_{42}^{\tilde{A}}$ | 0 | 0 | 0 | 0 | $6.1398709e-05$ $-0.00013985186i$ |
| $J_{43}^{\tilde{A}}$ | 0 | 0 | 0 | 0 | $-9.3234579e-05$ $-4.0932474e-05i$ |
| $J_{44}^{\tilde{A}}$ | 0 | 0 | 0 | $1.9056544e-05$ $-4.0102674e-05i$ | $-9.5921926e-05$ $+0.00052586569i$ |
| $J_{45}^{\tilde{A}}$ | 0 | 0 | 0 | 0 | $0.00010223599$ $-0.00022971381i$ |
| $J_{46}^{\tilde{A}}$ | 0 | 0 | $1.4141994e-05$ | $-6.8938801e-05$ $+4.4428386e-05i$ | $0.00018137956$ $-0.00021057982i$ |
| $J_{47}^{\tilde{A}}$ | 0 | 0 | 0 | $7.5753153e-12$ | $-0.00034846914$ $-0.00015002229i$ |
| $J_{48}^{\tilde{A}}$ | 0 | 0 | 0 | $3.1762740e-05$ $-6.5868683e-05i$ | $-0.00015622754$ $+0.00086515126i$ |

Table 8: Numerical results for the first five terms of the $\varepsilon$-expansion of the master integrals $J_1^{\tilde{A}}$-$J_{48}^{\tilde{A}}$ for the kinematic point specified by eq. (56).

| | $\varepsilon^0$ | $\varepsilon^1$ | $\varepsilon^2$ | $\varepsilon^3$ | $\varepsilon^4$ |
|---|---|---|---|---|---|
| $J_1^{\tilde{B}}$ | 1 | $-27.036518$ | $367.13159$ | $-3339.1035$ | $22888.178$ |
| $J_2^{\tilde{B}}$ | 1 | $-27.036518$ | $370.42146$ | $-3430.4542$ | $24170.732$ |
| $J_3^{\tilde{B}}$ | 3 | $18.849556i$ | $-64.152429$ | $-38.465821$ $-155.03138i$ | $245.95795$ $-241.68788i$ |
| $J_4^{\tilde{B}}$ | 1 | $-27.036522$ | $370.42157$ | $-3430.4557$ | $24170.745$ |
| $J_5^{\tilde{B}}$ | 1 | $-27.03652$ | $370.42151$ | $-3430.4549$ | $24170.738$ |
| $J_6^{\tilde{B}}$ | $-2$ | $27.036518$ $-6.2831853i$ | $-172.87372$ $+84.937727i$ | $696.44831$ $-563.76967i$ | $-1969.4134$ $+2467.3908i$ |
| $J_7^{\tilde{B}}$ | 0 | $-0.00098058067$ | $0.028472648$ | $-0.41694744$ | $4.1081553$ |
| $J_8^{\tilde{B}}$ | 0 | 0 | $2.4038461e-07$ | $-6.7395475e-06$ | $9.5473101e-05$ |
| $J_9^{\tilde{B}}$ | 1 | $-27.036513$ | $370.42132$ | $-3430.4524$ | $24170.716$ |

| | | | | | |
|---|---|---|---|---|---|
| $J^{\tilde{B}}_{10}$ | 0 | 1.3461548e − 06 | −3.5049179e − 05 | 0.00046359534 | −0.0041543258 |
| $J^{\tilde{B}}_{11}$ | 0 | 0 | 1.9543798e − 05 +4.2290642e − 06i | −0.00024013289 +1.2687191e − 05i | 0.0020435243 +3.2789125e − 06i |
| $J^{\tilde{B}}_{12}$ | −0.25 | −1.5707963i | 4.5235687 | 2.6044566 +7.7515692i | −5.343971 +16.364284i |
| $J^{\tilde{B}}_{13}$ | 0 | 0 | −1.9543798e − 05 −4.2290642e − 06i | 0.00037381959 +4.8711457e − 05i | −0.0037377082 −0.00028203719i |
| $J^{\tilde{B}}_{14}$ | 0 | 0 | 0 | 3.9541684e − 07 | −1.0584234e − 05 |
| $J^{\tilde{B}}_{15}$ | 0 | 0 | 1.9543798e − 05 +4.2290642e − 06i | −0.00035735832 −4.4482397e − 05i | 0.0034638444 +0.00023755484i |
| $J^{\tilde{B}}_{16}$ | 0 | 0 | 0 | −2.0889958e − 05 −4.2290656e − 06i | 0.00039240758 +4.4482410e − 05i |
| $J^{\tilde{B}}_{17}$ | 0 | 0 | 1.3461543e − 06 | −3.5527143e − 05 | 0.00047537778 |
| $J^{\tilde{B}}_{18}$ | 0 | 0 | 0 | −2.2143346e − 06 | 5.9271717e − 05 |
| $J^{\tilde{B}}_{19}$ | 0 | 0 | 0 | −1.8189173e − 06 | 4.8687470e − 05 |
| $J^{\tilde{B}}_{20}$ | 0 | 0 | 1.1057689e − 06 | −2.9182995e − 05 | 0.00039048872 |
| $J^{\tilde{B}}_{21}$ | 0 | 0 | −1.9543798e − 05 −4.2290642e − 06i | 0.00023791856 −1.2687192e − 05i | −0.0020517854 −1.7191978e − 05i |
| $J^{\tilde{B}}_{22}$ | −0.25 | 1.1057686e − 06 −1.5707963i | 4.5235525 +2.7186850e − 06i | 2.6046845 +7.7515773i | −5.3460644 +16.364295i |
| $J^{\tilde{B}}_{23}$ | 0 | 0 | 0 | −3.7303486e − 06 −7.5519008e − 07i | 4.1678955e − 05 −3.7759502e − 06i |
| $J^{\tilde{B}}_{24}$ | 0 | 0 | −3.4899620e − 06 −7.5518962e − 07i | 3.5420159e − 05 −3.7759477e − 06i | −0.00028155701 −8.1374147e − 06i |
| $J^{\tilde{B}}_{25}$ | 0 | 0 | 0 | −1.8675616e − 05 −4.2290638e − 06i | 0.00033313574 +4.4482390e − 05i |
| $J^{\tilde{B}}_{26}$ | 0 | 0 | 0 | −2.3104304e − 05 −4.2290685e − 06i | 0.0004516795 +4.4482440e − 05i |
| $J^{\tilde{B}}_{27}$ | 0 | 0 | −1.5379816e − 05 −3.5742116e − 06i | 0.00028778079 +4.1168700e − 05i | −0.0028072667 −0.00023836498i |
| $J^{\tilde{B}}_{28}$ | 0 | 0 | 0 | −2.4038461e − 07 | 6.3441307e − 06 |
| $J^{\tilde{B}}_{29}$ | 0 | 0 | 0 | 1.1057694e − 06 | −2.9183006e − 05 |
| $J^{\tilde{B}}_{30}$ | 0 | 0 | −1.2019401e − 07 | 2.8890454e − 06 | −3.5048783e − 05 |
| $J^{\tilde{B}}_{31}$ | 0 | 0 | 1.1658658e − 06 | −2.8415975e − 05 | 0.00034964705 |
| $J^{\tilde{B}}_{32}$ | 0 | 0 | 0 | 1.9543803e − 05 +4.2290656e − 06i | −0.00035735841 −4.4482410e − 05i |
| $J^{\tilde{B}}_{33}$ | 0 | 0 | 0 | 0 | −3.6378453e − 06 −2.3381840e − 12i |
| $J^{\tilde{B}}_{34}$ | 0 | 0 | −1.5379816e − 05 −3.5742116e − 06i | 0.00017292344 −1.0722632e − 05i | −0.0013253259 −2.7711815e − 06i |
| $J^{\tilde{B}}_{35}$ | 0 | 0 | 0 | −1.8189274e − 06 −2.3381830e − 12i | 4.1411880e − 05 −1.5198189e − 11i |
| $J^{\tilde{B}}_{36}$ | 0 | 0 | 1.9543798e − 05 | −0.00023609962 | 0.0019428406 |

| | | | | | |
|---|---|---|---|---|---|
| | | | +4.2290642e − 06i | +1.2687196e − 05i | +3.2789405e − 06i |
| $J^{\tilde{B}}_{37}$ | 0 | 0 | 0 | 0 | 7.9083368e − 07 |
| $J^{\tilde{B}}_{38}$ | 0 | 0 | 0 | 0 | −0.00015900538<br>−6.5627738e − 05i |
| $J^{\tilde{B}}_{39}$ | 0 | 0 | 0 | 0 | −3.6378346e − 06 |
| $J^{\tilde{B}}_{40}$ | 0 | −5.5288431e − 07 | 1.3289414e − 05 | −0.00016122206 | 0.0013157509 |
| $J^{\tilde{B}}_{41}$ | 0 | 0 | 0 | 0 | 7.9083370e − 07 |
| $J^{\tilde{B}}_{42}$ | 0 | 0 | 0 | 3.9541681e − 07 | −9.0025658e − 06 |
| $J^{\tilde{B}}_{43}$ | 0 | 1.2019307e − 07 | −2.8890228e − 06 | 3.5048510e − 05 | −0.00028603471 |
| $J^{\tilde{B}}_{44}$ | 0 | 0 | 0 | 3.9541690e − 07 | −9.0025675e − 06 |
| $J^{\tilde{B}}_{45}$ | 0 | 0 | 0 | 0 | −0.0001612198<br>−6.5627778e − 05i |
| $J^{\tilde{B}}_{46}$ | −0.25 | 6.7307738e − 07<br>−1.5707963i | 4.5235518 | 2.6048112<br>+7.7516306i | −5.3474596<br>+16.364084i |
| $J^{\tilde{B}}_{47}$ | 0 | 0 | 0 | 1.3461536e − 06 | −3.5527128e − 05 |
| $J^{\tilde{B}}_{48}$ | 0 | 0 | −1.2019362e − 07 | 2.8890361e − 06 | −3.5048674e − 05 |
| $J^{\tilde{B}}_{49}$ | 0 | 0 | 1.2860574e − 06 | −3.1390316e − 05 | 0.00038679918 |
| $J^{\tilde{B}}_{50}$ | 0 | 0 | 0 | 2.0889958e − 05<br>+4.2290656e − 06i | −0.0002334022<br>+2.1145329e − 05i |
| $J^{\tilde{B}}_{51}$ | 0 | 0 | 0 | 0 | −0.0001567912<br>−6.5627806e − 05i |
| $J^{\tilde{B}}_{52}$ | 0 | 0 | 0 | −0.00011672886<br>−5.1891363e − 05i | 0.0014749594<br>+0.00022383517i |
| $J^{\tilde{B}}_{53}$ | 0 | 0 | 0 | 0 | 0.00015315346<br>+6.5627849e − 05i |
| $J^{\tilde{B}}_{54}$ | 0 | 0 | 0 | −2.4921275e − 06<br>+1.1516883e − 05i | −1.2460637e − 05<br>−0.00011660494i |
| $J^{\tilde{B}}_{55}$ | 0 | 0 | 0 | −2.1438006e − 06<br>−7.5518779e − 07i | −1.9238182e − 07<br>−3.7759367e − 06i |
| $J^{\tilde{B}}_{56}$ | 0 | 0 | 0 | 3.4899640e − 06<br>+7.5519006e − 07i | −3.5334824e − 05<br>+3.7759504e − 06i |
| $J^{\tilde{B}}_{57}$ | 0 | 0 | −2.3950317e − 06<br>+1.0270039e − 05i | −8.9597899e − 06<br>−0.00010681247i | −1.0730412e − 05<br>+0.00079461541i |
| $J^{\tilde{B}}_{58}$ | 0 | 0 | 1.2019231e − 07 | −2.8890047e − 06 | 3.5048291e − 05 |
| $J^{\tilde{B}}_{59}$ | 0 | 0 | −1.2019231e − 07 | 2.8890049e − 06 | −3.5048295e − 05 |
| $J^{\tilde{B}}_{60}$ | 0 | 0 | 0.82246838 | 0.60101121<br>+5.167717i | −15.152188<br>+3.7764321i |
| $J^{\tilde{B}}_{61}$ | 0 | 0 | 0 | −1.8714536e − 06<br>+2.9941309e − 13i | −6.9818556e − 06<br>−1.1758654e − 05i |
| $J^{\tilde{B}}_{62}$ | 0 | 0 | 1.6449366 | 1.2020261<br>+10.335434i | −30.304407<br>+7.5528691i |
| $J^{\tilde{B}}_{63}$ | 0 | 0 | −1.6449164<br>+4.2290642e − 06i | −1.2022602<br>−10.335416i | 30.306373<br>−7.5528548i |

| | | | | | |
|---|---|---|---|---|---|
| $J_{64}^{\tilde{B}}$ | 0 | 0 | 3.2898727 | 2.4040864 $+20.670871i$ | $-60.609098$ $+15.105801i$ |

Table 9: Numerical results for the first five terms of the ε-expansion of the master integrals $J_1^{\tilde{B}}$-$J_{64}^{\tilde{B}}$ for the kinematic point specified by eq. (56).

| | $\varepsilon^0$ | $\varepsilon^1$ | $\varepsilon^2$ | $\varepsilon^3$ | $\varepsilon^4$ |
|---|---|---|---|---|---|
| $J_1^{\tilde{C}}$ | $-2$ | 27.036518 $-6.2831853i$ | $-172.87372$ $+84.937727i$ | 696.44831 $-563.76967i$ | $-1969.4134$ $+2467.3908i$ |
| $J_2^{\tilde{C}}$ | 1 | $-27.036518$ | 370.42146 | $-3430.4542$ | 24170.732 |
| $J_3^{\tilde{C}}$ | 0 | $1.3461548e-06$ | $-3.5049179e-05$ | 0.00046359534 | $-0.0041543258$ |
| $J_4^{\tilde{C}}$ | 1 | $-27.036513$ | 370.42132 | $-3430.4524$ | 24170.716 |
| $J_5^{\tilde{C}}$ | $-3$ | $-1.1802618$ | 4.7026326 | 40.376827 | 61.781473 |
| $J_6^{\tilde{C}}$ | 1 | $-27.036519$ | 370.42148 | $-3430.4545$ | 24170.735 |
| $J_7^{\tilde{C}}$ | 0 | $-2.4038459e-07$ | $6.2587777e-06$ | $-8.2784834e-05$ | 0.00074184352 |
| $J_8^{\tilde{C}}$ | 3 | $18.849556i$ | $-64.152429$ | $-38.465821$ $-155.03138i$ | 245.95795 $-241.68788i$ |
| $J_9^{\tilde{C}}$ | $-0.25$ | $-1.5707963i$ | 4.5235687 | 2.6044566 $+7.7515692i$ | $-5.343971$ $+16.364284i$ |
| $J_{10}^{\tilde{C}}$ | 0 | 0 | $1.6271369e-05$ | $-0.00018564339$ | 0.0017130853 |
| $J_{11}^{\tilde{C}}$ | 0 | 0 | 0 | $-2.0889958e-05$ $-4.2290656e-06i$ | 0.00039240758 $+4.4482410e-05i$ |
| $J_{12}^{\tilde{C}}$ | 0 | 0 | $1.9543798e-05$ $+4.2290642e-06i$ | $-0.00035735832$ $-4.4482397e-05i$ | 0.0034638444 $+0.00023755484i$ |
| $J_{13}^{\tilde{C}}$ | 0 | 0 | $1.9543798e-05$ $+4.2290642e-06i$ | $-0.00024013289$ $+1.2687191e-05i$ | 0.0020435243 $+3.2789125e-06i$ |
| $J_{14}^{\tilde{C}}$ | 2.25 | 0.59013088 $+4.712389i$ | $-15.980065$ $+1.8539508i$ | $-32.897804$ $-21.125246i$ | $-19.652476$ $-55.947576i$ |
| $J_{15}^{\tilde{C}}$ | 0 | 0 | 0 | $-3.7776390e-06$ | $3.9060762e-05$ |
| $J_{16}^{\tilde{C}}$ | 0 | 0 | $-3.5372564e-06$ | $3.2801999e-05$ | $-0.00029428849$ |
| $J_{17}^{\tilde{C}}$ | 0 | 0 | 0 | $1.7377134e-05$ | $-0.00017967946$ |
| $J_{18}^{\tilde{C}}$ | 0 | 0 | 0 | 0 | $-0.0001612198$ $-6.5627778e-05i$ |
| $J_{19}^{\tilde{C}}$ | 0 | 0 | 0 | $-0.00014032988$ $-6.1398731e-05i$ | 0.0014820288 $+0.00013358997i$ |
| $J_{20}^{\tilde{C}}$ | 0 | $2.4038459e-07$ | $-3.0091966e-06$ $+7.5519045e-07i$ | $1.7768896e-05$ $-9.4536695e-06i$ | $-6.5938874e-05$ $+5.8307098e-05i$ |
| $J_{21}^{\tilde{C}}$ | 0 | 0 | $4.8076920e-07$ | $-4.6836770e-05$ $-6.9477502e-06i$ | 0.00081023924 $+7.3078241e-05i$ |
| $J_{22}^{\tilde{C}}$ | 0 | 0 | 0 | $1.7159607e-05$ $+3.4738751e-06i$ | $-0.00019172322$ $+1.7369376e-05i$ |

| | | | | | |
|---|---|---|---|---|---|
| $J_{23}^{\tilde{C}}$ | 0 | 0 | $1.6053832e-05$ $+3.4738737e-06i$ | $-0.00016293279$ $+1.7369369e-05i$ | $0.0012951626$ $+3.7432138e-05i$ |
| $J_{24}^{\tilde{C}}$ | 0 | 0 | 0 | 0 | $-0.00015900538$ $-6.5627738e-05i$ |
| $J_{25}^{\tilde{C}}$ | 0 | 0 | $-1.1057695e-06$ | $2.7684618e-05$ | $-0.00035201996$ |
| $J_{26}^{\tilde{C}}$ | 0 | $-1.1057702e-06$ | $2.5473093e-05$ | $-0.00029665089$ | $0.002327632$ |
| $J_{27}^{\tilde{C}}$ | 0 | 0 | $2.4038460e-07$ | $-6.0183934e-06$ | $7.6526059e-05$ |
| $J_{28}^{\tilde{C}}$ | 0 | 0 | 0 | $1.9543803e-05$ $+4.2290656e-06i$ | $-0.00035735841$ $-4.4482410e-05i$ |
| $J_{29}^{\tilde{C}}$ | 0 | 0 | $-0.01934747$ $+0.61798362i$ | $-1.9646373$ $+1.2980771i$ | $-4.0698471$ $-2.4207396i$ |
| $J_{30}^{\tilde{C}}$ | 0 | 0 | 0 | $2.1154771e-05$ | $-0.0002187402$ |
| $J_{31}^{\tilde{C}}$ | 0 | 0 | $1.9808621e-05$ | $-0.00018369108$ | $0.0016480146$ |
| $J_{32}^{\tilde{C}}$ | 0 | 0 | 0 | $2.0889958e-05$ $+4.2290656e-06i$ | $-0.0002334022$ $+2.1145329e-05i$ |
| $J_{33}^{\tilde{C}}$ | 0 | 0 | $-1.6271369e-05$ | $8.7770955e-05$ $-5.1118013e-05i$ | $-0.00027624501$ $+0.0002693086i$ |
| $J_{34}^{\tilde{C}}$ | 0 | 0 | 0 | $-1.3599495e-05$ | $-0.0001241996$ $-0.00010918375i$ |
| $J_{35}^{\tilde{C}}$ | $-0.25$ | $-0.098354594$ | $0.39188958$ | $3.3647156$ | $5.1488257$ |
| $J_{36}^{\tilde{C}}$ | 0 | 0 | $-0.019347228$ $+0.61797591i$ | $-1.9646055$ $+1.2980616i$ | $-4.0698705$ $-2.420706i$ |
| $J_{37}^{\tilde{C}}$ | 0 | 0 | 0 | $2.1179951e-05$ $+3.4244210e-06i$ | $-0.0002291281$ $+1.7891312e-05i$ |
| $J_{38}^{\tilde{C}}$ | 0 | 0 | $0.82246838$ | $0.60101121$ $+5.167717i$ | $-15.152188$ $+3.7764321i$ |
| $J_{39}^{\tilde{C}}$ | 0 | $6.7307738e-07$ | $0.0096754898$ $-0.30899219i$ | $0.98232125$ $-0.6490469i$ | $2.0349896$ $+1.2103518i$ |
| $J_{40}^{\tilde{C}}$ | $-0.625$ | $-0.24588711$ | $-0.65552679$ $-0.30899143i$ | $7.8593857$ $-10.396162i$ | $42.739533$ $-6.4784629i$ |
| $J_{41}^{\tilde{C}}$ | 0 | $2.4038459e-07$ | $3.3188881$ $-0.92697505i$ | $6.0163061$ $+17.547209i$ | $-49.56018$ $+19.008627i$ |

Table 10: Numerical results for the first five terms of the $\varepsilon$-expansion of the master integrals $J_1^{\tilde{C}}$-$J_{41}^{\tilde{C}}$ for the kinematic point specified by eq. (56).

| | $\varepsilon^0$ | $\varepsilon^1$ | $\varepsilon^2$ | $\varepsilon^3$ | $\varepsilon^4$ |
|---|---|---|---|---|---|
| $J_1^{\tilde{D}}$ | 1 | $-27.036518$ | $370.42146$ | $-3430.4542$ | $24170.732$ |
| $J_2^{\tilde{D}}$ | 3 | $18.849556i$ | $-64.152429$ | $-38.465821$ $-155.03138i$ | $245.95795$ $-241.68788i$ |
| $J_3^{\tilde{D}}$ | 1 | $-27.036522$ | $370.42157$ | $-3430.4557$ | $24170.745$ |
| $J_4^{\tilde{D}}$ | 1 | $-27.03652$ | $370.42151$ | $-3430.4549$ | $24170.738$ |

| | | | | | |
|---|---|---|---|---|---|
| $J_5^{\tilde{D}}$ | 1 | −27.036519 | 370.42148 | −3430.4545 | 24170.735 |
| $J_6^{\tilde{D}}$ | 0 | −2.4038459e−07 | 6.2587777e−06 | −8.2784834e−05 | 0.00074184352 |
| $J_7^{\tilde{D}}$ | 0 | 0 | 1.9543798e−05 +4.2290642e−06i | −0.00024013289 +1.2687191e−05i | 0.0020435243 +3.2789125e−06i |
| $J_8^{\tilde{D}}$ | −0.25 | −1.5707963i | 4.5235687 | 2.6044566 +7.7515692i | −5.343971 +16.364284i |
| $J_9^{\tilde{D}}$ | 0 | 0 | 1.9543798e−05 +4.2290642e−06i | −0.00023791856 +1.2687192e−05i | 0.0020517854 +1.7191978e−05i |
| $J_{10}^{\tilde{D}}$ | −0.25 | 1.1057686e−06 −1.5707963i | 4.5235525 +2.7186850e−06i | 2.6046845 +7.7515773i | −5.3460644 +16.364295i |
| $J_{11}^{\tilde{D}}$ | 0 | 0 | 0 | −3.7303486e−06 −7.5519008e−07i | 4.1678955e−05 −3.7759502e−06i |
| $J_{12}^{\tilde{D}}$ | 0 | 0 | −3.4899620e−06 −7.5518962e−07i | 3.5420159e−05 −3.7759477e−06i | −0.00028155701 −8.1374147e−06i |
| $J_{13}^{\tilde{D}}$ | −0.25 | 4.8076917e−07 −1.5707963i | 4.5235436 −2.7186833e−06i | 2.6047751 +7.751561i | −5.3467737 +16.364273i |
| $J_{14}^{\tilde{D}}$ | −0.25 | 4.8076917e−07 −1.5707963i | 4.5235115 −9.6664308e−06i | 2.605166 +7.7515402i | −5.3501444 +16.364244i |
| $J_{15}^{\tilde{D}}$ | 0 | 0 | 0 | 1.7159607e−05 +3.4738751e−06i | −0.00019172322 +1.7369376e−05i |
| $J_{16}^{\tilde{D}}$ | 0 | 0 | 1.6053832e−05 +3.4738737e−06i | −0.00016293279 +1.7369369e−05i | 0.0012951626 +3.7432138e−05i |
| $J_{17}^{\tilde{D}}$ | 0 | 0 | −1.3461534e−06 | 3.3702995e−05 | −0.00042854584 |
| $J_{18}^{\tilde{D}}$ | 0 | 1.3461532e−06 | −3.1010684e−05 | 0.00036113981 | −0.0028336361 |
| $J_{19}^{\tilde{D}}$ | 0.25 | 6.7307660e−07 +1.5707963i | −5.3460317 +4.2290650e−06i | −3.2055237 −12.919265i | 20.49689 −20.140632i |
| $J_{20}^{\tilde{D}}$ | 1 | 6.2831853i | −29.608813 | −33.256908 −103.35426i | 211.86477 −208.95931i |
| $J_{21}^{\tilde{D}}$ | −1.5 | 4.3269202e−07 −9.424778i | 44.413222 +2.7186841e−06i | 49.885353 +155.03139i | −317.79722 +313.43895i |
| $J_{22}^{\tilde{D}}$ | 0.5 | 4.3269202e−07 +3.1415927i | −14.804405 +2.7186841e−06i | −16.628463 −51.677117i | 105.93233 −104.47968i |
| $J_{23}^{\tilde{D}}$ | −2.25 | 13.518259 −10.995574i | −131.33915 −1.7621149e−07i | 1778.5356 +193.78923i | −12488.6 +397.77801i |

Table 11: Numerical results for the first five terms of the ε-expansion of the master integrals $J_1^{\tilde{D}}$-$J_{23}^{\tilde{D}}$ for the kinematic point specified by eq. (56).

| | $\varepsilon^0$ | $\varepsilon^1$ | $\varepsilon^2$ | $\varepsilon^3$ | $\varepsilon^4$ |
|---|---|---|---|---|---|
| $J_1^{\tilde{E}}$ | 3 | 18.849556i | −64.152429 | −38.465821 −155.03138i | 245.95795 −241.68788i |
| $J_2^{\tilde{E}}$ | −3 | −1.1802618 | 4.7026326 | 40.376827 | 61.781473 |

| | | | | | |
|---|---|---|---|---|---|
| $J_3^{\tilde{E}}$ | 3 | 10.3366 | 12.872746 | −35.016679 | −190.47967 |
| $J_4^{\tilde{E}}$ | −0.25 | −1.5707963i | 4.5235687 | 2.6044566 +7.7515692i | −5.343971 +16.364284i |
| $J_5^{\tilde{E}}$ | 2.25 | 0.59013088 +4.712389i | −15.980065 +1.8539508i | −32.897804 −21.125246i | −19.652476 −55.947576i |
| $J_6^{\tilde{E}}$ | 0 | 0 | −0.01934747 +0.61798362i | −1.9646373 +1.2980771i | −4.0698471 −2.4207396i |
| $J_7^{\tilde{E}}$ | −2.25 | −5.1682998 −4.712389i | 11.58622 −16.236693i | 79.269683 −15.261768i | 183.48426 +59.534707i |
| $J_8^{\tilde{E}}$ | 0 | 0 | 1.4839624 −5.4122309i | 21.221271 −16.35364i | 56.133181 +2.8635786i |
| $J_9^{\tilde{E}}$ | 0 | 0 | −6.0992261 | −25.688328 | −54.333255 |
| $J_{10}^{\tilde{E}}$ | 1 | 6.2831853i | −29.608813 | −33.256908 −103.35426i | 211.86477 −208.95931i |
| $J_{11}^{\tilde{E}}$ | 2 | −0.17289727 +3.1415927i | −17.155723 −4.7942473i | 5.7871067 −51.677128i | 242.98467 +65.13519i |
| $J_{12}^{\tilde{E}}$ | 5.75 | −1.5767854 +12.566371i | −47.560258 −4.7942473i | −35.032504 −274.23836i | 578.59928 −786.48374i |

Table 12: Numerical results for the first five terms of the ε-expansion of the master integrals $J_1^{\tilde{E}}$-$J_{12}^{\tilde{E}}$ for the kinematic point specified by eq. (56).

# 6 Large logarithms

Of particular interest are numerically large contributions from the master integrals. These arise from large logarithms. In the kinematic region of interest

$$-t \;\lesssim\; s \;\ll\; m^2 \tag{57}$$

these are logarithms of the form

$$L \;=\; \ln\left(\frac{s}{m_Z^2}\right). \tag{58}$$

At order $\varepsilon^j$ we can have at most $j$ powers of large logarithms. The leading logarithms are the ones which occur to power $j$ at order $\varepsilon^j$. We remark that this counting defines at order $\varepsilon^0$ constants as leading logarithms. The leading logarithms are easily obtained from our full result. To this aim we write the matrix $A$ in eq. (40) in the form

$$A \;=\; \sum_{j=1}^{N_L} \tilde{M}_j \tilde{\omega}_j, \tag{59}$$

such that

$$\tilde{\omega}_1 \;=\; d\ln L \tag{60}$$

and all other $\tilde{\omega}_j$'s are regular at $m^2 = \infty$. $\tilde{M}_1$ (all all others $\tilde{M}_j$'s) are $(N_F \times N_F)$-matrices with constant entries. Let us denote the boundary values of the master integrals at the boundary point of eq. (45) by $J_{\text{boundary}}$ and the $\varepsilon^0$-part by $J_{\text{boundary}}^{(0)}$. The leading logarithms are then given by

$$ J_{\text{LL}} = \sum_{j=0}^{\infty} \frac{1}{j!} (\varepsilon L)^j \tilde{M}_1^j J_{\text{boundary}}^{(0)}. \tag{61} $$

We see that the leading logarithms are determined by the matrix $\tilde{M}_1$ and the $\varepsilon^0$-term of the boundary values. In the supplementary electronic file attached to the arxiv version of this article we give for all topologies the matrix $\tilde{M}_1$ and the boundary values.

# 7  Conclusions

In this paper we computed all planar and non-planar two-loop double-box integrals with the exchange of three electroweak gauge bosons, among which at least one is a photon. These integrals are relevant for the NNLO electroweak corrections to Møller scattering. The planar and non-planar double-box integrals are distinguished by their mass configurations. We considered the cases with zero, one or two internal massive gauge bosons. The complexity of the calculation increases with the number of internal massive gauge bosons. While the integrals with three internal photons have been computed long time ago, the non-planar double-box integrals with two internal massive gauge bosons involve elliptic curves and require state-of-the-art techniques. We presented for all topologies a basis of master integrals, such that the differential equation is in an $\varepsilon$-factorised form. We expressed the results for the simpler topologies in terms of multiple polylogarithms, the results for the more complicated topologies in terms of iterated integrals. For all integrals we provided numerical evaluation routines. Of particular interest are numerically large contributions. These arise from large logarithms. We extracted for all master integrals the leading logarithms.

The double-box integrals with the exchange of three massive gauge bosons are expected to give numerically suppressed contributions. But they are of interest from a theoretical perspective, as the non-planar double-box integral with three internal massive gauge bosons is related to a curve of genus two [78]. The computation of these integrals is an interesting project for the future.

## Acknowledgments

This work has been supported by the Cluster of Excellence Precision Physics, Fundamental Interactions, and Structure of Matter (PRISMA EXC 2118/1) funded by the German Research Foundation (DFG) within the German Excellence Strategy (Project ID 390831469).

# A    Feynman diagrams

In this appendix we show the diagrams for all master sectors. Red lines correspond to particles with mass *m* and the uncoloured lines indicate massless particles.

## A.1    Planar double-box integrals

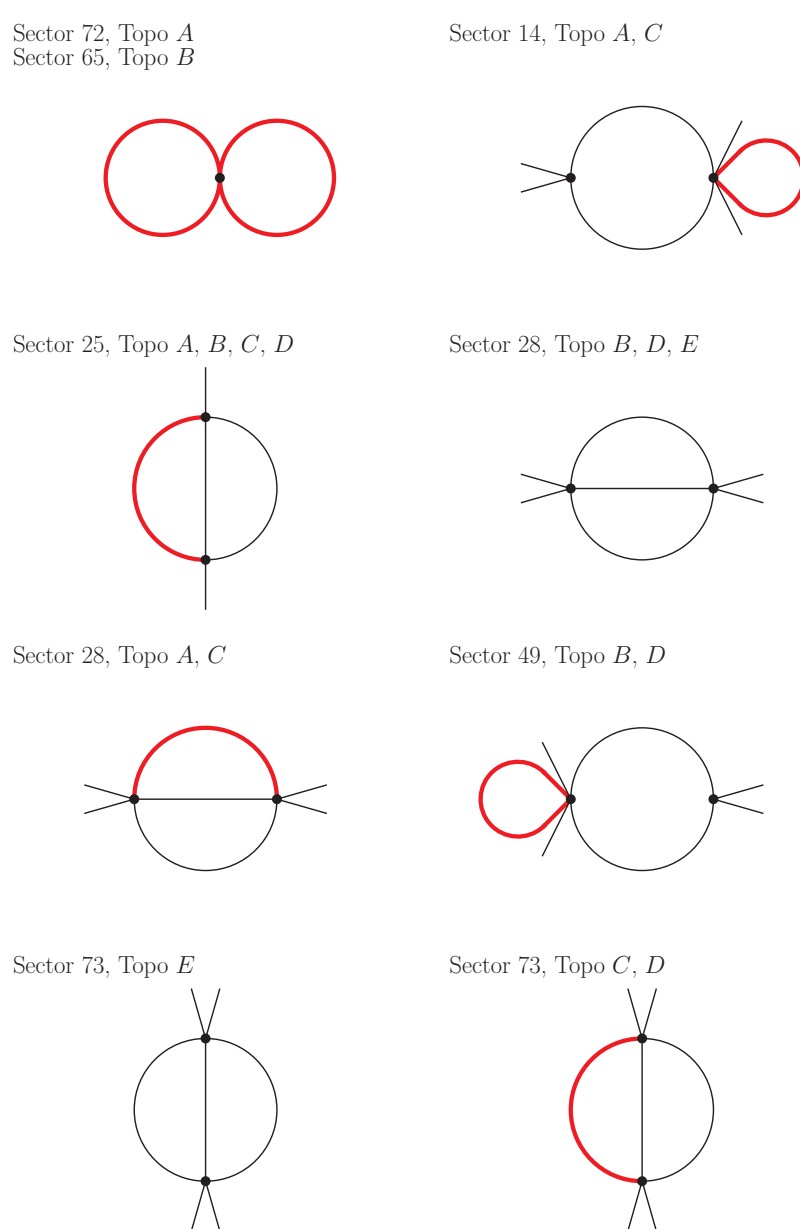

Figure 4: Master sectors for planar double-box integrals (part 1).

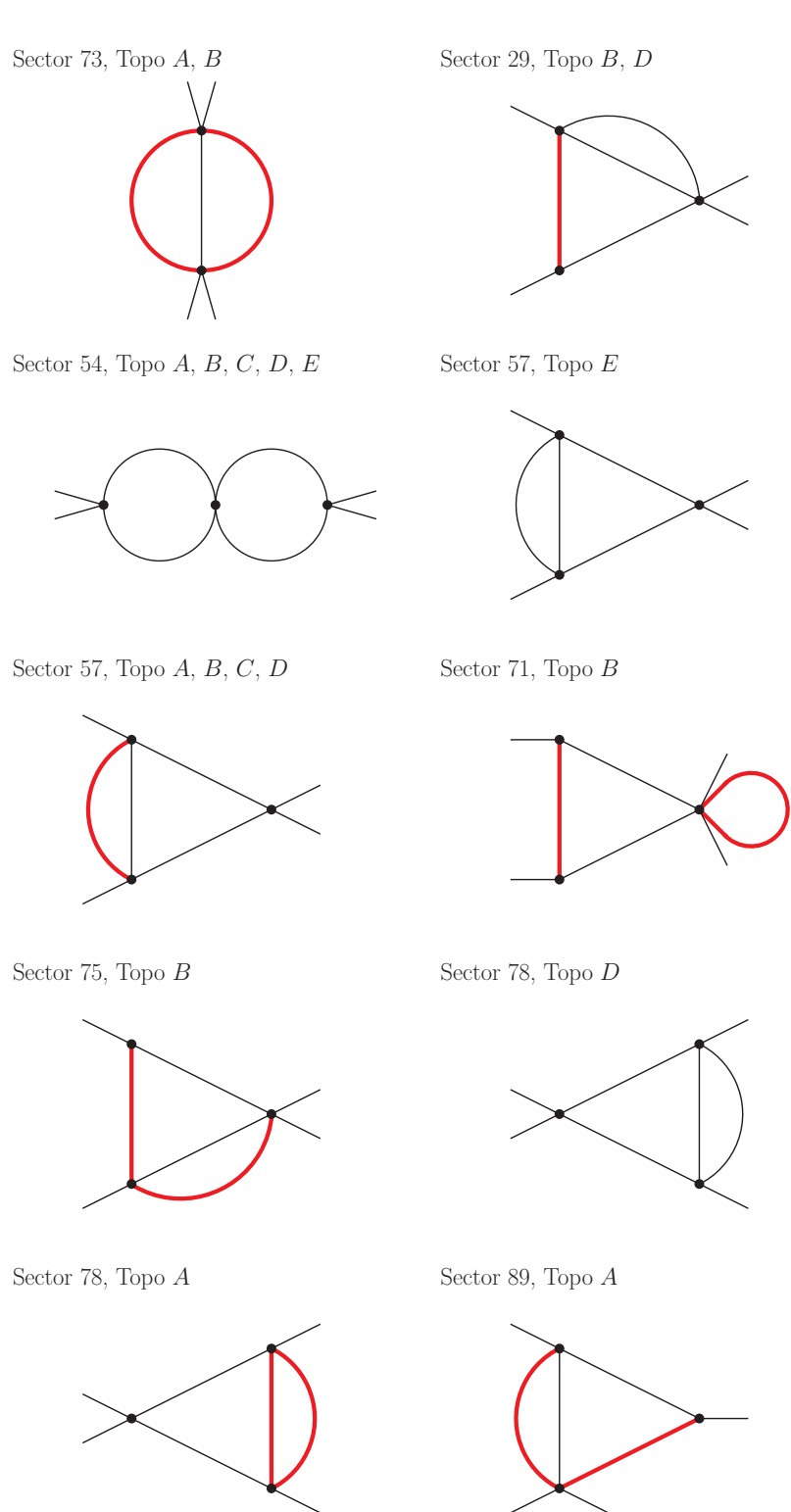

Figure 5: Master sectors for planar double-box integrals (part 2).

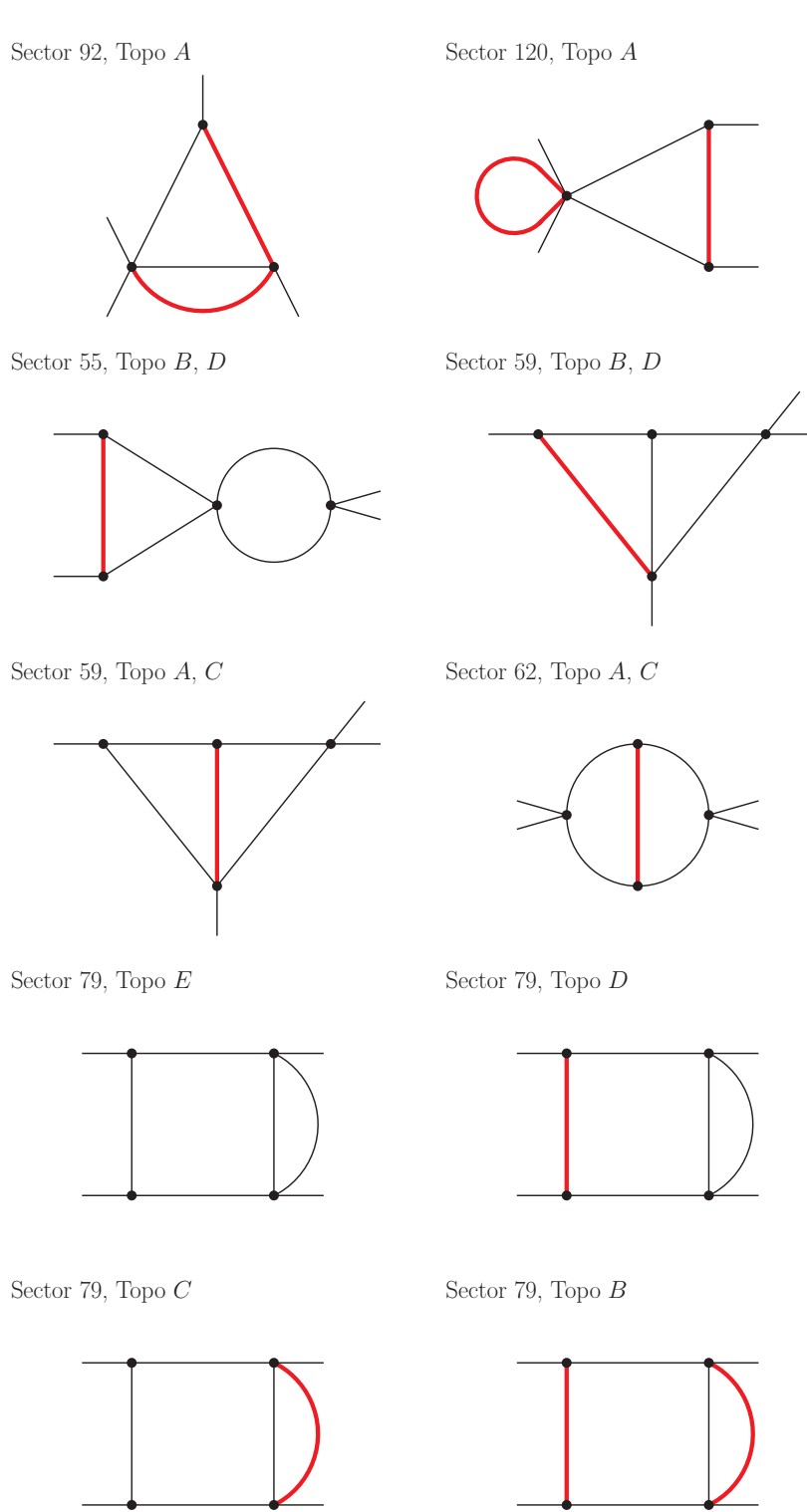

Figure 6: Master sectors for planar double-box integrals (part 3).

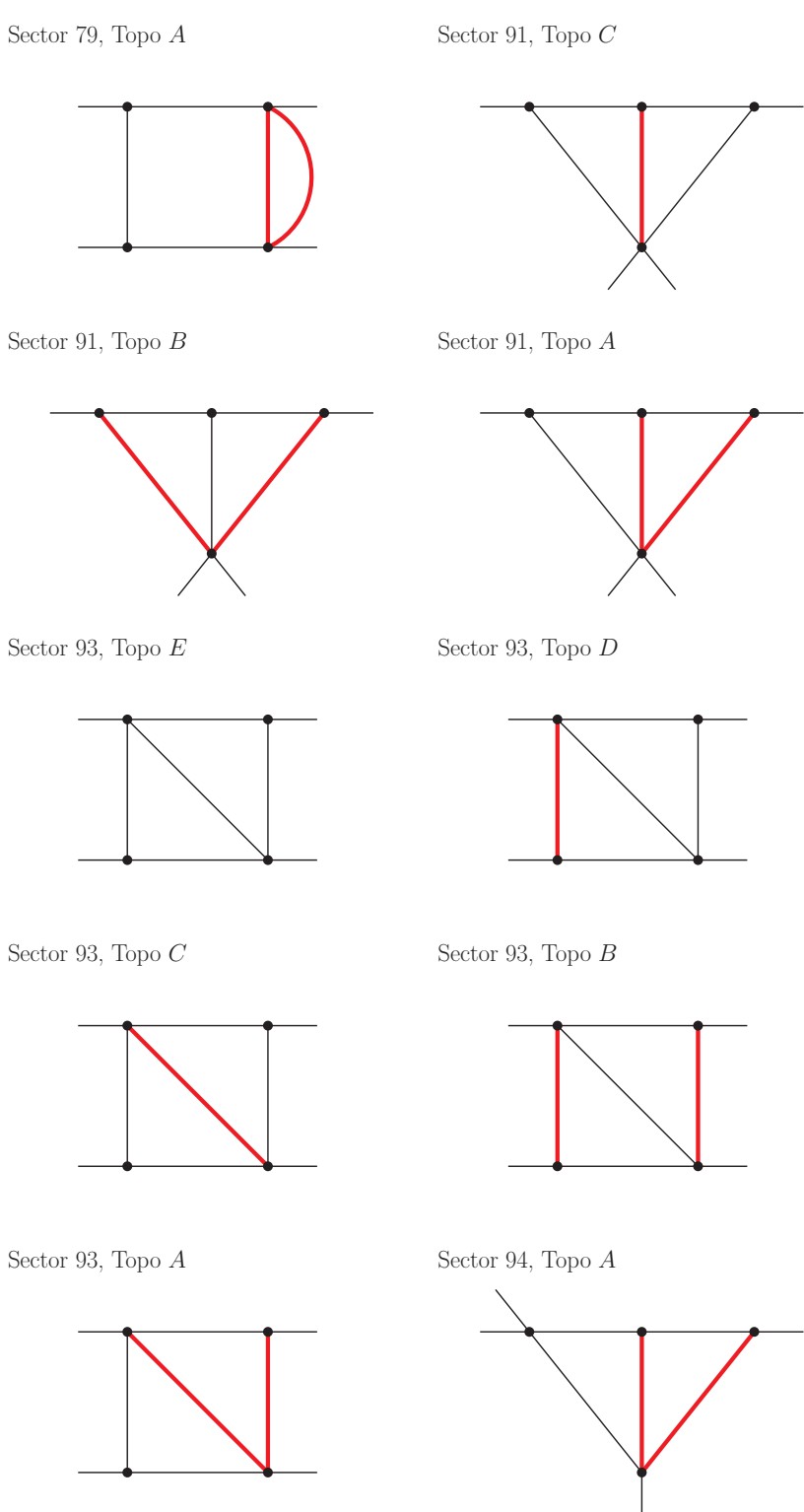

Figure 7: Master sectors for planar double-box integrals (part 4).

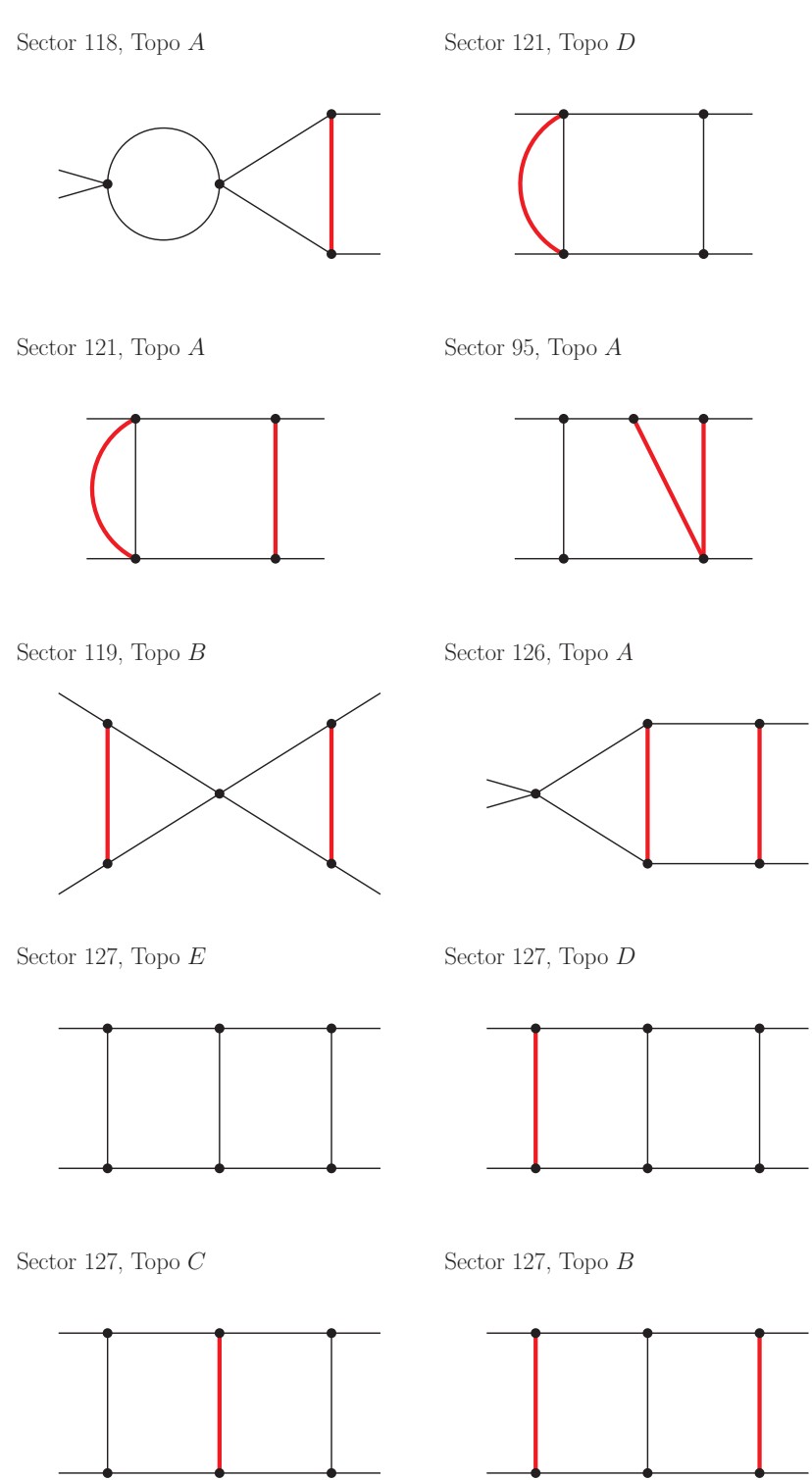

Figure 8: Master sectors for planar double-box integrals (part 5).

Sector 127, Topo $A$

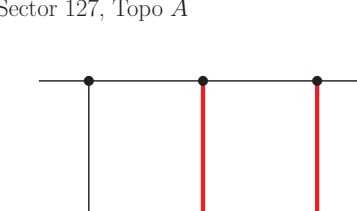

Figure 9: Master sectors for planar double-box integrals (part 6).

## A.2 Non-planar double-box integrals

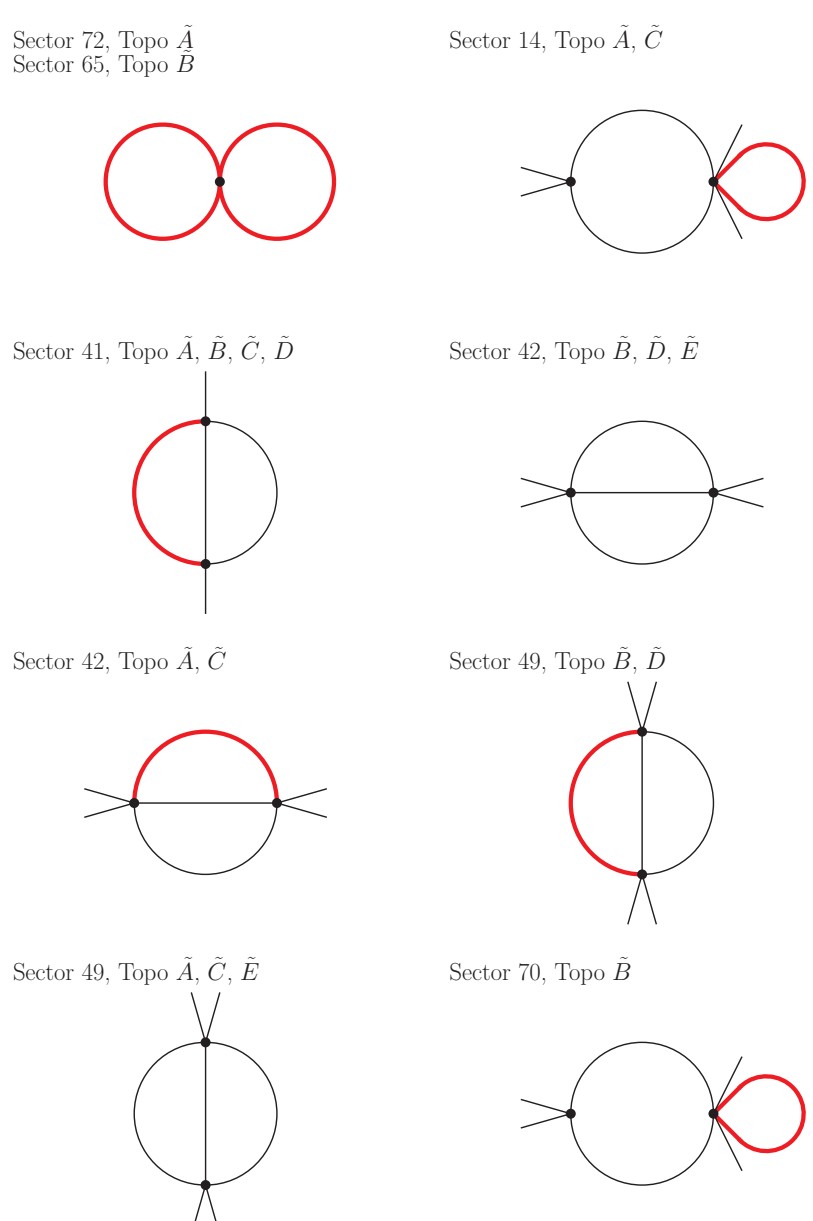

Figure 10: Master sectors for non-planar double-box integrals (part 1).

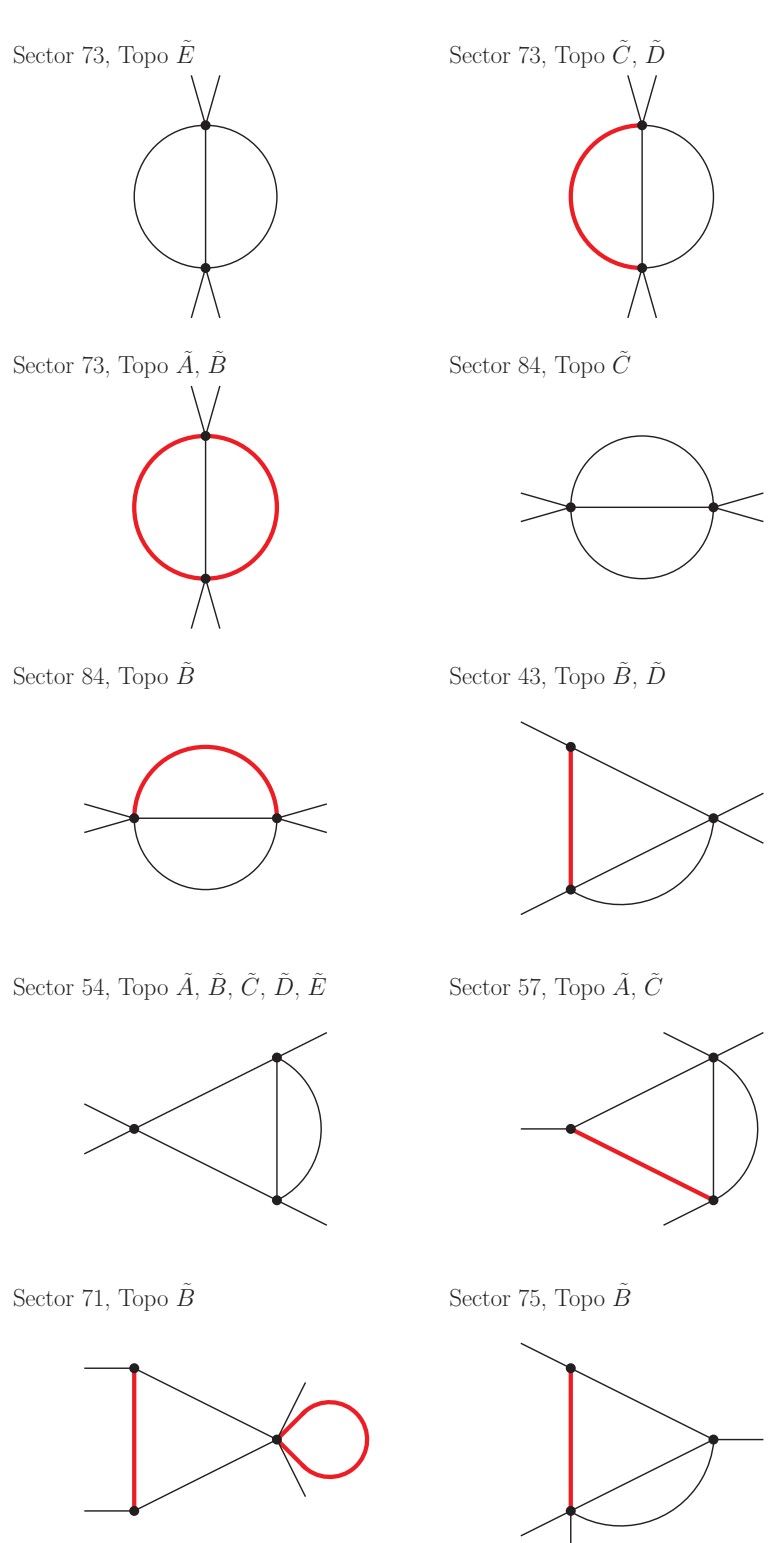

Figure 11: Master sectors for non-planar double-box integrals (part 2).

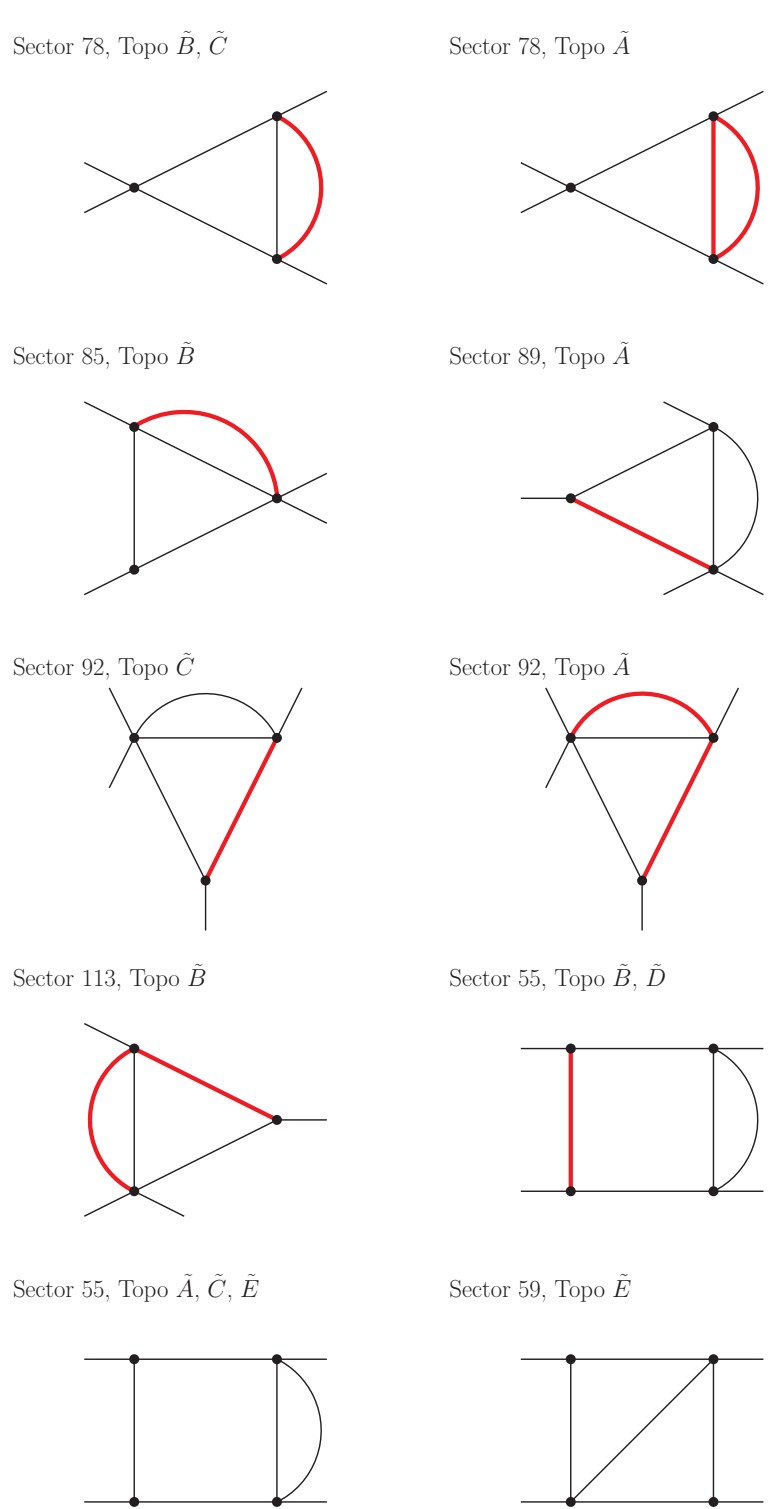

Figure 12: Master sectors for non-planar double-box integrals (part 3).

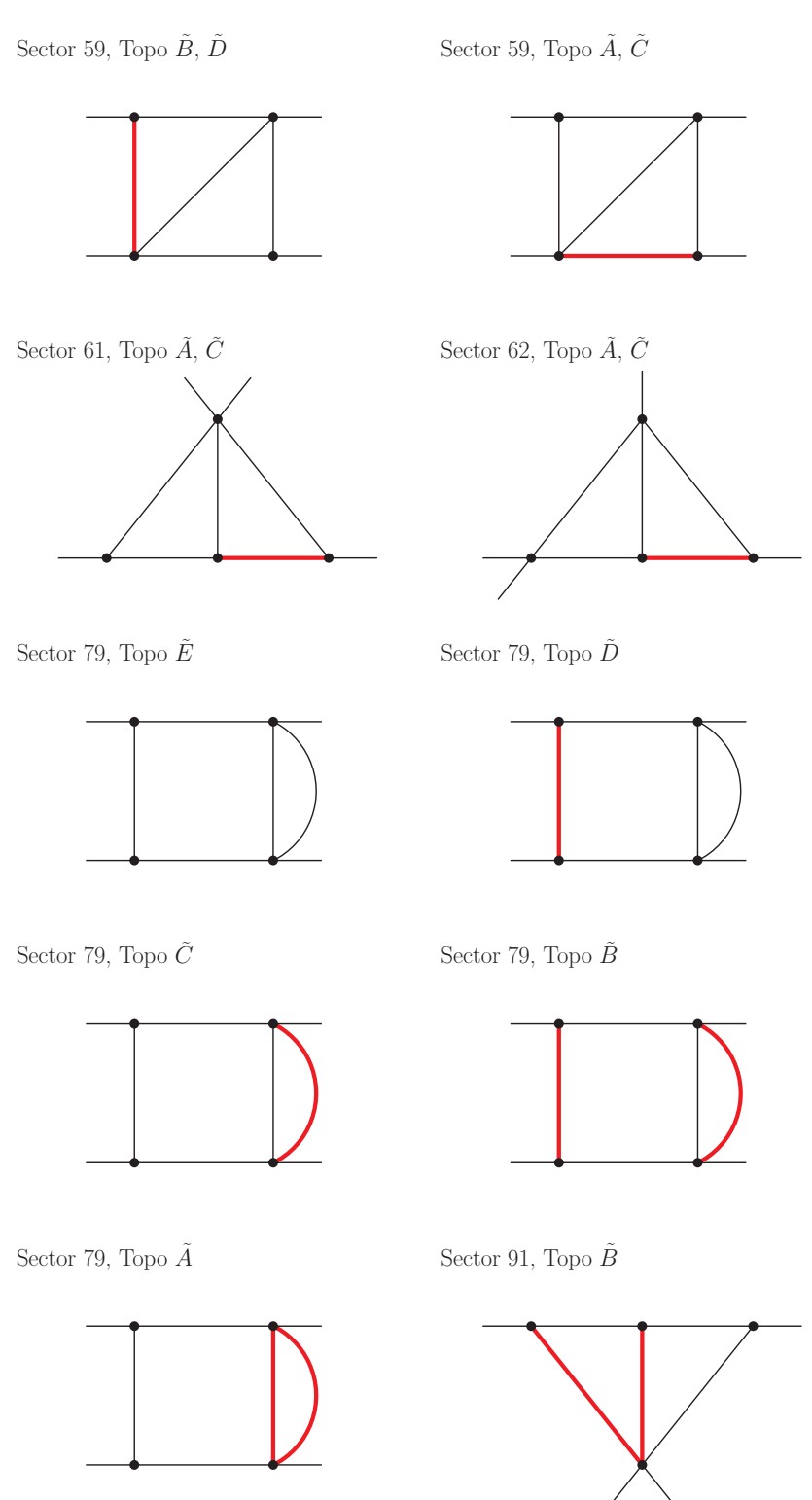

Figure 13: Master sectors for non-planar double-box integrals (part 4).

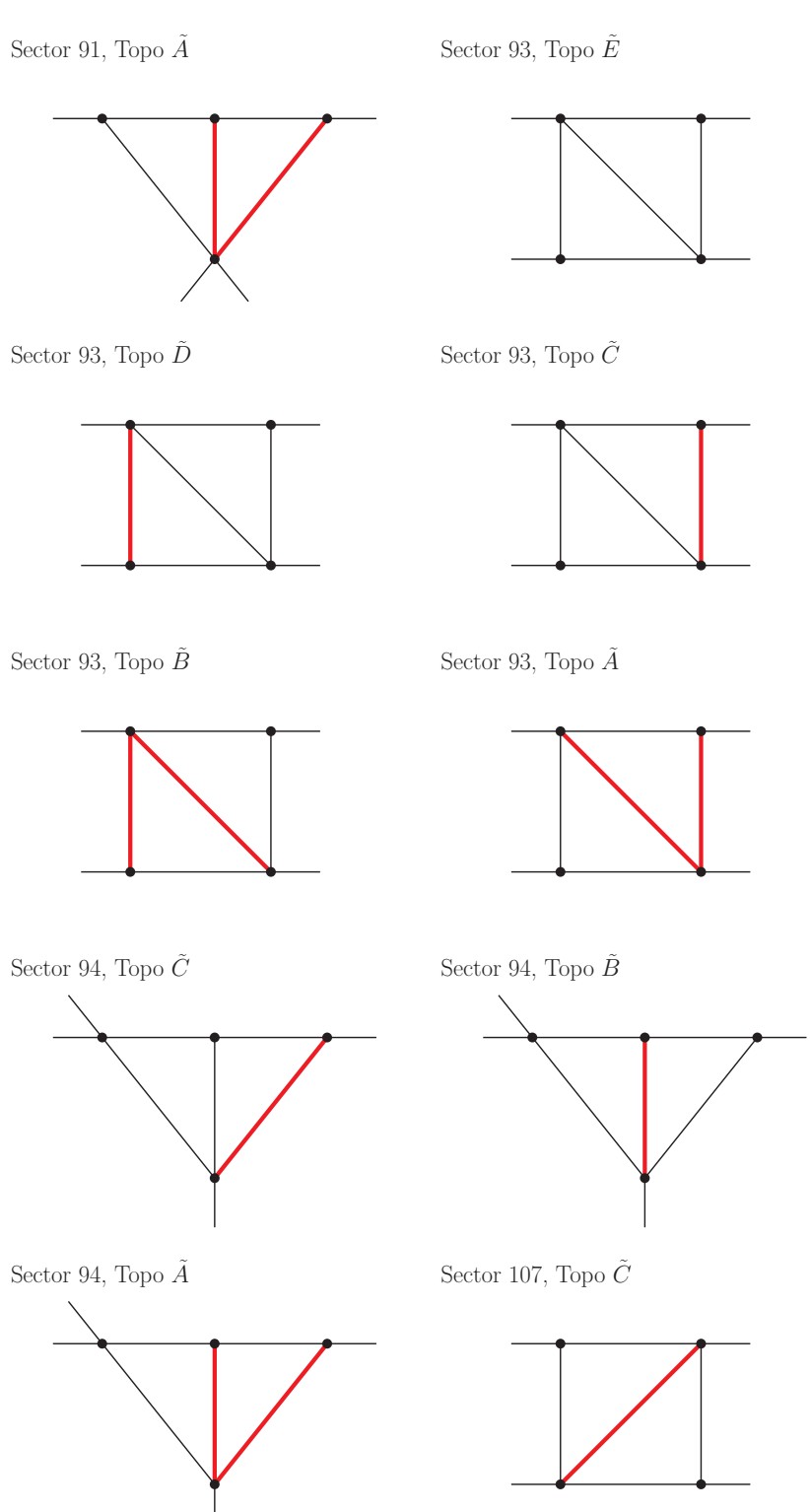

Figure 14: Master sectors for non-planar double-box integrals (part 5).

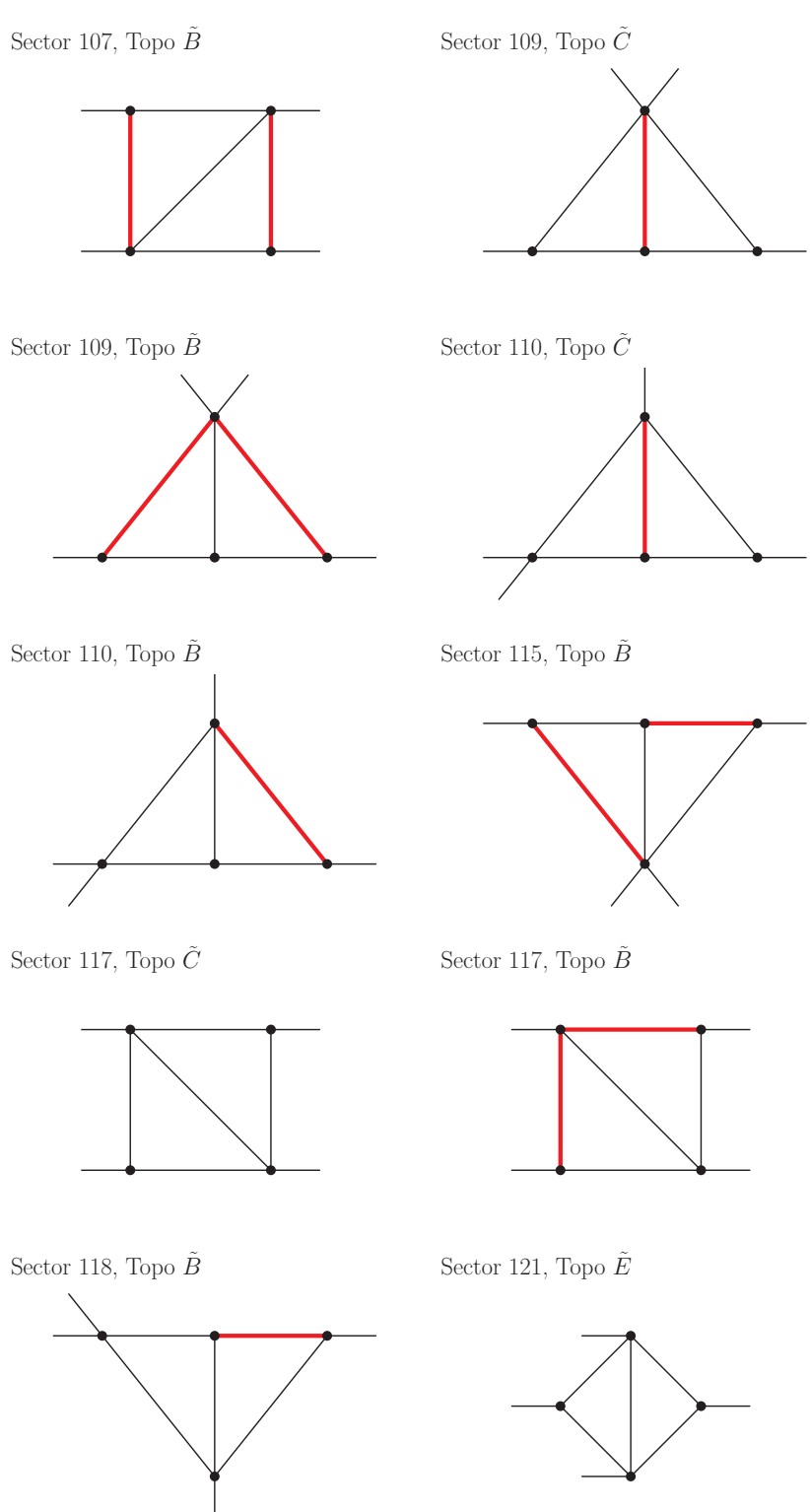

Figure 15: Master sectors for non-planar double-box integrals (part 6).

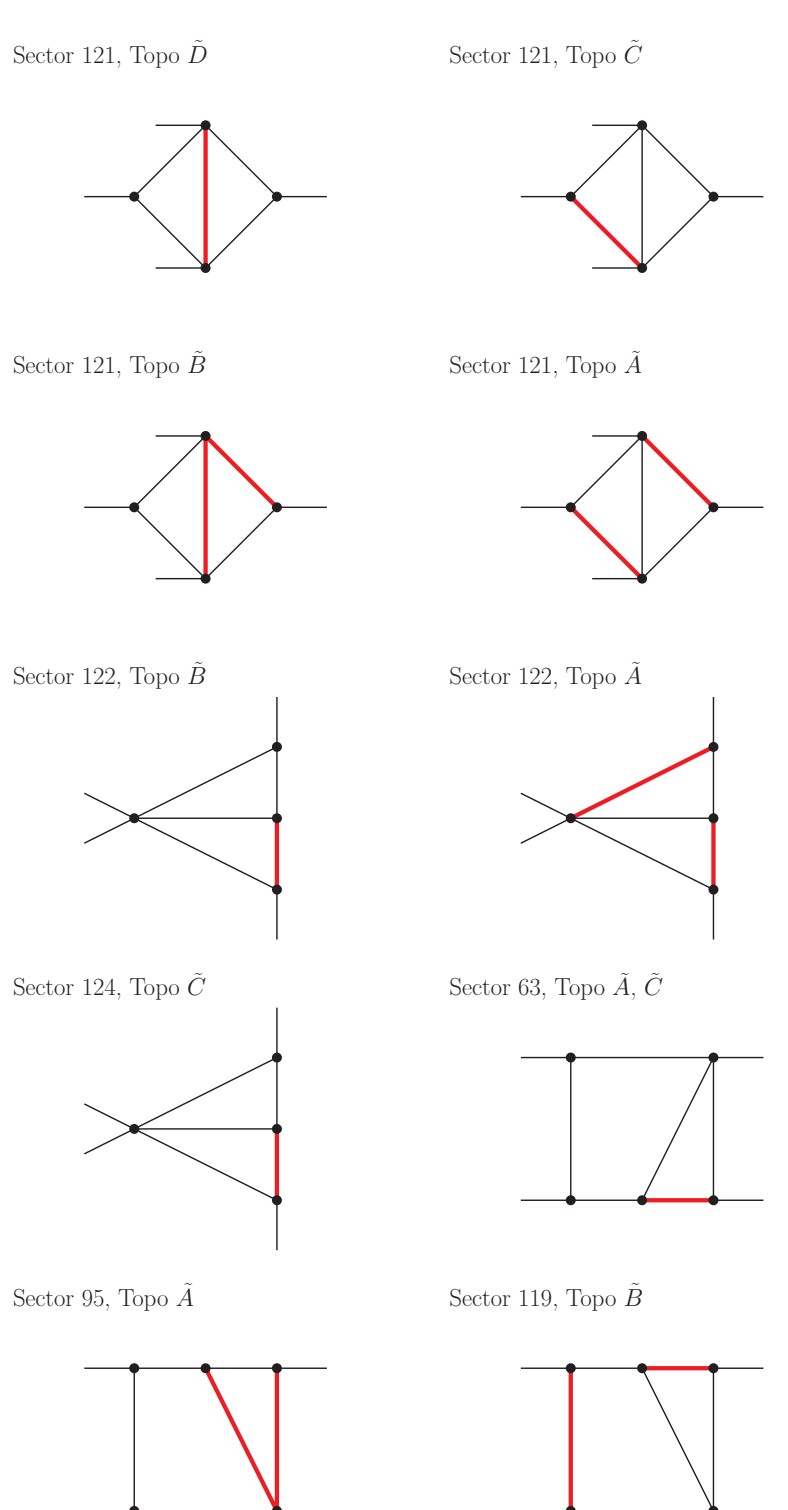

Figure 16: Master sectors for non-planar double-box integrals (part 7).

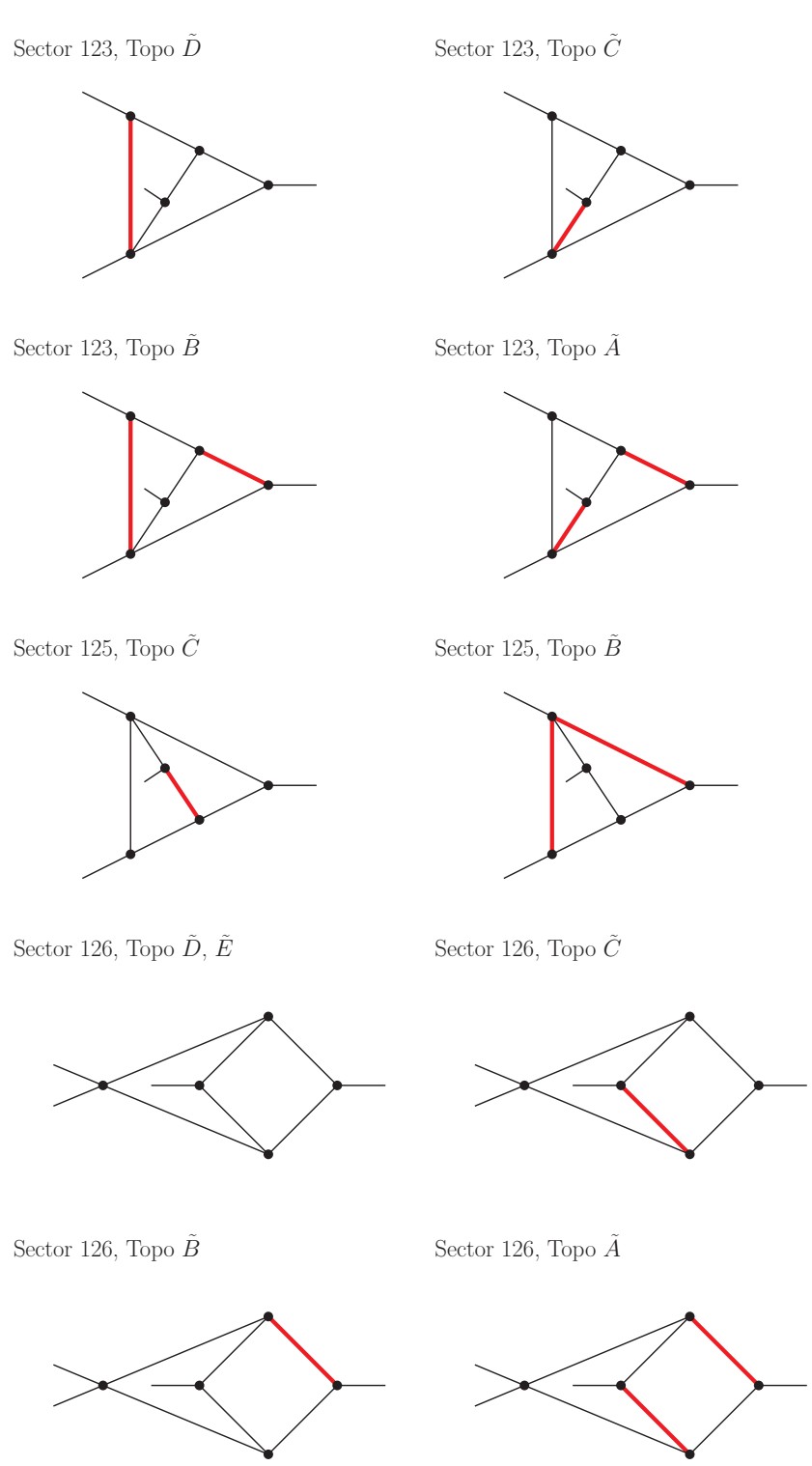

Sector 123, Topo $\tilde{D}$

Sector 123, Topo $\tilde{C}$

Sector 123, Topo $\tilde{B}$

Sector 123, Topo $\tilde{A}$

Sector 125, Topo $\tilde{C}$

Sector 125, Topo $\tilde{B}$

Sector 126, Topo $\tilde{D}$, $\tilde{E}$

Sector 126, Topo $\tilde{C}$

Sector 126, Topo $\tilde{B}$

Sector 126, Topo $\tilde{A}$

Figure 17: Master sectors for non-planar double-box integrals (part 8).

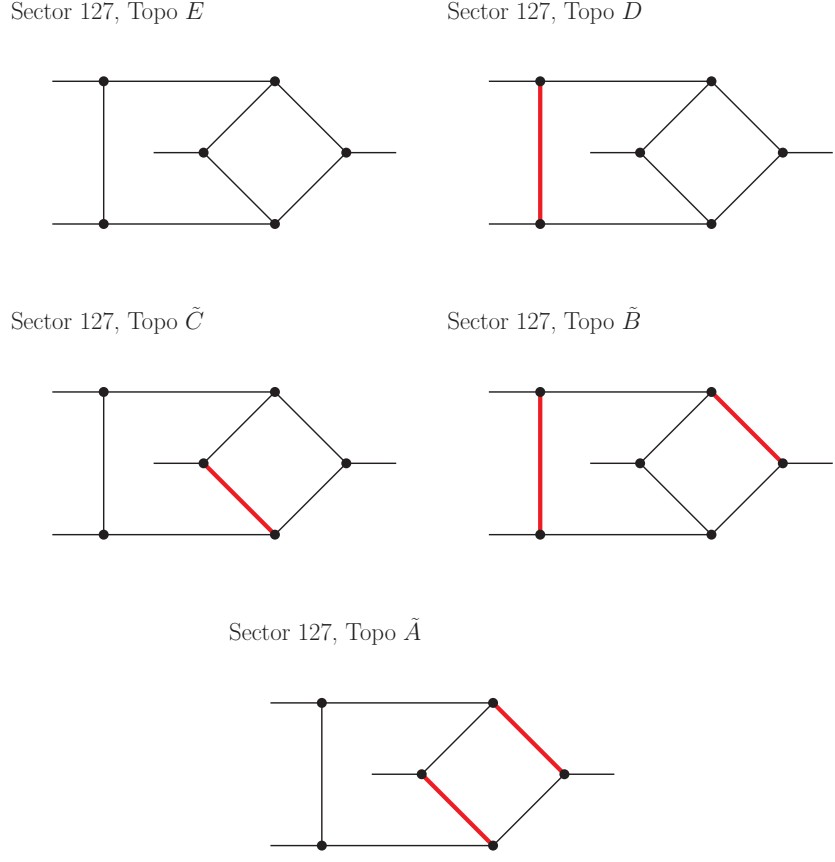

Figure 18: Master sectors for non-planar double-box integrals (part 9).

# B   Supplementary material

Attached to the arxiv version of this article are for each topology

$$X \;\in\; \big\{A,B,C,D,E,\tilde{A},\tilde{B},\tilde{C},\tilde{D},\tilde{E},\big\} \tag{62}$$

the electronic files

$$\texttt{topo\_X\_symbolic.mpl},\ \texttt{topo\_X\_numeric.cc}$$

and a common file

$$\texttt{input\_common.mpl}$$

The files `topo_X_symbolic.mpl` are in `Maple` syntax and define for each topology $X$ the transformation matrix $U^X$ appearing in eq. (14), its inverse $(U^X)^{-1}$, the pre-canonical basis $I^X$ appearing in the same equation (as pre-canonical basis $I^X$ we choose the `Kira` basis with integral ordering 5), the matrix $A^X$ appearing in the differential equation (15), the boundary values

$J_{\text{boundary}}^X$ at the boundary point defined by eq. (45), the alphabets for the various topologies (denoted by `alphabet_lst_X`), the explicit expressions for the differential one forms (given in `alphabet_subs_lst_X`) and the matrices $\tilde{M}_1^X$ determining the large logarithms (see section 6). These quantities are denoted as

```
U_X, Uinv_X, I_X, A_X, J_boundary_X,
alphabet_lst_X, alphabet_subs_lst_X, Mtilde_1_X.
```

The pre-canonical master integrals in the vector `I_X` are for example denoted as

```
I_1111111m10,
```

which stands for $I_{1111111(-1)0}$. The zeta value $\zeta_3$ is denoted by `zeta_3`. The square roots $r_i$, the periods $\psi_1^{(x)}$ and the functions $F_{ij}^X$ are denoted for example as

```
R1(s,t,m2), Psi1_a(s,t,m2), F_Atilde_39_38(s,t,m2),
```

and similar for the other ones. The file Maple `input_common.mpl` collects the relevant information on these functions. Without loss of generality we have set $\mu = 1$ in the Maple files.

The file `topo_X_numeric.cc` is a `C++`-program and provides numerical evaluation routines for all master integrals of a given topology. This `C++`-program requires the `GiNaC`-library [111].

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
