# Peer review of "Electroweak double-box integrals for Moller scattering"

_SciPost Physics_

## Round 1 · Referee Report · Anonymous (Referee 1) · 2025-1-27

Report

The authors discuss the analytic computation of a subset of the two-loop Feynman integrals required for the NNLO electroweak corrections to Møller/Bhabha scattering. They postpone the computation of the most complicated set of integrals$-$those with the exchange of three massive gauge bosons$-$to a future publication, and the methodology follows "to a large extent" that of ref. [79]. Moreover, most of the article is made of lengthy lists of formulae and numerical values that make it difficult to read. Nonetheless, the results are cutting-edge because of the appearance of types of special functions$-$elliptic functions$-$which are only beginning to be mastered, and the supplementary material contains useful examples of how to evaluate such functions numerically. For these reasons, I believe this article would meet SciPost's acceptance criteria, provided the authors address the following points of criticism.

Requested changes

My main point of criticism is that no detail at all is given about the construction of the bases of master integrals that satisfy canonical differential equations. This is the most difficult step, and the authors limit themselves to spelling out long lists of master integrals in section 3, spanning 21 pages but explaining nothing about them or about how they were chosen. This makes the paper difficult to read, and adds little value to it. I recommend that the authors move most of these lists to an appendix, and expand section 3 to at least outline the main steps of the procedure they followed.

Similarly, also the almost 13 pages of tables with the numerical values of the master integrals belong to an appendix and$-$most importantly$-$must be provided as machine-readable files to ensure reproducibility. The provided computer programs return the values in the tables, but benchmark values obtained by the authors themselves must be available as well so that the users can verify that the programs are working as expected.

In addition, there are a number of minor technical points that I think should be clarified.

1) In section 2.5, below eq. (32), the authors write: "For each elliptic curve we may express the derivatives of $\psi_i^{(X)}$ [...] as a linear combination of $\psi_i^{(X)}$ and $\frac{\partial}{\partial m^2}\psi_i^{(X)}$." I would like them to clarify how this is achieved in practice.

2) In section 4.1, the connection matrix is written as a linear combination of differential one-forms $\omega_j$, divided "into dlog-forms with rational arguments, dlog-forms with algebraic arguments and one-forms related to elliptic curves." How is the separation between dlog and non-dlog forms done? In other words, is it clear that no linear combination of one-forms related to elliptic curves can be expressed as the differential of a logarithm with a suitable and potentially unknown argument? Moreover, are the one-forms linearly independent over $\mathbb{Q}$ or some other field?

3) In section 4.2, the authors write "In addition we may always use the PSLQ algorithm to determine the boundary values from high-precision numerical results," but it is unclear whether they actually did that or not. If they did, how did they obtain the high-precision numerical results and how high was the precision?

4) The authors say nothing about whether they need to perform analytic continuation to evaluate their results numerically in the region of interest. As this can be highly non-trivial, especially for elliptic functions, the authors should specify whether and how they do it, or how they avoid it.

5) The notion of "canonical" differential equations is still somewhat under debate in the cases beyond multiple polylogarithms. One of the most commonly accepted definitions requires both the factorisation of $\epsilon$ and the presence of locally simple poles at all singular points. In other words, at every singular point there is always a way to choose the periods such that the connection matrices have no poles of order higher than one. Do the authors have any insight in this regard for the $\epsilon$-factorised differential equations obtained in this work? Any comment on this aspect would be of interest and add value.

6) In addition to the clear theoretical interest, the authors also refer to phenomenology as a motivation for this work. They should therefore clarify whether the results they provide are suitable to be used directly for phenomenology given the current efficiency/stability of the numerical evaluation, or whether further work$-$e.g., optimisations or the construction of interpolation grids$-$is required to this end, in addition to including the missing integral topologies.

7) Regarding the efficiency of the evaluation, the programs in the supplementary material have somewhat surprising relative timings. For example, the evaluation of topology $\tilde{B}$ appears to be faster than that of topology $A$ (27 s vs. 35 s), although the latter is in principle simpler, having fewer master integrals and containing only multiple polylogarithms. I think this behaviour calls for an explanation.

8) Regarding references, in section 1 the authors cite [80] and [81-83] for integration-by-parts identities and differential equation methods, respectively, whereas [87] and [91] should be cited as well, as they are in section 2.3.

Finally, I spotted a number of potential typos: - sec. 1: phenomenological relevant $\to$ phenomenologically relevant; - sec. 4.2 and 5: Moller $\to$ Møller; - sec. 4.2: we expand all differential one-forms in the variable $(w_{m^2}^{-1})^{25}$ $\to$ $w_{m^2}^{-1}$; - eq. (58): $m_Z^2$ $\to$ $m^2$.

Recommendation

Ask for major revision

---

## Round 1 · Referee Report · Anonymous (Referee 2) · 2025-2-3

Report

Dear Editor,

The article "Electroweak double-box integrals for Møller scattering" by N. Schwanemann and S. Weinzierl presents the calculation of a class of two-loop Feynman integrals relevant to four-fermion scattering. The authors focus in particular on Møller scattering and on contributions arising from electroweak (EW) corrections at second order. The main result is the analytic calculation of the master integrals involving up to two internal masses. Due to the presence of elliptic sectors, such calculations are notoriously challenging and may be at the frontier of current technology, especially when pursued analytically. The advantage of the latter approach is that it ensures higher performance and reliable results for phenomenological applications. Most of the results presented in the paper are novel and phenomenologically relevant. Furthermore, the authors provide them both in analytic form and through a numerical implementation, enhancing usability and accessibility. Although my general assessment is positive, before recommending publication in SciPost there are a few points I would like to address and possibly ask for a revision.

Comments on the general structure and style:

  • Sections 3 and 5 constitute a significant portion of the paper and contribute substantially to its overall length. I feel that they primarily contain highly technical information, which comes at the expense of readability. This is particularly true for Section 5, which consists of very little text but includes extensive tables that are mainly relevant to readers interested in a detailed comparison. I would suggest moving Section 5 to an appendix for better organisation.

  • Comment on style: The phrasing of the penultimate paragraph of the introduction (from "To compute the Feynman integrals..." to "The large logarithms are easily obtained from our full result.") could be refined. It currently reads as a sequence of brief, somewhat disconnected statements. I recommend rephrasing this section to improve coherence and flow.

  • Title of Section 2: "set-up" → "setup"

Comments on the article and requested changes:

  • References (14–77) provide an exhaustive list of works covering recent developments on elliptic integrals and their generalisations. Since an important aspect of this study involves finding suitable canonical integral bases, I suggest considering [2305.14090], which discusses an alternative procedure to the one used here, as well as [2412.02300], which appeared shortly before the current work and addresses similar issues.

  • The authors focus on Møller scattering, but one could argue that in different partonic configurations, the integrals considered here are also relevant for two-loop EW corrections to Bhabha and Drell-Yan scattering. These processes require s–t or s–u channel crossings. In fact, I would argue that Drell-Yan production, or quark-pair production in lepton collisions, would benefit from these results even more. I suggest adding a comment in the introduction on the choice to focus on Møller scattering, outlining its limitations and challenges, and situating the work within the broader context of EW corrections to four-fermion scattering.

  • In a similar context, I encourage the authors to clarify whether this subset of integrals contributes to a gauge-invariant subcomponent of the full scattering amplitude or not. While the problem is well-defined and the results are relevant, one could argue that the objects studied here do not necessarily enter a fully physically-motivated observable.

  • Another point of confusion concerns Figure 1, where the authors depict relevant double-box Feynman diagrams involving the exchange of up to two massive EW bosons. However, they appear to omit diagrams in which a photon is exchanged between two W bosons. In their notation, this would correspond to two "vertical" W bosons exchanged between the electron lines and a "horizontal" photon exchanged between the bosons. Why are such contributions excluded? If the omission is due to a strict definition of "double-box" that excludes such diagrams, this is quite a technical convention and should be explicitly stated in the text. Otherwise, the rationale for their exclusion should be clearly explained. Additionally, in the introduction, the authors mention that cases with two different masses are excluded due to initial- and final-state configurations, but this is also related to their restriction to up to two massive bosons. For instance, a Z boson could be exchanged between the Ws. This explanation should be made more precise.

  • At the beginning of page 3, the authors discuss the role of large logarithmic terms. I suggest specifying what type of logarithms they are referring to. More generally, in relation to Section 6, these logarithms are unlikely to be dominant at the observable level in low-energy experiments such as Møller scattering. Therefore, I find the motivation at the beginning of Section 6 unconvincing. The authors state that these logarithms are of particular interest, but one could argue—perhaps provocatively—that they are not particularly relevant in the context of low-energy scattering. I do, however, acknowledge that controlling these terms is necessary for obtaining numerically reliable results at the amplitude level, and given the high precision of experimental data, they could still play a role. I suggest rephrasing this section to provide a more substantiated motivation for the analytical control of these logarithms.

  • In the "kinematics" section, the authors write: "The Mandelstam variables are defined as usual by..." I recommend removing "usual", as the definition of the invariant t is not universally standard; in some conventions, t refers to (p1+p3)^2 rather than (p2+p3)^2. Additionally, I suggest moving Eq. (5) earlier, as it provides the justification for the approximation that in low-energy scattering, contributions from the exchange of multiple massive bosons are suppressed.

  • I also have some questions regarding the numerical integration of the elliptic integrals, particularly towards the end of Section 4. Do the authors precompute a suitable series expansion for topologies A~ and B~, implementing the expansion up to a fixed order (unlike for topology A)? Do they require different expansions for different regions of phase space? How do they ensure good convergence and accuracy across the full physical region? Maybe discussing such points in greater detail would inform the reader better.

Summary:

The article is well written and presents novel and physically relevant results, particularly in light of future high-precision studies of low-energy scattering experiments. However, I believe it could be further improved by addressing the comments outlined above. I therefore cannot recommend publication of the manuscript in its current form but would certainly be willing to reconsider this assessment after a minor revision.

Recommendation

Ask for minor revision

---

## Editorial Decision

resubmitted